JCB Journal of Cell Biology

# Conformational transitions of the Spindly adaptor underlie its interaction with Dynein and Dynactin

Ennio A. d'Amico[1], Misbha Ud Din Ahmad[2]*, Verena Cmentowski[1,3]*, Mathias Girbig[4], Franziska Müller[1], Sabine Wohlgemuth[1], Andreas Brockmeyer[5], Stefano Maffini[1], Petra Janning[5], Ingrid R. Vetter[1], Andrew P. Carter[4], Anastassis Perrakis[2], and Andrea Musacchio[1,3]

**Cytoplasmic Dynein 1, or Dynein, is a microtubule minus end–directed motor. Dynein motility requires Dynactin and a family of activating adaptors that stabilize the Dynein–Dynactin complex and promote regulated interactions with cargo in space and time. How activating adaptors limit Dynein activation to specialized subcellular locales is unclear. Here, we reveal that Spindly, a mitotic Dynein adaptor at the kinetochore corona, exists natively in a closed conformation that occludes binding of Dynein–Dynactin to its CC1 box and Spindly motif. A structure-based analysis identified various mutations promoting an open conformation of Spindly that binds Dynein–Dynactin. A region of Spindly downstream from the Spindly motif and not required for cargo binding faces the CC1 box and stabilizes the intramolecular closed conformation. This region is also required for robust kinetochore localization of Spindly, suggesting that kinetochores promote Spindly activation to recruit Dynein. Thus, our work illustrates how specific Dynein activation at a defined cellular locale may require multiple factors.**

## Introduction

Eukaryotic cells maintain their internal order through the concerted action of a variety of functionally diverse energy-harnessing enzymes. Among these are molecular motors that convert the chemical energy of ATP into mechanical work to dispatch various cargoes to different subcellular locations (Klinman and Holzbaur, 2018). Many molecular motors move along microtubules, polarized cellular tracks with plus and minus ends, the latter normally localized near microtubule-organizing centers, such as centrosomes. Motors use microtubules to transport cargoes of various sizes, ranging from individual protein complexes, to viruses, to organelles. Molecular motors also move chromosomes and transport, crosslink, and reciprocally slide microtubules to promote the assembly of the mitotic spindle in mitotic cells (Pavin and Tolic, 2021). The two main classes of intracellular molecular motors are the kinesins and Dynein. Kinesins populate a wide and diverse family of motors, prevalently with plus end–directed polarity, with each family member specializing in the transport of distinct cargoes (Klinman and Holzbaur, 2018). Cytoplasmic Dynein-1 (Dynein), on the other hand, is a minus end–directed multi-subunit assembly whose motor subunit, Dynein heavy chain (DHC), is encoded by a single gene. Its association with different

cargoes relies, therefore, on various activating adaptors, each with a distinct cargo preference (Canty et al., 2021; Reck-Peterson et al., 2018; Roberts et al., 2013).

In humans, the 1.4 MDa Dynein complex consists of 12 subunits, with six different polypeptides all present in two copies, including the DHC, the intermediate chains, the light intermediate chains (LIC), and the three Dynein light chains LC8, Roadblock, and Tctex (Reck-Peterson et al., 2018). Dynein motility requires the 1.1 MDa complex Dynactin (Carter et al., 2016; Reck-Peterson et al., 2018). Dynactin consists of 23 polypeptides and 11 individual subunits, organized in four main structural domains: (1) a central actin-like filament consisting of eight ARP1 subunits and one actin; (2) a four-subunit pointed-end (PE) capping complex including the subunits p25, p27, p62, and Arp11; (3) a barbed-end capping complex containing CapZαβ; and (4) a shoulder domain containing p24, p150[glued], and p50/dynamitin (Chowdhury et al., 2015; Lau et al., 2021; Urnavicius et al., 2015).

The interaction of Dynein and Dynactin (DD) is weak but strongly promoted by activating cargo adaptors (Hoogenraad and Akhmanova, 2016; Olenick and Holzbaur, 2019; Reck-Peterson et al., 2018). The prototypical adaptor, BICD2 (bicaudal

[1]Department of Mechanistic Cell Biology, Max Planck Institute of Molecular Physiology, Dortmund, Germany;   [2]Oncode Institute and Department of Biochemistry, The Netherlands Cancer Institute, Amsterdam, The Netherlands;   [3]Centre for Medical Biotechnology, Faculty of Biology, University Duisburg-Essen, Essen, Germany;   [4]MRC Laboratory of Molecular Biology, Cambridge, UK;   [5]Department of Chemical Biology, Max-Planck-Institute of Molecular Physiology, Dortmund, Germany.

*M. Ud Din Ahmad and V. Cmentowski contributed equally to this paper.   Correspondence to Andrea Musacchio: andrea.musacchio@mpi-dortmund.mpg.de

E.A. d'Amico's present address is MRC Laboratory of Molecular Biology, Cambridge, UK.   M. Ud Din Ahmad's present address is ZoBio BV, Leiden, The Netherlands.   M. Girbig's present address is European Molecular Biology Laboratory, Structural and Computational Biology Unit, Heidelberg, Germany.

D homologue 2), favors the incorporation of DD into a single complex with greatly increased motility and processivity in comparison with isolated Dynein (McKenney et al., 2014; Schlager et al., 2014a; Schlager et al., 2014b; Splinter et al., 2012; Urnavicius et al., 2015). The DD-binding segment maps to the BICD2 N-terminal region, which forms an apparently uninterrupted dimeric coiled-coil of ≈250 residues that bind alongside the Dynactin filament. The BICD2 N-terminal region makes contacts near the barbed end, while a more C-terminal region of BICD2 makes contacts near the PE. This arrangement promotes binding to Dynein through several contacts with the N-terminal tail domain of the DHC as well as with the C-terminal region of the LICs (Celestino et al., 2019; Gama et al., 2017; Lee et al., 2020; Lee et al., 2018; Renna et al., 2020; Schroeder et al., 2014; Schroeder and Vale, 2016; Urnavicius et al., 2018; Urnavicius et al., 2015; Yeh et al., 2012).

In addition to the paradigmatic BICD2 adaptor and its family, several other proteins are or are likely to be activating adaptors, including CCDC88B, FIP3, HAP1, HOOK1-3, JIP3, Ninein and Ninein-like, NuMa, RILP, Spindly (SPDL1), and TRAK1 (Gama et al., 2017; Hueschen et al., 2017; McKenney et al., 2014; Olenick et al., 2016; Redwine et al., 2017; Renna et al., 2020; Schroeder and Vale, 2016). At least some of these adaptors promote the interaction of Dynactin with two Dynein dimers, which increases processivity, speed, force production, and unidirectional movement (Grotjahn et al., 2018; Urnavicius et al., 2018). Collectively, the interactions of Dynein with Dynactin and adaptors appear to induce large conformational changes that align the motor domains for concomitant binding to microtubules (Zhang et al., 2017), an effect that likely extends to both Dynein dimers, when present.

While diverse, most adaptors share at least six structural and functional features. First, adaptors show propensity to form long coiled-coils with a dimeric parallel organization (Olenick and Holzbaur, 2019). Second, the N-terminal region of adaptors interacts specifically with a conserved helix of LIC isoforms (Celestino et al., 2019; Gama et al., 2017; Lee et al., 2020; Lee et al., 2018). LIC-interacting sequences on adaptors belong to at least three different subfamilies, containing either a CC1 box, a HOOK domain, or EF-hand pairs (Olenick and Holzbaur, 2019; Reck-Peterson et al., 2018), which bind the LIC in different manners (Lee et al., 2020; Lee et al., 2018). The CC1 box (Fig. 1 A and Fig. S1 A) encompasses a highly conserved AAXXG sequence, where X denotes any amino acid (Gama et al., 2017; Hoogenraad and Akhmanova, 2016; Lee et al., 2018; Schlager et al., 2014b). Third, adaptors contain a second conserved motif in the CC1 coiled-coil, the CC2 box, that is adjacent to the CC1 box and that binds to the DHC (Sacristan et al., 2018; Fig. 1 A and Fig. S1 A). Here, we refer to this motif as HBS1 (for heavy chain binding site 1), as we have gathered evidence that it interacts with the DHC (Chaaban and Carter, 2022 Preprint; Lau et al., 2021). Fourth, many adaptors contain a more C-terminal Spindly box motif (LΘXEΘ, where Θ indicates an aliphatic or aromatic side chain; Fig. 1 A and Fig. S1 B), which mediates binding to the four subunits of the Dynactin PE subcomplex (p25, p27, p62, Arp11; Gama et al., 2017; Lau et al., 2021). Fifth, an extended stretch of coiled-coil positioned between the CC1 box and the Spindly motif of adaptors (and corresponding to CC1), lodges between DD mini-filament, strongly enhancing complex formation (Chowdhury et al., 2015; Urnavicius et al., 2018; Urnavicius et al., 2015). Sixth, a C-terminal domain after the DD-binding region was shown or hypothesized to bind to cargo (Akhmanova and Hoogenraad, 2015; Hoogenraad et al., 2003).

The exquisitely spatial and temporal regulation of Dynein activation implies that the interaction with adaptors is tightly regulated. Dynein in solution adopts a so-called phi-particle conformation unable to interact with Dynactin and adaptors (Amos, 1989; Zhang et al., 2017). The Dynactin p150 subunit can dock onto the PE subcomplex, sterically preventing adaptor binding (Lau et al., 2021; Urnavicius et al., 2015). Finally, the cargo-binding domains of several adaptors, and prominently of BICD2, may fold back onto the Dynein-binding domains, forcing an autoinhibited conformation that can be relieved either by cargo-binding, or by removing the cargo-binding domain in recombinant protein (Hoogenraad et al., 2003; Liu et al., 2013; Splinter et al., 2012; Stuurman et al., 1999; Terawaki et al., 2015; Urnavicius et al., 2015).

Here, we addressed the organization and regulation of Spindly (Fig. 1 A), a farnesylated mitotic regulator of DD (605 residues in humans). In early mitosis, Spindly promotes recruitment of DD to kinetochores, the structures that connect chromosomes to spindle microtubules (Musacchio and Desai, 2017). This function of Spindly is enabled by the 800-kD hexameric ROD–Zwilch–ZW10 (RZZ) cargo complex, to which Spindly binds directly through its farnesylated C-terminal region (Holland et al., 2015; Mosalaganti et al., 2017; Moudgil et al., 2015). At kinetochores, phosphorylation by the MPS1 kinase promotes the polymerization of the RZZ–Spindly (RZZS) complex into a mesh (Fig. 1 B), the kinetochore corona, which adopts a characteristic crescent shape and contributes to the initial phases of chromosome alignment that precede end-on microtubule attachment and chromosome bi-orientation (Kops and Gassmann, 2020; Magidson et al., 2015; Pereira et al., 2018; Raisch et al., 2022; Rodriguez-Rodriguez et al., 2018; Sacristan et al., 2018). Besides DD, the corona also recruits the MAD1:MAD2 complex (Barisic et al., 2010; Chan et al., 2009; Cheerambathur et al., 2013; Gassmann et al., 2008; Gassmann et al., 2010; Griffis et al., 2007; Rodriguez-Rodriguez et al., 2018; Starr et al., 1998; Yamamoto et al., 2008), a central component of the spindle assembly checkpoint (SAC), which synchronizes cell-cycle progression with the completion of chromosome alignment (Kops and Gassmann, 2020; Musacchio, 2015). Upon achievement of end-on attachment, the DD–RZZS complex becomes active and moves from kinetochores to spindle poles, causing corona disassembly—a process known as "corona shedding" or "stripping" (Auckland et al., 2020; Basto et al., 2004; Howell et al., 2001; Mische et al., 2008; Sivaram et al., 2009; Varma et al., 2008; Williams et al., 1996; Wojcik et al., 2001). As the MAD1:MAD2 complex is removed from kinetochores together with DD–RZZS, stripping also suppresses SAC signaling. Mutations in the Spindly motif abrogate kinetochore recruitment of DD, blocking concomitantly corona shedding and SAC silencing (Cheerambathur et al., 2013; Gassmann et al., 2010).

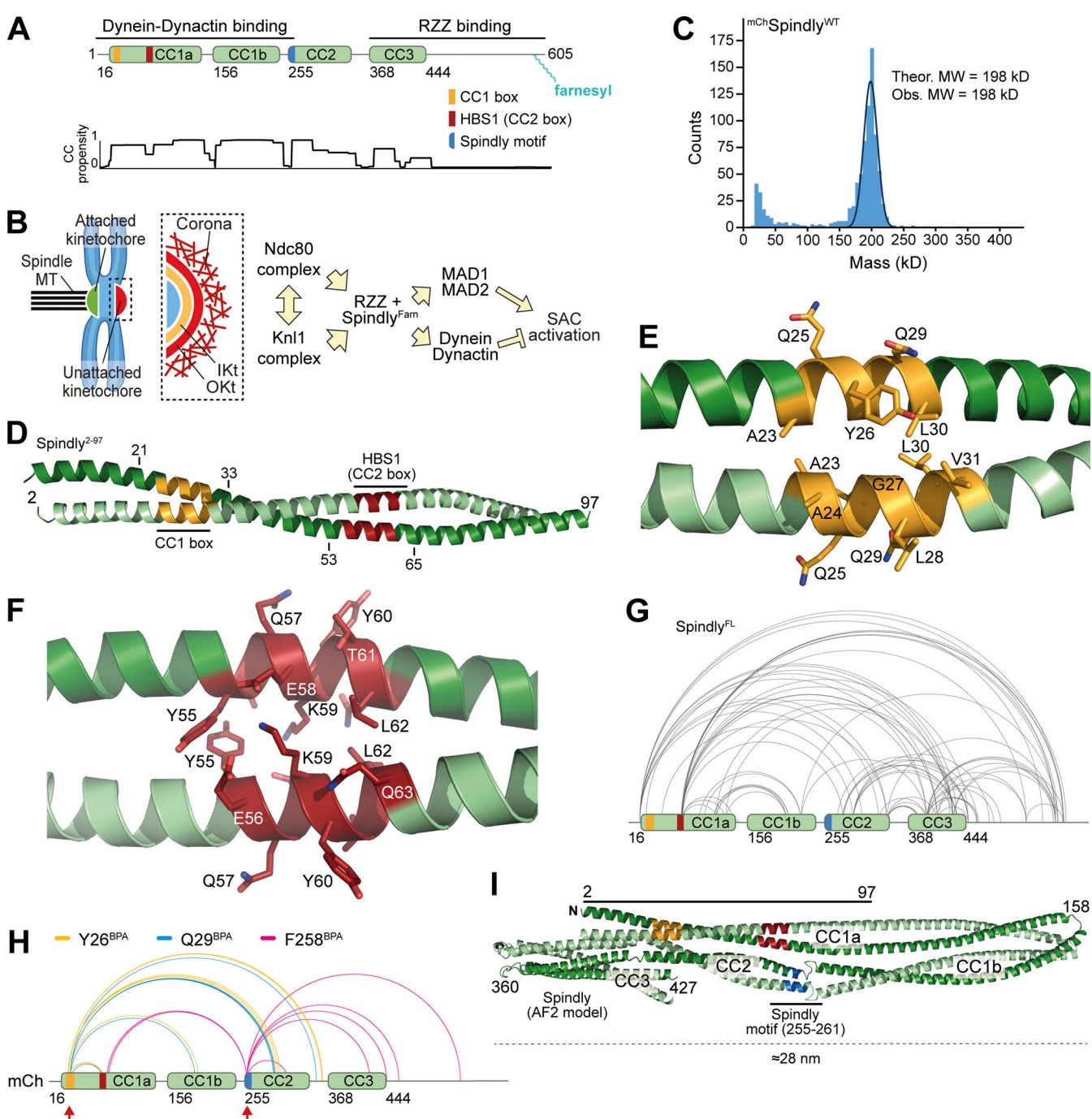

Figure 1. **Spindly is a folded adaptor. (A)** Schematic representation of the organization of the coiled-coil regions of Spindly, and relevant coiled-coil prediction (COILS, ExPaSy suite). **(B)** Organization of the kinetochore and corona. **(C)** Results of mass photometry measurement of a sample of mCh-tagged Spindly (mChSpindlyFL). The measurements are consistent with Spindly being a dimer in solution, even at very low concentration (10 nM). MW, molecular weight. **(D)** Crystallographic structure of a Spindly1–100 construct. Only residues 2–97 were visible in the electron density. **(E)** Structure of the Spindly CC1 box (orange) and surrounding sequence. **(F)** Structure of the Spindly HBS1 (also known as CC2 box; red) and surrounding sequence. **(G)** Summary of XL-MS data reporting Spindly intramolecular crosslinks. For ease of viewing, only crosslinks detected ≥3 times and involving sites ≥40 residues apart are depicted. See also Table S2 for a detailed list of all crosslinks. **(H)** Summary of XL-MS data reporting Spindly intramolecular crosslinks found through amber codon suppression experiments. Red arrows indicate the sites where the BPA residues were introduced. Crosslinking results from three mutants are merged: Y26BPA (orange), Q29BPA (cyan), and F258BPA (magenta). A few crosslinks identified between the BPA residues and the mCh tag were considered spurious and not displayed. See also Table S2 for a detailed list of crosslinks. **(I)** AF2 Multimer prediction of Spindly structure. The CC1 box is in orange, the HBS1 in red, the Spindly motif in blue. The C-terminal unstructured tail of Spindly (aa 440–605) was omitted from the model due to the very low confidence index (pLDDT) of the prediction for this region (unpublished results).

Spindly has all the sequence credentials of a bonafide DD-activating adaptor (Gama et al., 2017; Fig. 1 A); however, it activates DD only weakly in motility assays in vitro (McKenney et al., 2014). This suggests that Spindly may exist in auto-inhibited and active forms, as its presence alone is not sufficient for DD activation. Previous studies have also highlighted interactions between the N-terminal and C-terminal domains on Spindly, supporting the idea of an autoinhibitory interaction that hinders DD binding (Mosalaganti et al., 2017; Sacristan et al., 2018). Here, we identify crucial intramolecular contacts that regulate Spindly autoinhibition to prevent binding to DD. Relief of Spindly autoinhibition causes large conformational changes and requires the interaction with the RZZ and an additional kinetochore trigger that can be bypassed mutationally. These results have important general implications for the mechanism of DD activation.

## Results

### Spindly adopts a complex dimeric structural organization

We have previously shown through solution scattering studies and hydrodynamic analyses that Spindly is elongated and possibly a dimer in solution (Mosalaganti et al., 2017; Sacristan et al., 2018). Indeed, mass photometry identified recombinant full-length Spindly (indicated as Spindly[FL] or Spindly[WT]) as a dimer, with an excellent agreement between theoretical and observed molecular masses (Fig. 1 C). We then determined the crystal structure of human Spindly[1–100] (Table S1; only residues 2–97 were clearly resolved in the electron density), a fragment containing both the CC1 box and the HBS1 (Fig. 1, D–F). Spindly[1–100] forms a parallel dimeric coiled-coil, similar in its outline to that observed in structures of other adaptors captured in complex with DD (Lau et al., 2021; Urnavicius et al., 2018; Urnavicius et al., 2015). Thus, both Spindly[FL] and an N-terminal segment of Spindly are stable dimers. The structure of Spindly[2–97] is closely reminiscent of the structure of BICD2[1–98] in complex with a peptide encompassing the LIC1 helix (residues 433–458, Protein Data Bank [PDB] accession no. 6PSE [Lee et al., 2020]; Fig. S1 C). In Spindly, Ala23 and Gly27 in the CC1 box occupy $a$ and $d$ positions within the coiled-coil's heptad repeats, similarly to Ala43 and Gly47 in BicD2, a pattern also conserved in BICDL1 and BICD1 (Fig. S1 A). This unusual composition for $a$ and $d$ residues generates a cavity along the BICD2 coiled-coil axis that interacts with aromatic and hydrophobic LIC side chains (Lee et al., 2020). Its conservation in Spindly was also supported by a high-confidence prediction by AlphaFold2 (AF2) in the variants Colabfold and AF2-Multimer (Evans et al., 2021 Preprint; Jumper et al., 2021; Mirdita et al., 2021 Preprint), which only became available during the final phases of this study (Fig. S1 D). Indeed, Spindly and BICD2 interact with the LIC with similar affinity (Lee et al., 2020; Lee et al., 2018).

Structures of complexes of DD with various adaptors demonstrated the existence of an uninterrupted coiled-coil spanning the distance between the LIC-binding CC1 box in BICD2 and a coiled-coil break that immediately precedes the Spindly motif. The coiled-coil, referred to as coiled-coil 1 (CC1), is cradled between DD (Urnavicius et al., 2018; Urnavicius et al., 2015). In Spindly, the coiled-coil propensity between the CC1 box and the break immediately preceding the Spindly motif (around residue 256) is generally high, but there is a conserved two- or three-residue insertion around residue 155 that coincides with an interruption of the register of CC1 not expected in BICD1 and BICD2 (Fig. S1, E and F). Conservation of this feature in the Spindly family prompted us to investigate the possibility that the CC1 coiled-coil of Spindly splits into distinct segments (CC1a and CC1b). For this, we subjected full-length Spindly to crosslinking-mass spectrometry (XL-MS) experiments with the bifunctional crosslinker DSBU (disuccinimidyl dibutyric urea; Pan et al., 2018). This revealed multiple intramolecular contacts within Spindly, including "concentric" crosslinks of the putative CC1b region with the second half of CC1a, consistent with the idea that they may be arranged as an anti-parallel pair separated by a loop containing the conserved insertion around residues 154–155. In addition, we observed crosslinks of the first half of CC1a with CC2 and CC3, and of CC2 with CC3 (Fig. 1 G and Fig. S1 G and Table S2). Essentially identical crosslinks were observed in experiments with Spindly[1–440] (lacking the flexible C-terminal region that contributes to binding the RZZ complex [Mosalaganti et al., 2017]), with the expected exception of contacts involving the C-terminal disordered region (Fig. S1, H and I).

Thus, at least in isolation, Spindly may adopt a compact conformation relative to the extended conformations observed for adaptors bound to DD (Urnavicius et al., 2018; Urnavicius et al., 2015). To probe this, we harnessed amber codon suppression to introduce the UV-photoactivatable crosslinker $p$-Benzoyl-L-phenylalanine (BPA; Ai et al., 2011; Davis and Chin, 2012) into selected positions of an mCherry (mCh)-tagged construct of Spindly ([mCh]Spindly). These included Tyr26 and Gln29 (Y26[BPA] and Q29[BPA], respectively) in the CC1 box and Phe258 (F258[BPA]) in the Spindly box. After irradiation with UV light (Fig. S1, J and K), Y26[BPA] and Q29[BPA] generated largely equivalent crosslinking patterns, with a majority of targets near the center of the predicted CC2 coiled-coil, around residue 300 (e.g., K297, L298, Q299, I300, L303, and M306; Fig. 1 H and Table S2 and Fig. S1 L). F258[BPA], on the other hand, crosslinked to residues E74 and L76 immediately after the CC1 HBS1 in the central half of CC1a, as well as to residues in CC3 and the tail (Fig. 1 H and Table S2). Collectively, the XL-MS results confirm Spindly is a dimer that folds as four interacting coiled-coil segments (CC1a, CC1b, CC2, and CC3), with extensive interactions of the first half of CC1a with CC2 and CC3, and of the second half of CC1a with CC1b. A possible confounding factor, however, is that Spindly may form higher order oligomers at the low micromolar concentrations of the crosslinking experiments, so that the observed crosslinks may reflect inter-dimer rather than intra-dimer contacts. To address this, we diluted the crosslinked samples, and measured their mass with mass photometry (Fig. S1, M and N). The crosslinked samples remained dimeric and were essentially indistinguishable from the untreated controls.

We used AF2 to rationalize these observations (Fig. 1 I and Fig. S2). AF2 models depict Spindly as having a complex organization, where CC1 is almost invariably predicted to be

interrupted around residues 154–155, giving rise to CC1a and CC1b coiled-coil segments. The ≈23 nm CC1a coiled-coil, partly captured in our crystal structure (Fig. 1 D), is predicted to encompass the majority of the long axis of the Spindly dimer (≈28 nm), in good agreement with values derived from 2D class averages of negatively stained Spindly samples and small-angle x-ray scattering experiments for both Spindly[FL] and Spindly[1–440] (Sacristan et al., 2018). CC1b packs against CC1a in such a way that the Spindly motif, located at the beginning of CC2, is positioned roughly halfway along the complex, facing the segment immediately C-terminal to the HBS1 (Fig. 1 I and Fig. S2), in excellent agreement with the BPA crosslinking experiments. A loop around residue 360 separates CC2 and CC3, so that both CC2 and CC3 are in contact with the first part of CC1. The CC1 box faces precisely the region centered on residue 300 identified by crosslinking experiments with Y26[BPA] and Q29[BPA]. Importantly, the model predicts that both the CC1 box and the Spindly motif will be largely inaccessible to DD. While all parallel coiled-coils are roughly symmetric, residues in the two chains experience different environments due to the asymmetric intramolecular interactions of the coiled-coils. Thus, the folded structure of Spindly predicted by AF2 is inherently asymmetric.

## Spindly is not accessible to DD

A targeted AF2 analysis of BICD2, BICDL1, HOOK1, HOOK3, and TRAK1 confirmed that, unlike Spindly, their CC1 coiled-coil continues more or less uninterrupted until a break of variable length (where coiled-coil propensity drops). This precedes CC2, which usually begins with, or is even preceded by, the Spindly motif (Fig. S3 A). This is true also of BICD2 (Fig. 2 A), even if a coiled-coil prediction algorithm suggested a ≈30-residue drop in coiled-coil propensity after the CC1 HBS1 (Fig. S1 E). The AF2 models show that the CC1 in all these adaptors has a rather regular length of 35–39 nm. In other adaptors, including JIP3 and RILP, CC1 is considerably shorter (∼20 nm) and there is no obvious Spindly motif (Celestino et al., 2022; Reck-Peterson et al., 2018). As established in Figs. 1 and S2, Spindly can also adopt a closed compact conformation, but its constellation of DD-binding motifs predicts that it opens as a canonical adaptor under appropriate conditions. Indeed, AF2 predicted a closed conformation for Spindly[1–309] (Fig. 2 B and Fig. S2), while it did not predict convincing intramolecular interactions for Spindly[1–275] (Fig. 2 C and Fig. S2). This suggests that residues 276–309, predicted by our XL-MS analysis to face the CC1 box, contain determinants of a conformational transition from a closed to an open form.

In this context, a question of significant mechanistic relevance is whether binding of Spindly to its cargo is sufficient to relieve autoinhibition and trigger the formation of a complex with DD, as postulated for other adaptors (Olenick and Holzbaur, 2019; Terawaki et al., 2015). To address this, we used analytical size-exclusion chromatography (SEC) to monitor the formation of complexes of DD with either Spindly or BICD2[1–400], which served as positive control (Schlager et al., 2014a). Because the Dynein phi-particle might prevent Dynein from engaging into complexes with Dynactin (purified from pig brain, and indicated as [PB]Dynactin) and adaptors, we used a Dynein tail construct

(CDHC[1–1455]) that does not form the phi-particle (Urnavicius et al., 2015; Zhang et al., 2017). As expected, a large fraction of Dynein[tail], [PB]Dynactin, and BICD2[1–400] interacted in a complex that eluted before any of the individual components, indicative of an increased Stokes' radius (Fig. 2 D). Conversely, very little farnesylated full-length Spindly (Spindly[F]) entered a complex with Dynein[tail] and [PB]Dynactin (Fig. 2 E). This result is consistent with the hypothesis that Spindly adopts an auto-inhibited conformation refractory to interact with DD in the absence of adequate triggers. The RZZ complex, to which Spindly[F] binds directly, mediates Spindly's kinetochore recruitment and can therefore be considered Spindly's cargo, or a connector of Spindly to its chromosome cargo. We asked therefore if Spindly[F] interacted with DD in the presence of the RZZ complex. As the RZZ complex and Spindly are both known to be phosphorylated, with phosphorylation being critical for corona function (Raisch et al., 2022; Rodriguez-Rodriguez et al., 2018), we tested binding to DD after dephosphorylation ([D]RZZ and [D]Spindly[F], Fig. 2 F), or after additional incubation with a mix of ATP and mitotic kinases, including CDK1/Cyclin B, MPS1, and Aurora B ([P]RZZ and [P]Spindly[F], Fig. 2 G and Fig. S4 A). In either case, no interaction of the RZZS[F] complex with DD was detected.

To corroborate these results, we met the significant technical challenge of producing recombinant human Dynactin ([R]Dynactin) in the HEK293[expi] expression system, using a modified version of the pBiG2 plasmid from the biGBac system (Weissmann et al., 2016; Fig. S4 B). The resulting pBiG2 was used to transfect insect cells to produce a baculovirus and then infect Expi293F cells (see Materials and methods). The purified [R]Dynactin is biochemically pure and similar to [PB]Dynactin, except for the absence of the p135 isoform of p150[glued], which is not expressed in our system (a band in the same position is likely caused by degradation of the p150[glued] subunits and is marked with an asterisk in Fig. S4 C). All Dynactin subunits were identified by MS (Table S3). SEC combined with multiangle light-scattering (SEC-MALS) measurements demonstrated that [R]Dynactin has the expected molecular mass (Fig. S4 D). [R]Dynactin appeared morphologically indistinguishable from [PB]Dynactin (Urnavicius et al., 2015), as judged by 2D classes from negative stain electron microscopy (Fig. S4 E). Analytical SEC experiments with [R]Dynactin showed that BicD2, but not Spindly, can interact directly with DD (Fig. S4, F–I). Thus, the RZZ is not a sufficiently robust trigger to relieve Spindly auto-inhibition and DD binding. Results presented below suggest that the kinetochore itself may play a role in the activation of Spindly required for DD binding.

## Relieving Spindly auto-inhibition

We next attempted to investigate how Spindly's autoinhibited state is maintained. The PE subcomplex of Dynactin interacts with the conserved Spindly motif and binds adaptors with limited but measurable binding affinity also without Dynein (Gama et al., 2017; Lau et al., 2021; Yeh et al., 2012). As the Spindly motif is predicted to reside within the autoinhibited portion of Spindly, we hypothesized that the PE-Spindly motif interaction could be used as a proxy to monitor Spindly autoinhibition, bypassing the need to form the entire DD–Spindly complex.

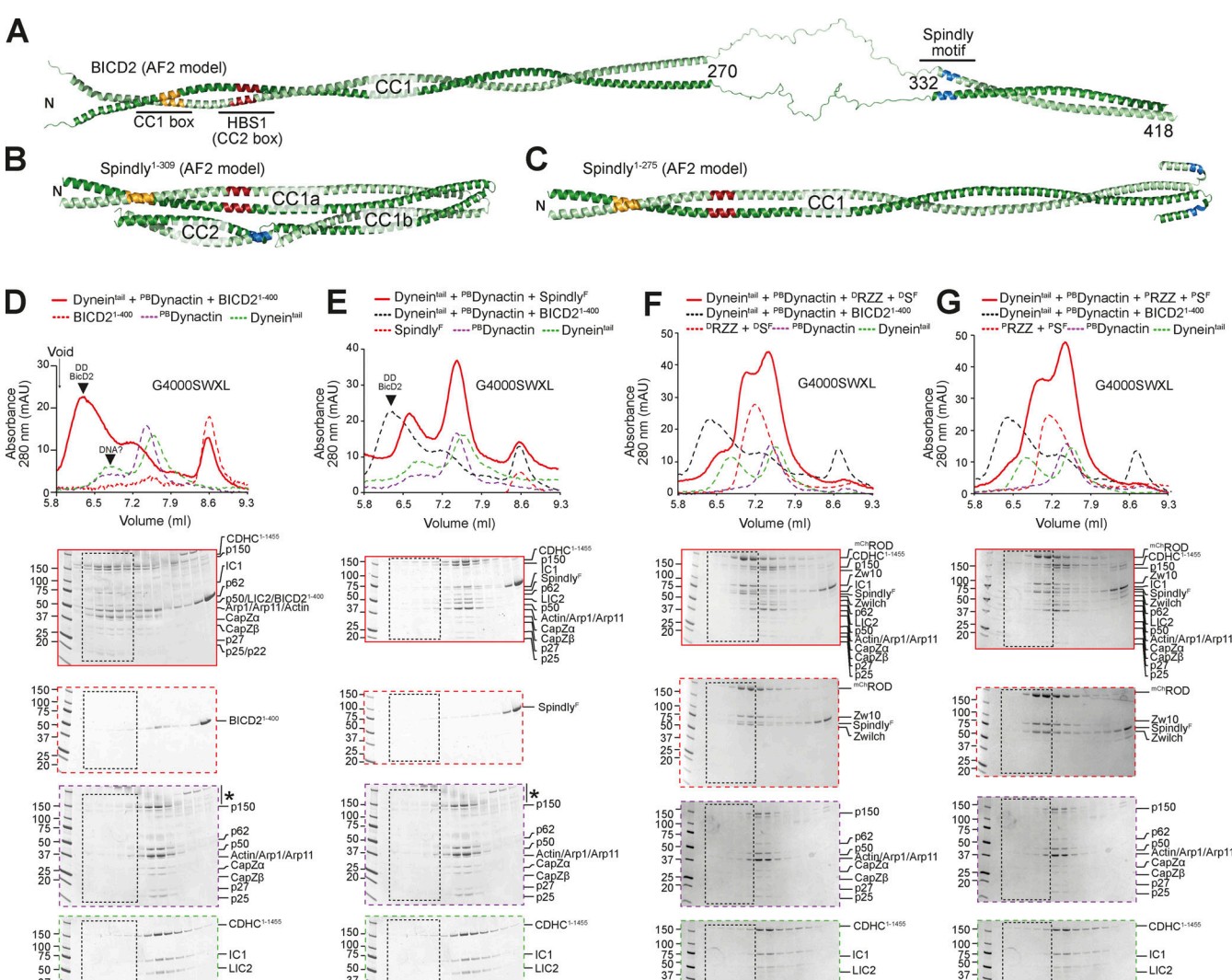

Figure 2. **Spindly autoinhibition prevents its interaction with DD. (A)** AF2 Multimer was used to predict a model of BicD2. The flexible region between 270 and 332 has very low reliability and has therefore been artificially linearized for visualization purposes (see Materials and methods). The tail region has been omitted due to limited reliability of the predictions. **(B)** AF2 Multimer model of Spindly[1–309]. **(C)** AF2 Multimer model of Spindly[1–275]. PAE plots and pLDDT scores for B and C are displayed in Fig. S2. **(D)** Analytical SEC elution profile from a G4000WXL column and SDS-PAGE to compare complex formation between BicD2 (red, dashed), [PB]Dynactin (purple, dashed), and Dynein tail (green, dashed). Experiments assessing complex formation are shown in continuous red line. Every second 100 µl elution fraction within the indicated volume range was loaded for SDS-PAGE analysis. **(E–G)** Analytical SEC elution profile and SDS-PAGE of complex formation between an adaptor–cargo/adaptor complex (red, dashed), [PB]Dynactin (purple, dashed), Dynein tail (green, dashed), and the complex run shown in red. Overlaid (black, dashed) the Dynein tail, Dynactin, BicD2[1–400] complex run of D. **(E)** Farnesylated Spindly[FL]. **(F)** Full-length Spindly[F] and RZZ treated with λ-phosphatase. **(G)** Full-length Spindly[F] and RZZ pretreated with a mix of mitotic kinases (MPS1, Aurora B, CDK1/Cyclin B). Note that the Dynein tail and Dynactin controls are both shared between D and E, and F and G. The vertical line with an asterisk in D and E marks the accumulation of unknown contaminants in the upper part of the gel. In all SEC experiments in this figure, Spindly was full length and farnesylated. Dynein tail: 1 nM; Dynactin: 1.5 µM; Spindly: 8 µM; RZZ: 2 µM. mAU, milli absorbance units. Molecular weights are in kD. Source data are available for this figure: SourceData F2.

Thus, we developed a minimal recombinant adaptor-binding subcomplex of Dynactin containing only the subunits of the PE-capping complex, p25, p27, p62, and Arp11 (Fig. 3 A). After purification to homogeneity, we tested whether this PE complex interacted with different fragments of Spindly. In SEC experiments, the PE subcomplex did not bind [mCh]Spindly[FL], in agreement with the possibility that Spindly is auto-inhibited (Fig. 3 B). Conversely, Spindly[1–275], a fragment that contains the Spindly box and that our AF2 predictions identified as having an open, elongated coiled-coil (Fig. 2 C), bound to the PE complex, albeit weakly, as indicated by a partial shift in its elution volume

(Fig. 3 C). The interaction of Spindly[1–275] with the PE complex required the Spindly motif, as Spindly[1–250], a construct lacking it, was unable to bind the PE (Fig. 3 D). C-terminal deletions have been shown to relieve autoinhibition in BICD2, mimicking the effect of cargo binding (McKenney et al., 2014; Schlager et al., 2014b). Thus, we tested various Spindly C-terminal deletions for their ability to interact with the PE complex in the absence of cargo and other activators. Constructs lacking only the (disordered) C-terminal tail (Spindly[1–440]) or lacking the C-terminal tail and the CC3 (Spindly[1–354]) did not bind the PE complex in SEC experiments (Fig. S5 A), most likely because they adopt a

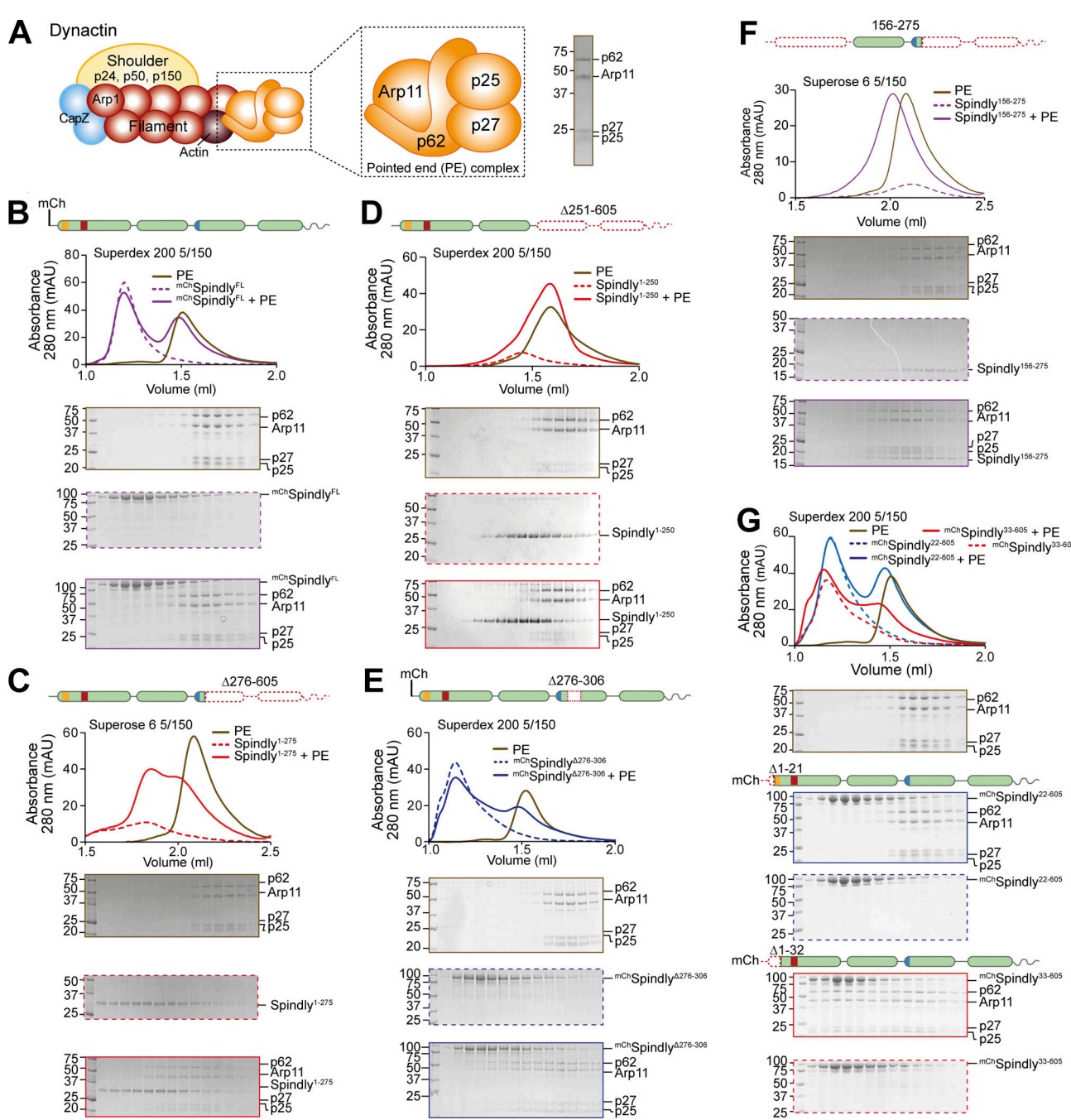

**Figure 3. Spindly autoinhibition is relieved by N- and C-terminal deletions. (A)** Schematic representation of the PE complex in the context of Dynactin, and SDS-PAGE of its chromatographic peak in gel filtration. **(B–G)** Analytical SEC binding assays between the Dynactin PE (brown) and Spindly constructs. The complex run is always represented with a continuous line, the Spindly construct with a dashed line. **(B)** ^mCh^Spindly (purple). **(C)** Spindly^1–275^ (red). **(D)** Spindly^1–250^ (red). **(E)** ^mCh^Spindly^Δ276–306^ (blue). **(F)** Spindly^156–275^ (purple). **(G)** ^mCh^Spindly^22–605^ (blue); ^mCh^Spindly^33–605^ (red). **(B–E and G)** PE: 3 µM, Spindly constructs: 8 µM. **(F)** PE: 4 µM, Spindly^156–275^: 10 µM. The control gels with the PE alone are shared between B, G, C, F, and D and Fig. S3 A. mAU, milli absorbance units. Molecular weights are in kD. Source data are available for this figure: SourceData F3.

closed conformation related to that predicted by AF2 for the Spindly^1–309^ construct (Fig. 2 B). We reasoned therefore that a Spindly deletion mutant lacking determinants of autoinhibition in the CC2 region ought to show features of the open complex and bind the PE complex even in the context of full-length Spindly. Indeed, a ^mCh^Spindly mutant lacking residues 276–306 (^mCh^Spindly^Δ276–306^), a segment already identified for

its interactions with the CC1 box, also showed affinity for the PE complex (Fig. 3 E). Neither ^mCh^Spindly^Δ276–306^ nor Spindly^1–275^ fully co-eluted with the PE complex, however, possibly indicative of low binding affinity. These constructs may be only partially open, or their binding site for the PE may be partly disrupted. Both constructs populate a dimer-tetramer equilibrium, and the tetramer, especially for Spindly^1–275^, prevails at

Figure 4. **Spindly autoinhibition involves a direct interaction between N- and C-terminal regions. (A)** Analytical SEC elution profile and SDS-PAGE analysis for interaction assays between the Spindly N-terminal and C-terminal domains. Spindly[1–250] (red, dashed) interacts with Spindly[250–605] (alone: purple, dashed; complex: red, continuous), but not with Spindly[250–275_307–605] (alone: green, dashed; complex: green, continuous). Spindly[51–250] (blue, dashed) does not interact with Spindly[250–605] (complex: blue, continuous). Concentration of all fragments: 10 μM. **(B)** schematic representation of Spindly constructs referred to in C–E. **(C)** Mass photometry results for the constructs in B. **(D)** Data table from hydrodynamic and mass photometry results in C and D. The Stokes' radius and frictional ratio were estimated from the AUC-measured sedimentation coefficient and from the theoretical molecular weights (MW). **(E)** AUC results for the constructs in B. The smaller sedimentation coefficient indicates higher drag, which is caused by an increased Stokes' radius. Molecular weights are in kD. Source data are available for this figure: SourceData F4.

the micromolar concentration used in these experiments, possibly partly counteracting the interaction with the PE expected of these constructs (unpublished observations and Fig. 4, C and D, discussed below).

We suspected that the N-terminal region of Spindly, which our experiments have suggested to be split in CC1a and CC1b segments, contributes to stabilize the autoinhibitory interaction that controls access of the PE complex to the Spindly motif. To test this, we asked if the mChSpindly[76–605] construct, lacking the segment of CC1a where both the CC1 box and the HBS1 are located, bound the PE complex. Indeed, mChSpindly[76–605] bound the Dynactin PE in SEC assays (Fig. S5 B), confirming that the N-terminal segment of Spindly contributes to maintain the autoinhibited state of Spindly. Finally, a mutant lacking the entire CC1a region and also truncated in the CC2 after the Spindly motif, Spindly[156–275], interacted robustly with the PE complex (Fig. 3 F).

To identify the role of the CC1 box in this process, we designed two sequential N-terminal truncations. mChSpindly[22–605], which retains the CC1 box, did not bind the PE complex. Conversely, mChSpindly[33–605], which does not retain the CC1 box, bound the PE complex (Fig. 3 G). Two further short deletion mutants within the CC1, mChSpindly[Δ26–28] and mChSpindly[Δ26–32], respectively, did not bind and bound weakly to the PE complex (Fig. S5, C and D), indicating that shorter deletions elicit a less penetrant effect on Spindly auto-inhibition. Collectively, these observations indicate that the Spindly auto-inhibition

mechanism involves a tight intramolecular interaction of the conserved Dynein-binding CC1 box with a regulatory segment of the Spindly CC2 coiled-coil roughly comprised between residues 276 and 306. Our results also imply that this inhibitory control cannot be readily relieved by DD, not even in the presence of cargo (RZZ), possibly implying that an additional trigger at the kinetochore catalyzes opening.

## Testing structural predictions

The interactions between CC1 and CC2 in the AF2 model of Spindly in Fig. 1 I, together with our extensive analysis in Fig. 3, explain why Spindly[1–250] (CC1) and Spindly[250–605] (CC2 and CC3) interact with high affinity and coelute from an SEC column (Sacristan et al., 2018), a result that we could readily reproduce (Fig. 4 A). When tested in the same assay, however, Spindly[51–250], lacking the N-terminal segment of CC1a predicted to bind CC2, did not interact with Spindly[250–605]. Similarly, Spindly[1–250] was unable to bind Spindly[250–275_307–605], where the CC2 segment predicted to face the CC1 box is deleted (Fig. 4 A). This is further supported by surface plasmon resonance experiments that showed the region between residues 259 and 306 is essential for interaction with Spindly[1–250] (Sacristan et al., 2018).

Disruption of interactions responsible for intramolecular folding may be expected to render the Spindly deletion mutants more elongated. To test this prediction, we verified by mass photometry that mChSpindly[76–605] and mChSpindly[Δ276–306]

remained dimeric like wild-type $^{mCh}$Spindly ($^{mCh}$Spindly$^{\Delta276-306}$ was predominantly dimeric but with a tendency to form tetramers already at very low concentration, Fig. 4, C and D). We then analyzed the sedimentation behavior of these constructs by analytical ultracentrifugation (AUC) and derived their Stokes' radii and frictional ratios (see Materials and methods; Fig. 4, D and E). For a given molecular mass, the product of the sedimentation coefficient and of the Stokes' radius is a constant (Siegel and Monty, 1966). Indeed, the sedimentation coefficient of the deletion mutants was reduced in comparison with that of $^{mCh}$Spindly$^{FL}$, indicative of larger Stokes' radii and frictional ratios and therefore of a more elongated conformation (Fig. 4, D and E). Thus, deletion of regions in CC1a and CC2 predicted to interact with each other in the folded conformation of Spindly cause an at least partial opening of the Spindly structure.

**Opening up Spindly with point mutations**
Because the region of Spindly downstream of the Spindly box (residues 276–305) is crucial for autoinhibition, we tried to target the autoinhibitory mechanism with individual point mutations in this segment. Downstream of the Spindly box, sequences of Spindly orthologues diverge from BICD family adaptors (Fig. S1 L). Within this region, we mutated Spindly's positively charged residues R295 and K297 to glutamate. The resulting construct, indicated as Spindly$^{CC2*}$ (where CC2* indicates the R295E-K297E mutant in the CC2 coiled-coil), bound the PE complex in SEC (Fig. 5 A). The interaction with the PE complex was mediated by the Spindly motif, because combining the CC2* mutation with a mutation in the Spindly motif (F258A, indicated as SM*) to generate the $^{mCh}$Spindly$^{SM*-CC2*}$ mutant, abolished the interaction (Fig. 5 B). In an orthogonal approach, we developed $^{mCh}$Spindly$^{\Delta RV}$, a construct deleted of the Spindly-specific two-residue insert (residues R154–V155) between the CC1a and CC1b coiled-coil segments (Fig. S1 F) with the goal of favoring a full extension of CC1 like in BICD2, which lacks the insertion. $^{mCh}$Spindly$^{\Delta RV}$ bound the PE complex (Fig. 5 C), albeit with reduced affinity, suggesting that the two-residue insertion into the CC1 of Spindly favors autoinhibition.

In mass photometry measurements, $^{mCh}$Spindly$^{CC2*}$ and $^{mCh}$Spindly$^{\Delta RV}$ had masses expected of dimers (Fig. 5 D) and essentially indistinguishable from those of $^{mCh}$Spindly$^{FL}$ (Fig. 1 C), suggesting that their ability to interact with the PE complex does not result from changes in stoichiometry. This was further confirmed by fusing Spindly$^{FL}$ and Spindly$^{CC2*}$ to GST to reinforce their dimerization. GST-Spindly$^{FL}$ did not bind the PE complex, whereas GST-Spindly$^{CC2*}$ did (Fig. S6, A and B). AUC demonstrated a decreased sedimentation coefficient for $^{mCh}$Spindly$^{CC2*}$ and $^{mCh}$Spindly$^{\Delta RV}$ (Fig. 5 E), indicative of a more extended conformation, as already shown for the Spindly deletion mutants in Fig. 4, D and E. Further, we developed an mCh-tagged BICD2-Spindly chimeric construct that combined the BICD2 N-terminal region until the Spindly motif (residues 1–292$^{BICD2}$), which is believed to contain an uninterrupted CC1, with the Spindly motif and C-terminal RZZ-binding domain of Spindly (residues 251–605$^{Spindly}$; Fig. 5 F). Indeed, AF2 modelled this construct (Spindly$^{Chimera}$) with a continuous CC1 until the flexible region that precedes the Spindly box (Fig. S6 C).

Spindly$^{Chimera}$ promoted Spindly-dependent oligomerization of the RZZ complex in filaments in vitro (Fig. 5 G), which requires MPS1 kinase and mimics kinetochore corona assembly (Raisch et al., 2022; Rodriguez-Rodriguez et al., 2018; Sacristan et al., 2018). Thus, the Spindly segment in Spindly$^{Chimera}$ is sufficient for polymerization. Spindly$^{Chimera}$ was even able to trigger formation of filaments at room temperature in the absence of MPS1, a condition where Spindly$^{WT}$ did not stimulate filament formation (Fig. S6 D). This behavior implies loss of auto-inhibition and is reminiscent of Spindly's deleted of the N-terminal region, which polymerizes in vivo in interphase cells without a requirement for MPS1 phosphorylation (Sacristan et al., 2018). In SEC experiments, $^{mCh}$Spindly$^{Chimera}$ bound the PE complex (Fig. 5 H), in agreement with our expectation that an uninterrupted CC1 allows the PE to access the Spindly motif.

Finally, we asked whether relief of autoinhibition would increase the affinity of Spindly for Dynein in addition to increasing the affinity for the PE complex of Dynactin. As the CC1 box, required for the interaction with the Dynein LIC, is directly involved in the autoinhibitory interaction, we asked whether we could see increasing binding of LIC upon straightening Spindly to render the CC1 box more accessible. As the affinity of the LIC for adaptors containing the CC1 box is too low for accurate study by SEC, we used a pull-down assay. We produced recombinantly a GST-tagged construct of the LIC2 isoform, and tested its ability to pull-down Spindly wild-type and "open" mutants. As a negative control, we used the $^{mCh}$Spindly$^{AA/VV}$ (A23V-A24V) mutant, which has been previously shown to inhibit the interaction with the LIC1 isoform in a similar assay (Gama et al., 2017). As open mutants, we used $^{mCh}$Spindly$^{CC2*}$ and $^{mCh}$Spindly$^{\Delta RV}$. LIC2 pulled down $^{mCh}$Spindly$^{WT}$, matching previously published observations (Gama et al., 2017), but the open mutants showed increased affinity for the LIC, with $^{mCh}$Spindly$^{CC2*}$ showing an even slightly higher affinity than $^{mCh}$Spindly$^{\Delta RV}$, possibly due to an only partial restoration of coiled-coil continuity in the latter (Fig. 5, I and J), and in line with the lower apparent affinity of $^{mCh}$Spindly$^{\Delta RV}$ for the PE. Collectively, these results indicate that open mutants of Spindly interfering with the stability of the CC1–CC2 interaction or with the bending of the CC1 coiled-coil are more easily accessible to DD.

**Spindly$^{CC2*}$ binds DD with higher affinity than Spindly$^{WT}$**
After showing that Spindly$^{WT}$ does not interact with DD, we were eager to test whether the open Spindly mutants formed a super-complex with DD. As we have shown that fragments from both Dynein (the LIC) and Dynactin (the PE) bind the CC2* mutant independently, we performed binding assays using stoichiometric ratios of Spindly, Dynein$^{tail}$, and $^R$Dynactin to maximize complex formation (Fig. 6 A). $^{mCh}$Spindly$^{WT}$ was unable to form the super-complex with DD, as expected, but $^{mCh}$Spindly$^{CC2*}$ and $^{mCh}$Spindly$^{Chimera}$ were, as assessed by the shift of the adaptor into the expected super-complex peak (Fig. 6, A and B). We also found that $^{mCh}$Spindly$^{33-605}$ interacted with DD, albeit apparently with less affinity than $^{mCh}$Spindly$^{CC2*}$ or $^{mCh}$Spindly$^{Chimera}$. Both $^{mCh}$Spindly$^{CC2*}$ and $^{mCh}$Spindly$^{Chimera}$ interacted with the RZZ complex, indicating that the mutations do not affect the cargo-binding region of RZZ (Fig. 6, C and D).

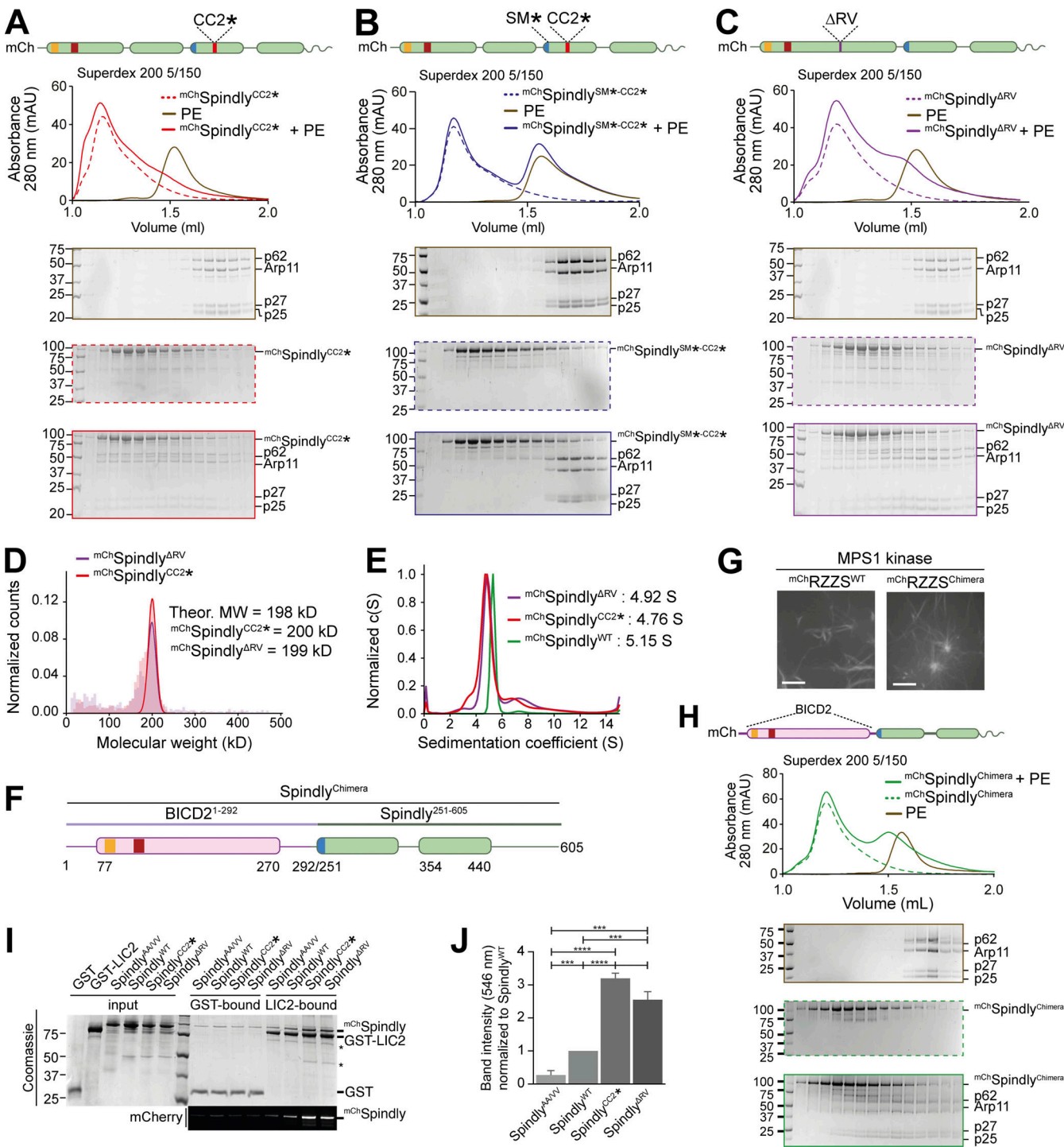

**Figure 5. Point mutations relieve Spindly autoinhibition. (A–C and H)** Analytical SEC analyses on a Superdex 200 5/150 column to assess complex formation between the Dynactin PE (brown) and various Spindly constructs. The complex run is always represented with a continuous line, the Spindly construct with a dashed line. **(A)** mChSpindly^CC2* (red). **(B)** mChSpindly^SM*-CC2* (blue). **(C)** mChSpindly^ΔRV (purple). **(H)** mChSpindly^Chimera (green). PE: 3 µM; Spindly construct: 8 µM. **(D)** Mass photometry results for mChSpindly^CC2* (red) and mChSpindly^ΔRV (purple). The main peaks' "shoulders" are consistent with minor sample degradation. **(E)** AUC profile of mChSpindly^CC2* (red), mChSpindly^ΔRV (purple), and mChSpindly^WT (green). c(S), sedimentation coefficient. **(F)** schematic representation of the mChSpindly^Chimera. **(G)** Spinning-disk confocal fluorescence microscopy-based filamentation assay at 561 nm shows the indicated mChRZZS^F species (4 µM RZZ, 8 µM farnesylated Spindly) form filaments when incubated at 20°C with MPS1 kinase. Scale bar: 5 µm. **(I)** SDS-PAGE analysis of pulldown assay with either GST or GST-tagged LIC2 as bait, and mCh-tagged Spindly as prey. Coomassie staining and fluorescent signal in the red channel are displayed. Asterisks mark contaminants or degradation products. **(J)** Quantification of the mChSpindly fluorescent signal and SDs calculated from three technical replicates. Statistical analysis was performed with a parametric test comparing two unpaired groups. ***, P ≤ 0.001; ****, P ≤ 0.0001. The PE alone controls in A and C are shared with the control in Fig. 3 E. mAU, milli absorbance units. Molecular weights are in kD. Source data are available for this figure: SourceData F5.

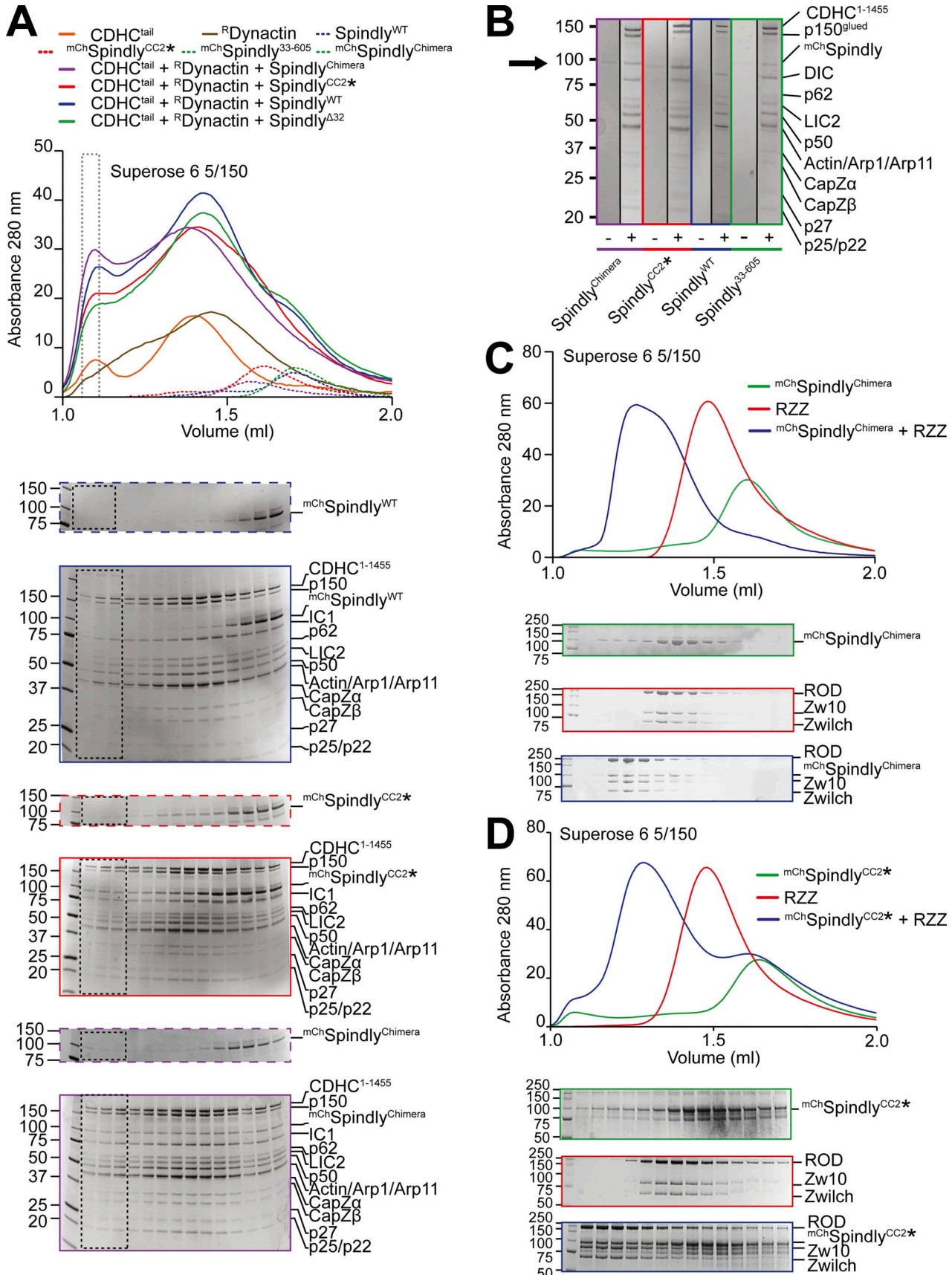

Figure 6. **Complex formation assay between DD and Spindly mutants. (A)** Elution profiles and SDS-PAGE of complex formation assays between Dynein tail, recombinant Dynactin, and Spindly constructs. Experiment run on a Superose 6 5/150 column, in stoichiometric conditions. Only selected gels are

displayed. The gray-dotted box indicates the fraction loaded in the SDS-PAGE shown in B. Dynein tail: 0.75 µM, Dynactin: 0.75 µM, Spindly: 2 µM. **(B)** Comparison of the fractions of the expected DDS complex peak shown in A. A minus sign indicates adaptor-only runs, a plus indicates full complex runs. The arrow points at the expected position of <sup>mCh</sup>Spindly. **(C and D)** Analytical SEC experiments on a Superose 6 5/150 column to assess complex formation (blue) between the RZZ complex (red) and the indicated Spindly constructs (green). (C) <sup>mCh</sup>Spindly<sup>Chimera</sup>. (D) <sup>mCh</sup>Spindly<sup>CC2*</sup>. RZZ: 2 µM; Spindly constructs: 6 µM. Both Spindly constructs were pre-farnesylated. Molecular weights are in kD. Source data are available for this figure: SourceData F6.

### The localization of Spindly to human kinetochores

Finally, we assessed the ability of the different Spindly mutants to reach kinetochores in human cells arrested in mitosis with a spindle poison. Endogenous Spindly was depleted by RNAi and recombinant, purified <sup>mCh</sup>Spindly protein variants were introduced in cells through electroporation as summarized in Fig. S7, A–C. As shown previously (Gassmann et al., 2010), depletion of Spindly prevented kinetochore recruitment of Dynactin (Fig. 7, A–D). Electroporation of <sup>mCh</sup>Spindly<sup>WT</sup> largely rescued these effects. Despite being present in cells at levels lower than those of <sup>mCh</sup>Spindly<sup>WT</sup>, <sup>mCh</sup>Spindly<sup>33–605</sup> decorated kinetochores indistinguishably. It also promoted recruitment of comparable levels of Dynactin (Fig. 7, A–D, and Fig. S7, D–F). This result was unexpected, because the CC1 box has been previously shown to be required for kinetochore localization of Dynactin (Sacristan et al., 2018). By suggesting that binding of the CC1 box to LIC1 is not required for robust DD recruitment at human kinetochores, this result may seem at odd with the observation that the Spindly<sup>A23V</sup> mutant (where the CC1 box is mutated rather than absent) strongly impairs kinetochore recruitment of Dynactin (Sacristan et al., 2018). Our <sup>mCh</sup>Spindly<sup>AA/VV</sup> mutant could not be used to further investigate the issue, as—for unclear reasons—it was unable to reach kinetochores (unpublished results), preventing us from comparing it to <sup>mCh</sup>Spindly<sup>33–605</sup> in the same assay. Nonetheless, the new results with <sup>mCh</sup>Spindly<sup>33–605</sup> suggest that the deletion of the CC1 box or its mutation result in fundamentally distinct behaviors, and support a role of the LIC subunits as triggers of adaptor opening more than as decisive contributors to the binding affinity of the interaction, a speculative conclusion that will require further investigation.

Both <sup>mCh</sup>Spindly<sup>CC2*</sup> and <sup>mCh</sup>Spindly<sup>Chimera</sup>, on the other hand, showed strongly reduced kinetochore levels (Fig. 7, A–D). The cellular levels of <sup>mCh</sup>Spindly<sup>CC2*</sup> were lower than those of the other constructs (Fig. S7 C), and we cannot exclude that reduced kinetochore levels reflect this protein's lower cellular levels. The general levels of <sup>mCh</sup>Spindly<sup>Chimera</sup>, on the other hand, were comparable to those of <sup>mCh</sup>Spindly<sup>33–605</sup>, supporting the significance of its reduced kinetochore localization (Fig. S7 C). The latter correlated with reduced levels of Dynactin, as measured with an antibody against the p150<sup>glued</sup> subunit. The Spindly and Dynactin signals at kinetochores were well correlated for all four constructs (Fig. 7 E and Fig. S7, D–H). This argues that the reduction in kinetochore levels of Dynactin arises due to a reduction in the kinetochore levels of the <sup>mCh</sup>Spindly<sup>CC2*</sup> and <sup>mCh</sup>Spindly<sup>Chimera</sup> constructs, rather than to an inability of Dynactin to interact with them. This conclusion is also in line with our biochemical data showing that these Spindly mutants interact with DD. Because Spindly<sup>Chimera</sup> promoted RZZ filamentation with enhanced efficiency in vitro (Fig. S6 D), we reasoned that its reduction at kinetochores may reflect

the assembly of ectopic complexes with DD no longer limited to kinetochores, thus reducing the pool available for kinetochore binding. Indeed, <sup>mCh</sup>Spindly<sup>Chimera</sup> formed ectopic corona-like filaments with Dynactin in interphase cells in the absence of kinetochores (Fig. 7 F).

Kinetochore localization by Spindly<sup>CC2*</sup> was evidently severely impaired but this construct did not form ectopic corona-like filaments (unpublished results). Filamentation assays with RZZ showed it was almost unable to promote MPS1-dependent filament assembly (Fig. 7 G). Thus, it appears that the CC2* mutations (R295E-K297E) affect at the same time Spindly's filament formation and kinetochore recruitment, without impairing DD binding. CC2* shares these characteristics with Spindly<sup>Δ276–306</sup> (Fig. 3 E and Raisch et al., 2022). Collectively, these observations suggest that the region of CC2 containing these residues is at the same time implicated in kinetochore binding and filament formation, suggesting a possible mechanism of kinetochore regulation of corona expansion.

## Discussion

Spindly has been mainly studied as a mitotic adaptor of DD, but recent observations suggest roles also in interphase cells (Clemente et al., 2018; Conte et al., 2018; Del Castillo et al., 2020). Here, we have dissected a mechanism of conformational control that addresses the functions of Spindly at kinetochores, and that might inform future studies of Spindly in other cellular locales and cell cycle phases. More generally, our studies have implications for the control of DD in time and space. Adaptors are known to activate the motility and processivity of Dynein by stabilizing a complex with Dynactin (Hoogenraad and Akhmanova, 2016; McKenney et al., 2014; Schlager et al., 2014b). Studies so far have focused on N-terminal segments of adaptors that overcome intramolecular regulation, exemplified by the BICD2<sup>1–400</sup> construct. Previous studies had demonstrated that the C-terminal cargo-binding region decreases the affinity of BICD2 for DD (Hoogenraad et al., 2003; Liu et al., 2013; Splinter et al., 2012; Terawaki et al., 2015). This effect was clearly evident also with Spindly, whose full-length form failed to bind DD in a variety of assays. However, Spindly undergoes an apparently more complex regulation, that is not limited to cargo binding, but also includes a folded-back conformation of the N-terminal coiled-coil.

The RZZ complex, considered the Spindly cargo at kinetochores, binds directly to farnesylated Spindly (Mosalaganti et al., 2017; Raisch et al., 2022). The observation that binding of RZZ to Spindly was insufficient to unleash a conformational change compatible with DD binding motivated our detailed investigation of the determinants of Spindly's closed conformation. A Spindly fragment encompassing residues 354–605 is sufficient

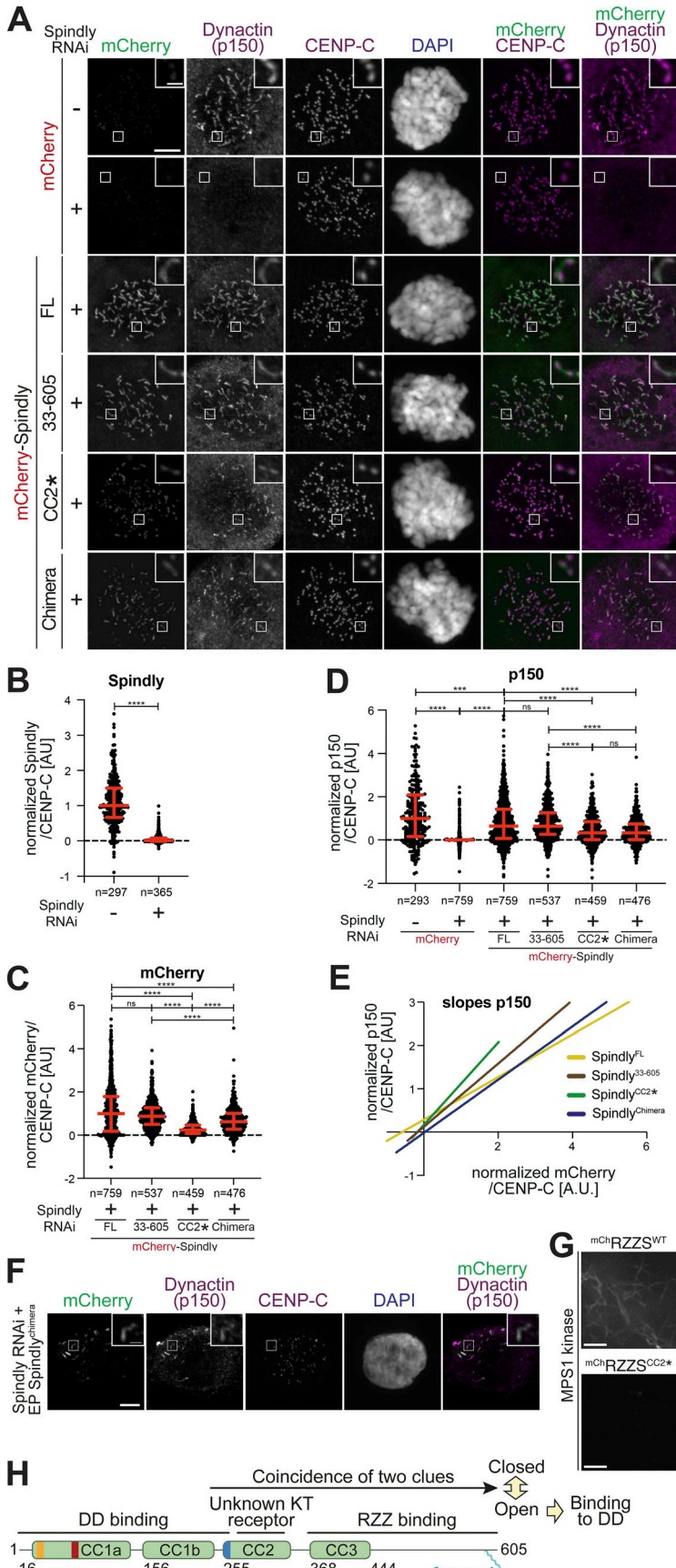

**Figure 7. Kinetochore levels of Dynactin in presence of Spindly mutants. (A)** Representative images showing the effects of a knockdown of the endogenous Spindly in HeLa cells on Dynactin recruitment monitored through the p150$^{glued}$ subunit. RNAi treatment was performed for 48 h with 50 nM siRNA (see Fig. S7 A). Before fixation, cells were synchronized in G2 phase with 9 µM RO3306 for 16 h and then released into mitosis. Subsequently, cells were immediately treated with 3.3 µM nocodazole for an additional hour. CENP-C was used to visualize kinetochores and DAPI to stain DNA. Scale bar here and in F: 5 µm (whole cell) or 1 µm (inset). **(B)** Quantification of residual Spindly levels at kinetochores. A representative image is shown in Fig. S7 B. *n* refers to individual measured kinetochores. Statistical analysis (also for C and D) was performed with a nonparametric *t* test comparing two unpaired groups (Mann–Whitney test). Symbols indicate: n.s., P > 0.05; ***, P ≤ 0.001; ****, P ≤ 0.0001. Red lines, here and in C and D, indicate mean and SD. Three biological replicates were performed for experiments in B–D. **(C)** Quantification of kinetochore levels of the indicated electroporated $^{mCh}$Spindly proteins. *n* refers to individual measured kinetochores. **(D)** Kinetochore levels of Dynactin in cells depleted of endogenous Spindly and electroporated with the indicated Spindly proteins. *n* refers to individual measured kinetochores. **(E)** Least square linear fitting through the distribution of data points reporting for each kinetochore the CENP-C–normalized $^{mCh}$Spindly intensity on the x-axis and the CENP-C–normalized p150$^{glued}$ intensity on the y-axis. The individual distributions are shown in Fig. S7, C–G. **(F)** Electroporated $^{mCh}$Spindly$^{Chimera}$ is observed forming polymers in Spindly-depleted cells in interphase, causing ectopic recruitment of p150$^{glued}$. **(G)** Spinning-disk confocal fluorescence microscopy-based filamentation assay at 561 nm with the indicated $^{mCh}$RZZS$^F$ species (4 µM RZZ, 8 µM Spindly$^F$) at 20°C in presence of MPS1 kinase. Scale bar: 5 µm. **(H)** Model for the activation of Spindly.

to bind RZZ (Henen et al., 2021; Raisch et al., 2022), but the autoinhibited conformation requires a fragment of Spindly comprised between residues 276 and 309, and therefore positioned upstream of the minimal RZZ-binding region (Fig. 7 H). AF2 modeling (Evans et al., 2021 *Preprint*; Jumper et al., 2021) suggests that Spindly[1–275] adopts an open conformation that is closely reminiscent of that of BICD2[1–400], whereas Spindly[1–309] may adopt a closed conformation. While these predictions must be taken with caution in the absence of supporting experimental evidence, they are entirely consistent with our detailed biochemical and biophysical analysis. Specifically, this model is supported by crosslinking analyses that predict the Spindly CC1 box to be in close proximity with residues 295–305, and by evidence that the closed conformation correlates with a break in the CC1 coiled-coil that allows Spindly to bend back on itself around residue 155. This model also explains a pattern of intramolecular contacts revealed by XL-MS that would otherwise not be expected for a parallel coiled-coil like the one in Spindly. Our mutational analysis, combined with a binding assay monitoring the interaction with the PE complex of Dynactin, and thus measuring the accessibility of the Spindly motif, was entirely consistent with the hypothesis that these two structural determinants, namely, the two-residue insertion in CC1 and the 295–305 region, are crucial for the auto-inhibitory mechanism of Spindly. This model was later confirmed by demonstrating binding of Spindly mutants to the entire DD complex. For these experiments, we established the expression of a recombinant form of Dynactin, a development that will enable production of significantly more homogeneous samples and the design of mutants, as well as enabling the use of tags for the individual subunits.

If RZZ binding is insufficient for unleashing the potential of Spindly to bind DD, what else might be required? Previous work demonstrated that the RZZ complex is necessary for the recruitment of Spindly to the kinetochore and that Spindly, in turn, recruits DD (Barisic et al., 2010; Chan et al., 2009; Cheerambathur et al., 2013; Gassmann et al., 2008; Gassmann et al., 2010; Griffis et al., 2007; Raaijmakers et al., 2013; Starr et al., 1998; Yamamoto et al., 2008). Because the RZZ complex does not appear to be sufficient to promote binding of Spindly to DD, our analysis implies that a second, unknown interaction in the kinetochore elicits the opening of Spindly. The identity of the binding partner is unknown and will represent the focus of our future investigations. A possible working model is that the still unknown kinetochore binder interacts with the region comprised between residues 276 and 306, relieving it from its intra-molecular control of the Spindly closed conformation, and promoting the transition to the open conformation. Evidence supporting this idea is that mutations in the 276–306 region, including the deletion of this entire fragment or the introduction of charge-inverting point mutations at residues 295 and 297, respectively, abolish or largely decrease kinetochore recruitment of Spindly (this study and Raisch et al., 2022; Sacristan et al., 2018), implying that they impinge directly not only on the closed-open transition but also on the Spindly recruitment mechanism. In vitro, the 276–306 region is also required for the assembly of RZZS filaments (this study and Raisch et al., 2022). Furthermore, residues 274–287 of Spindly are necessary for

RZZS filament formation in cells (Sacristan et al., 2018). However, this is unlikely to explain the kinetochore localization defect, because corona expansion is not required for robust recruitment of Spindly to the kinetochore (Raisch et al., 2022; Rodriguez-Rodriguez et al., 2018).

Our recent report of the cryo-EM structure of the RZZ complex, together with the investigation of the mechanism of corona assembly (Raisch et al., 2022), has considerably advanced our mechanistic understanding of corona assembly. In agreement with previous studies (Barbosa et al., 2020; Rodriguez-Rodriguez et al., 2018), we found important roles for protein phosphorylation in corona assembly. Nonetheless, how Spindly promotes corona assembly and the detailed organization of the corona remain poorly understood. The RZZ is evolutionarily and structurally related to precursors of the coats that surround membrane vesicles during intra-cellular transport (Civril et al., 2010; Mosalaganti et al., 2017) and is thus likely to polymerize through similar mechanisms. Plausibly, the solution to this conundrum will require biochemical reconstitutions addressing the spectrum of interactions that both RZZ and Spindly establish at the kinetochore.

In conclusion, we have studied in mechanistic detail the basis of Spindly activation at the kinetochore and shown it to reflect a structural transition from a closed to an open conformation. We identified several structural determinants likely involved in the transition, and proposed a two-step activation mechanism comprising RZZ binding and binding to an unknown second kinetochore receptor ultimately required for binding of Spindly to DD. These studies have important general implications for the mechanism of activation of adaptors, as they imply that the definition of "cargo" must be nuanced to every particular situation in which an adaptor is involved. In our particular analysis, the RZZ complex continues to be legitimately considered an element of the adaptor's cargo, but our data imply that a second, equally important component remains to be identified. Whether a similar two-step or multistep mechanism applies to additional cargo-adaptor systems is an important question for future studies.

## Materials and methods

### Mutagenesis and cloning

cDNA segments encoding for wild-type Spindly or truncated constructs were subcloned into a pET28-mCh plasmid, with an intervening PreScission cleavage site, or into a pLib plasmid for insect cell expression, with a 5′ insert coding for a His$_6$-tag. Mutations were introduced by site-directed mutagenesis by Gibson assembly (Gibson et al., 2009). All constructs were sequence verified. The BicD2-Spindly chimeric construct (mChSpindly[Chimera]) was created by Gibson Assembly. We generated a sense primer containing the last 25 bp of the coding sequence for the BicD2 (1–292) segment, and the first 25 bp of the coding sequence for the Spindly (251–605) segment, as well as an antisense primer containing the same region. We used the former to expand the Spindly (251–605) sequence with an additional 3′ overlap with the multiple cloning site of a pLib-mCh plasmid, and the latter to expand the BicD2 (1–292) coding sequence with an additional 5′ overlap with the pLib-mCh plasmid

multiple cloning site. The two PCR products were then assembled by Gibson Assembly into a pLib-mCh plasmid opened by restriction cloning (BamHI-HindIII). A plasmid for the expression of the Dynactin PE was generated using the biGBac system (Weissmann et al., 2016). Coding sequences for Arp11, p62, p25-6His, and p27 were subcloned into individual pLib plasmids, which were then used to build a pBiG1 plasmid, which was then used for expression of the entire complex. For the GST-LIC2 construct, cDNA encoding for human LIC2 was subcloned into a pLib vector containing an N-terminal TEV-cleavable GST.

The following Dynactin coding genes were codon-optimized for protein expression in insect cells and synthesized (Epoch Life Science): DCTN1 (Protein name p150, Uniprot Isoform 1, NCBI Reference Sequence NM_004082.4), DCTN2 (p50, Isoform 1, NM_006400.4), DCTN3 (p24, Isoform 1, NM_007234.4), DCTN4 (p62, Isoform 1, NM_016221.3), DCTN5 (p25, Isoform 1, NM_032486.3), DCTN6 (p27, NM_006571.3), CAPZA1 (CapZα-1, NM_006135.2), CAPZB (CapZβ, Isoform 2, NM_004930.4), AC-TR1A (Arp1, NM_005736.3), ACTR10 (Arp11, NM_018477.2), and ACTB (β-actin, NM_001101.4). All genes were cloned into the pACEBac1 vector (Geneva Biotech) in between the polyhedrin promoter and Simian virus 40 poly A sequences by Gibson cloning. A sequence encoding the ZZ affinity tag plus a TEV-cleavage site was fused in-frame to the 5′-end of the DCTN1 gene. For cloning of mammalian cell expression vectors, the genes were cloned out of pACEBac1 into pcDNA4/TO (Thermo Fisher Scientific) in between the citomegalovirus (CMV) promoter and the bovine growth hormone poly A signal. To assemble the dynactin genes into a single expression plasmid, biGBac cloning was used and adapted for mammalian expression. The biGBac cloning plasmids (pBig1a, pBig1b, pBig1c, pBig2abc) were generated by Gibson cloning and using the pACEBac1 vector as the backbone. To amplify the gene expression cassettes by PCR and to assemble them into pBig1 plasmids, a set of CasMam oligonucleotides was designed and synthesized (Sigma-Aldrich/Merck, desalt purification). The oligonucleotides contained the optimized linker sequences (α/β/γ/δ/ε/ω) as described in Weissmann et al. (2016) and were modified to be complementary to the 5′-end of CMV promoter sequence (5′-GTTGACATTGATTATTGACTAG-3′—forward oligonucleotide) and reverse-complementary to the 3′-end of the bovine growth hormone polyA sequence (3′-CCATAGAGCCCACCGCATCC-5′—reverse oligonucleotide). The gene expression cassettes were then used to build three pBiG1 plasmids: pDCTN A, containing the ZZ-TEV-tagged p150$^{glued}$, p50, and p24; pDCTN B, containing Arp1, Arp11, β-actin, CapZα, and CapZβ; and pDCTN C, containing p25, p27, and p62 subunits. The three pBiG1 plasmids were then used to build a single pDCTN FL plasmid. Successful assembly of pDCTN FL was confirmed by complete plasmid sequencing (Center for Computational and Integrative Biology DNA Core Facility at Massachusetts General Hospital).

### Expression and purification of RZZ, mCh-Spindly, and Spindly constructs

The RZZ complex was expressed and purified using the biGBac system, with an mCh N-terminally fused tag on the ROD subunit, as previously described (Sacristan et al., 2018). Expression of all mCh-Spindly constructs and mutants except the $^{mCh}$Spindly$^{Chimera}$ was carried out in *Escherichia coli*. BL21 CodonPlus cells were transformed with the plasmid, and grown in TB at 37°C to an OD$_{600}$ of 0.5. Expression was induced with 0.4 mM IPTG. The culture was then transferred into an incubator pre-cooled to 18°C, and grown overnight before harvesting. The pellet was then snap-frozen and stored at –80°C until purification. $^{mCh}$Spindly mutants containing the unnatural amino acid Bpa were expressed in *E. coli* BL21 strains containing the pEVOL-pBpF plasmid (Chin and Schultz, 2002). Cells were cultured in selective (kanamycin, chloramphenicol) TB media, supplemented with 0.2% arabinose, to trigger expression of the tRNA synthetase/tRNA pair. Cells were grown at 37°C until an OD$_{600}$ of 0.6 was reached. Expression was induced with 0.5 mM IPTG, and the Bpa was added to the bacterial culture at a concentration of 1 mM. The culture was then transferred into an incubator pre-cooled to 18°C and grown overnight before harvesting. The pellet was then snap-frozen and stored at –80°C until purification. Spindly constructs without the mCh tag and the $^{mCh}$Spindly$^{Chimera}$ were expressed using the biGBac system as His$_6$ fusions. Baculovirus was generated in Sf9 culture and used to infect TnAO38 cells, which were grown for 72 h at 27°C before harvesting. The pellet was then snap-frozen and stored at –80°C until purification. All $^{mCh}$Spindly and Spindly constructs were purified using the same protocol. Pellets were resuspended in lysis buffer (50 mM Hepes, pH 8.0, 250 mM NaCl, 50 mM imidazole, 2 mM Tris[2-carboxyethyl]phosphine [TCEP]) supplemented with protease inhibitor cocktail and lysed by sonication. The lysate was clarified by centrifugation followed by sterile filtration and loaded onto a HisTrap HP column (Cytiva), which was then washed with at least 10 column volumes (CV) lysis buffer. Elution was performed with lysis buffer with 300 mM imidazole. The eluate was diluted 1:5 in no salt buffer (50 mM Hepes, pH 8.0, 2 mM TCEP), and applied to a 6 ml Resource Q anion exchange column (Cytiva). Elution was then performed over a 50–500 mM NaCl gradient. Fractions of the peak were analyzed by SDS-PAGE and those containing the protein of interest were pooled and concentrated. The concentrated sample was loaded onto a Superdex 200 10/300 pre-equilibrated in Spindly buffer (50 mM Hepes, pH 8.0, 250 mM NaCl, 2 mM TCEP). The eluate was concentrated to 10 mg/ml, flash-frozen, and stored at –80°C until use. The Spindly$^{250–275\_307–605}$ construct was expressed in bacteria as an mCh fusion, and the mCh tag was removed after the SEC purification step by overnight incubation with PreScission protease purified in-house. The protease and the cleaved mCh tag were then separated from the Spindly sample by a further run of SEC on a Superdex 200 10/300 column pre-equilibrated in Spindly buffer, and the eluate was concentrated, flash-frozen, and stored at –80°C until use.

### Expression and purification of Spindly$^{1–100}$

The N-terminal 1–100 residues of Spindly were cloned in the pET-NKI-His-3C-LIC vector (Luna-Vargas et al., 2011) for expression in *E. coli* BL21 (DE3) cells. Cells were grown in LB medium at 37°C to an OD$_{600}$ of 0.6. Overexpression was induced by adding IPTG to a final concentration of 0.2 mM. After induction, the cells were further grown for 18 h at 18°C. For purification,

cell pellet from a 2-liter culture was resuspended in lysis buffer (40 mM Hepes/HCl, pH 7.5, 500 mM NaCl, 10 mM Imidazole, 2 mM TCEP [Thermo Fisher Scientific]) supplemented with a protease inhibitor tablet (Roche) and 0.1 mM PMSF. Cells were lysed by sonication (10 s ON/30 s OFF; 70% Amplitude; 180 s), and the lysate was centrifuged at 53,000 $g$ for 30 min. The supernatant was filtered through a 0.45-μM filter (Millipore) and incubated with 1 ml of Ni-Sepharose beads (Qiagen) on a rotator for 1 h at 4°C. The protein was eluted from the beads with elution buffer (40 mM Hepes, pH 7.5, 100 mM NaCl, 500 mM Imidazole, 2 mM TCEP). Protein containing fractions were pooled together and incubated with 3 mg/ml of 3C protease (1:100 molar ratio) overnight at 4°C to cleave off the N-terminal 6x-His tag. The protein was further purified by ion-exchange chromatography using a 6 ml Porous XQ column and eluted with a linear NaCl gradient (50–1,000 mM). As a final purification step, the protein was loaded onto a S200 10/300 SEC column equilibrated with buffer containing 40 mM Hepes/HCl, pH 7.5, 100 mM NaCl, 2 mM TCEP. The elution fractions were analyzed by SDS-PAGE, and the protein containing fractions were pooled together, concentrated to 11 mg/ml, and stored at –80°C.

## Expression and purification of GST-LIC2
Baculoviruses for GST-LIC2 expression were generated in Sf9 culture and used to infect TnAO38 cells, which were grown for 72 h at 27°C before harvesting. The pellet was then snap-frozen and stored at –80°C until purification. The pellet was resuspended in Spindly buffer supplemented with protease inhibitor cocktail, and lysed by sonication. The lysate was clarified by centrifugation followed by sterile filtration and loaded onto a GSTrap column. The column was washed with 10 CV Spindly buffer. Elution was performed in SEC buffer supplemented with 50 mM glutathione, pre-buffered to pH 8.0. The eluate was pooled and concentrated for gel filtration. Gel filtration was performed on a Superdex 200 10/300 column pre-equilibrated in Spindly buffer. The fractions of the peak were analyzed by SDS-PAGE, and the ones containing the protein of interest were pooled, concentrated, snap-frozen, and stored at –80°C until use.

## Expression and purification of the Dynactin PE
Baculoviruses for PE complex expression were generated in Sf9 culture and used to infect TnAO38 cells, which were grown for 72 h at 27°C before harvesting. The pellet was then snap-frozen and stored at –80°C until purification. The pellet was resuspended in lysis buffer supplemented with protease inhibitor cocktail and 1 mg/ml DNAse, and lysed by sonication. The lysate was clarified by centrifugation followed by sterile filtration and loaded onto a HisTrap HP column, which was then washed with at least 10 CV lysis buffer. Elution was performed with lysis buffer with 250 mM imidazole. The eluate was diluted 1:5 in no salt buffer and applied to a 6-ml Resource Q anion exchange column. Elution was performed over a 50–500 mM NaCl gradient. Fractions of the peak were analyzed by SDS-PAGE, and those containing the entire complex were pooled and concentrated. The concentrated sample was loaded onto a Superdex 200 16/60 column pre-equilibrated in Spindly buffer. The eluate

was concentrated to 15 mg/ml, flash-frozen, and stored at –80°C until use.

## Expression and purification of recombinant Dynein tail
Expression of the Dynein tail (residues 1–1,455) construct was performed using a previously described plasmid and protocol (Schlager et al., 2014a; Urnavicius et al., 2015). Pellets were resuspended on ice in Dynein buffer (50 mM Hepes-KOH, pH 7.3, 150 mM KCl, 5 mM MgCl₂, 2 mM TCEP, 0.2 mM ATP, 10% vol/vol glycerol) supplemented with protease inhibitor cocktail and 1 mg/ml DNAse, and lysed by sonication. The lysate was cleared by centrifugation and sterile filtration. The lysate was loaded on a 5 ml IgG column three times, and the column was washed with 20 CV Dynein buffer. Elution was performed by TEV cleavage of the ZZ tag overnight at 4°C. The cleaved protein was eluted in Dynein lysis buffer with fractionation. The eluate was pooled and concentrated for gel filtration. The concentrated sample was loaded on a Superose 6 10/300 column pre-equilibrated in Dynein buffer. The eluate from gel filtration was pooled and concentrated to ~5 mg/ml, snap-frozen, and stored at –80°C until use.

## Purification of Dynactin from pig brain
[PB]Dynactin was purified essentially as previously described (Zhang et al., 2017). Pig brains were homogenized by blending at 4°C in a Waring blender, in PMEE buffer (35 mM PIPES-KOH, pH 7.2, 5 mM MgSO₄, 1 mM EGTA, 0.5 mM EDTA, 0.1 mM ATP, 2 mM TCEP), supplemented with 1 mg/ml DNAse and protease inhibitor cocktail. The lysate was cleared by two rounds of centrifugation, first at 30,000 $g$ for 20 min at 2°C, then at 235,000 $g$ for 45 min at 4°C, followed by two rounds of filtration, first through a glass fiber filter, then through a 0.45-μm filter. The cleared lysate was loaded on 300 ml SP Sepharose XL resin packed in an XK 50/30 column (Cytiva) and equilibrated in PMEE buffer. The column was washed with 4 CV PMEE. Elution was then performed with a 0–250 mM KCl gradient. Fractions containing Dynactin were initially identified by blotting for p150[glued], pooled, and diluted 1:1 in PMEE. The diluted eluate was loaded on a MonoQ HR 16/10 column equilibrated in PMEE. The column was then washed with 10 CV PMEE. Elution was performed over a 150–350 mM KCl gradient. Dynactin eluted with a peak around 34 mS/cm conductivity. The fractions containing Dynactin were identified by SDS-PAGE, pooled, concentrated, snap-frozen, and stored at –80°C. As a final step of purification, the products of three MonoQ runs were pooled together and loaded on a Superose 6 10/300 column equilibrated in Dynactin buffer (25 mM Hepes-KOH, pH 7.4, 150 mM KCl, 1 mM MgCl₂, 2 mM TCEP, 0.1 mM ATP). The fractions of the peak were analyzed by SDS-PAGE, and those containing Dynactin were pooled and concentrated to a final concentration of 3 mg/ml. The sample was snap-frozen and stored at –80°C until use.

## Expression and purification of recombinant human Dynactin
Recombinant human Dynactin was expressed using Expi293F cells (Invitrogen) and Expi293 expression medium (Invitrogen). Cells were freshly thawed for each expression and passed at least three times before infection. All Expi293F cell cultures were

performed at 37°C, 8% $CO_2$, on an orbital shaker set to 125 rpm. Baculovirus for expression was made fresh for every expression. Sf9 cells were transfected with bacmid produced in EMBacY cells to produce baculovirus, which was then amplified through three 4-d rounds of amplification. The final $V_2$ virus to be used for expression was made by infecting 60 ml $V_2$ Sf9 culture supplemented with 5% FBS, and incubating it for 4 d at 27°C. The culture was then centrifuged at low speed to remove Sf9 cells, and the supernatant was decanted. 20 ml of sterile-filtered PEG solution (32% PEG6000, 400 mM NaCl, 40 mM Hepes, pH 7.4) was added to the supernatant, mixed thoroughly, and the solution was allowed to precipitate overnight at 4°C in the dark. The precipitated virus was pelleted by centrifugation at 4,000 rpm at 4°C for 30 min. The pellet was solubilized in 10 ml Expi293 expression medium prewarmed to 37°C and immediately used to infect 500 ml Expi293F culture, at a density of 3.5–5 x $10^6$ cells per ml. After 8 h from infection, 5 ml of a 1 M stock solution of sterile sodium butyrate in PBS was added to 500 ml Expi293F culture. The expression was incubated at 37°C, 8% $CO_2$ for 48 h after infection. Cells were harvested by centrifugation at 350 $g$ for 15 min, the pellet was washed with PBS, snap-frozen, and stored at –80°C until purification.

The pellet was resuspended in Dynein buffer, supplemented with 1 mg/ml DNAse, protease inhibitor cocktail and 0.3% vol/vol Triton X-100, and lysed by sonication. The lysate was cleared by centrifugation, followed by sterile filtration, and loaded on a 5-ml IgG column. The flowthrough from the IgG column was further loaded three times on 4 × 1 ml IgG beads in gravity flow columns. The column and the beads were washed with 10 CV each Dynein buffer. Elution was performed by TEV cleavage of the ZZ-tag overnight at 4°C. Fractions containing Dynactin were identified by SDS-PAGE, pooled and diluted 1:4 in MonoQ buffer (50 mM Hepes, pH 7.3, 100 mM KCl, 5 mM $MgCl_2$, 2 mM TCEP, 0.1 mM ATP, 10% glycerol). The diluted eluate was loaded on a MonoQ HR 16/10 column. After loading, the column was washed with 10 CV MonoQ buffer. Elution was performed over a 100–500 mM KCl gradient. Recombinant Dynactin eluted around 34 mS/cm conductivity, matching the results for pig brain Dynactin. The eluate was then concentrated in an Amicon 0.5 ml concentrator with a 100 kD molecular mass cut-off to a final concentration of around 3–4 mg/ml. Typical yield from a 500-ml culture was in the range of 100–200 µg.

### In vitro dephosphorylation and phosphorylation

RZZ and Spindly were dephosphorylated using λ-phosphatase. Proteins were diluted to 10 µM in Spindly buffer, and λ-phosphatase was added to a final concentration of 500 nM, and incubated on ice for 15 min. Afterwards, $MnCl_2$ was added at 10 mM concentration. The reaction mixture was incubated overnight at 10°C, and loaded on a Superose 6 10/300 column equilibrated in SEC buffer. The eluate was pooled, concentrated, snap-frozen, and stored at –80°C until use. Pre-dephosphorylated RZZ and Spindly were phosphorylated with the mitotic kinases CDK1:CyclinB, MPS1, and Aurora B, which were all purified in-house (Huis In 't Veld et al., 2021; Raisch et al., 2022; Sessa et al., 2005). Proteins were diluted to 10 µM in Spindly buffer, and kinases were added at a final concentration of 500 nM, and

incubated on ice for 15 min. Afterwards, ATP was added at 1 mM, together with 10 mM $MgCl_2$. The reaction mixture was incubated overnight at 10°C and loaded on a Superose 6 10/300 column equilibrated in SEC buffer. The eluate was pooled, concentrated, snap-frozen, and stored at –80°C until use.

### In vitro farnesylation

Farnesyltransferase α/β mutant (W102T/Y154T) was expressed and purified as previously described (Mosalaganti et al., 2017). Spindly was diluted to 100 µM in farnesylation buffer (50 mM Hepes, pH 8.0, 250 mM NaCl, 10 mM $MgCl_2$, 2 mM TCEP), and farnesyltransferase was added to a final concentration of 30 µM. Farnesyl pyrophosphate was added stepwise to a final concentration of 300 µM. The reaction mixture was incubated at RT for 6 h, after which it was centrifuged at 16,000 $g$ for 10 min to remove precipitate that formed during the reaction. The cleared reaction mixture was then loaded on a Superose 6 column equilibrated in SEC buffer to remove the farnesyltransferase. The fractions containing Spindly were identified by SDS-PAGE and pooled, concentrated, snap-frozen, and stored at –80°C until use.

### Analytical SEC

Analytical SEC was performed under isocratic conditions at 4°C in Spindly buffer on an ÄKTAmicro system. Elution profiles were obtained by monitoring absorbance at 280 nm wavelength. 50 µl fractions were collected and analyzed by SDS-PAGE. Complex formation assays were performed by mixing the samples at the indicated concentrations in 60 µl Spindly buffer and incubating them for at least 1 h on ice before the SEC assay was performed.

### Mass photometry

Mass photometry experiments were performed essentially as described in Sonn-Segev et al. (2020). Standard microscope coverslips were cleaned with MilliQ water and isopropanol, and dried under an air stream. Silicon buffer gaskets were attached to the glass slides, and the slides were then mounted on a Refeyn TwoMP mass photometer (Refeyn Ltd). For measurement, Spindly construct samples were diluted to 100 nM in Spindly buffer immediately before measurement. The gasket was filled with Spindly buffer and the focal plane was automatically estimated. Proteins were diluted 1:10 into the buffer-filled gasket, to a final concentration of 10 nM. A 60-s movie was then recorded through AcquireMP (Refeyn Ltd). Data were processed using the DiscoverMP program. Contrast-to-mass calibration was performed with several known-mass proteins. Mass distributions were plotted with DiscoverMP and mean mass peaks determined by Gaussian fitting.

### AUC

Sedimentation velocity AUC was performed at 42,000 rpm at 20°C in an An-60 Ti rotor (Beckman Coulter), using standard charcoal double-sector centerpieces. Protein samples were diluted in buffer containing 40 mM Hepes, pH 8.0, 200 mM NaCl, and 2 mM TCEP. Approximately 300 radial absorbance scans at 587 nm were collected per each run, with a time interval of

1 min. Buffer density and viscosity, as well as the protein partial specific volume, were estimated using the program SEDNTERP. Sedimentation peaks were identified through an analysis using the SEDFIT suite (Brown and Schuck, 2006), in terms of continuous distribution function of sedimentation coefficients. Frictional ratios were manually set to match the sedimentation coefficient of the main peak to the theoretical mass of the sample, at the stoichiometry predicted by mass photometry data. A fitting run was then performed within SEDFIT, and the resulting estimations for frictional ratio and Stokes' radius were collected as output. Figures were produced with the program GUSSI.

## SEC-MALS

SEC-MALS was performed using a Heleos II 18 angle light scattering instrument (Wyatt), which was coupled to an Optilab rEX online refractive index detector (Wyatt). 100 µl of pig or human dynactin (0.15 mg/ml) were loaded onto a TSKgel G4000SWXL column with a TSKgel SWXL guard column (TOSOH Bioscience) equilibrated in GF150 buffer (25 mM Hepes, pH 7.4, 150 mM KCl, 1 mM $MgCl_2$, 5 mM DTT, 0.1 mM ATP) at RT. The column outlet was directly connected to the light scattering instrument and the refractive index detector. Data collection and determination of molecular weight were performed using the ASTRA 5.3.4 software (Wyatt).

## Negative stain EM

For negative stain EM, 400-square-mesh copper grids (Electron Microscopy Sciences) were plasma cleaned, and 3 µl of protein (≈100 nM in GF150 buffer) were applied to the grid. 20 µl of 2% (wt/vol) uranyl acetate were added to the sample, and excessive stain was removed with a filter paper. Micrographs were recorded on an FEI Tecnai G2 Spirit (120 kV) transmission electron microscope equipped with a Gatan Ultrascan 1000 XP CCD detector. Image acquisition was performed at 260,00× nominal magnification and 1.5 µm underfocus. For 2D classification of recombinant human dynactin, ~5,000 particles were picked semi-automatically using EMAN2 (Tang et al., 2007) and 2D-classified using RELION 2.1 (Scheres, 2012).

## DSBU and UV crosslinking

For DSBU crosslinking, all proteins in the assay were diluted to 5 µM in Spindly buffer. DSBU was added at a final concentration of 3 mM, and the reaction was incubated for 1 h at RT. At the end of the reaction, TRIS-HCl, pH 8.0, at a final concentration of 100 mM was added and the solution was thoroughly mixed to quench the reaction. Successful crosslinking was evaluated by SDS-PAGE analysis. For UV crosslinking, the BPA-containing Spindly mutants were diluted in Spindly buffer to a final concentration of 5 µM. The samples were irradiated with LED UV light at 365-nm wavelength for 15 min to induce complete crosslinking. Successful crosslinking was evaluated by SDS-PAGE analysis.

## Processing for LC/MS

Crosslinked samples were precipitated by dilution into four volumes of acetone pre-cooled to −20°C, followed by overnight incubation at −20°C. Pellets were resuspended and denatured in 8 M urea, 1 mM DTT, and alkylated with 5.5 mM chloroacetamide. The concentration of urea was then lowered to 4 M by dilution in 20 mM ammonium bicarbonate, pH 8.0, and the protein solution was then digested overnight at RT with Trypsin. Digestion was stopped by adding trifluoroacetic acid to a final concentration of 0.2%. Samples were subjected to SEC on a Superdex 30 Increase 3.2/300 (Cytiva), followed by purification on a tC18 Sep-Pak column (50 mg, Waters) as previously reported (Pan et al., 2018). For the [mCh]Spindly[Y26BPA] and [mCh]Spindly[Q29BPA], the SEC step was omitted.

## LC-MS/MS analysis

LC-MS/MS analysis was performed as described in Pan et al. (2018). Data analysis was performed for the DSBU-crosslinked samples in MeroX 2.0.0.8. The analysis of Bpa-crosslinked samples was performed in MeroX 2.0.1.4. An artificial amino acid "Z" was added with the composition C16H13NO2 (mass 251.09463), and Bpa was defined as a crosslinker with parameters: "Specificity site 1:Z", "Specificity Site 2: [ABCDEFGHIKLMPQRSTVWY]", and "Maximum Cα-Cα -distance: 30 Å." For the Spindly[F258BPA] sample, the analysis was performed within MeroX 2.0.1.4 with a false discovery rate (FDR) of 50%. Crosslinks were then exported into XiView (Graham et al., 2019 Preprint), and crosslinks with a score lower than 50 were filtered out. Visual representations of crosslinking data were produced using the xVis website (https://xvis.genzentrum.lmu.de [Grimm et al., 2015]) and then edited to produce the final figures.

## Molecular modeling and bioinformatical methods

AF2 was used for all molecular modeling (Jumper et al., 2021). For the Spindly[1–275], Spindly[1–309], and Spindly[1–440] constructs, the original version of AlphaFold Multimer was used (Evans et al., 2021 Preprint). For all other constructs, the ColabFold version of AF2 was used (Mirdita et al., 2021 Preprint). A general feature of AF2 Multimer is the higher sensitivity to intra- and intermolecular interactions compared to ColabFold and thus a tendency to predict extended protein structures as a more or less compact model, making it difficult to distinguish artificial contacts from structurally significant ones. This problem is worse for fragments containing long, disordered stretches. Therefore, the disordered C-terminal region of Spindly (441–605) was not included in the predicted sequences, unless otherwise specified. In the absence of clear predicted intramolecular contacts from the "predicted alignment errors" (PAEs) plots, the models of the adapters shown in Fig. S3 (obtained with ColabFold) were manually "unfolded" with PyMol and Coot to extend them and make them comparable. To determine the putative interactions in the autoinhibited Spindly fragments, the predicted local distance difference test (pLDDT) scores of the models and especially the PAEs were scrutinized to see which parts of the models were predicted to have defined relative orientations to each other (Fig. S2). Also, some features, like the "kink" at Spindly residues 155–156, were very consistently predicted in almost all models. Other features, like predicting extended conformations of e.g., the Spindly N-terminal coiled coil, were more or less frequent,

depending on the presence of the interacting, autoinhibitory regions, which was interpreted as being indicative of the strength of the co-evolutionary signal present in those regions.

## Crystallographic structure of Spindly[1–100]
Crystallization trials were set up in MRC two-well sitting drop plates by mixing the protein and the reservoir solution in a 1:1 ratio. Initial crystallization hits were optimized, and final crystals were obtained in a condition containing Bis-Tris Propane/HCl, pH 6.5, 19% PEG3350, and 0.2 M potassium thiocianate. For data collection, crystals were cryoprotected in mother liquor containing 20% glycerol prior to flash cooling in liquid $N_2$. Data for native crystals were collected at the MASSIF-1 beamline (Bowler et al., 2015) at the European Synchrotron Radiation Facility, Grenoble, France. For SelenoMethionine protein production, *E. coli* BL21 DE3 cells were grown in ready to use SelenoMethionine medium (Molecular Dimensions) and purified using the same protocol as for the native protein. SeMet crystals could be reproduced in similar crystallization conditions, and anomalous data were collected at I04 beamline of Diamond Light Source. Details for all steps are available in Table S1. Data processing was done with XDS (Kabsch, 2010) and the structure was solved by Se-SAD using the CRANK2 (Pannu et al., 2011) pipeline from the CCP4 (Winn et al., 2011) software suite. The initial model contained 206 residues built in eight fragments and clearly showed the parallel dimeric coiled-coil. Further model building was performed in Coot (Emsley et al., 2010) and subsequent refinement cycles were carried out in REFMAC (Murshudov et al., 2011) and PDB-REDO (Joosten et al., 2014) using non-crystallographic and jelly-body restraints. The structure was refined to 2.8 Å resolution to an $R_{free}$ of 29%. All residues are in the favorable regions of the Ramachandran plot and the structure is in the 97th percentile of Molprobity (Williams et al., 2018).

## Cell culture and drug treatment
HeLa cells were grown in DMEM (PAN Biotech) supplemented with 10% tetracycline-free FBS (PAN Biotech), and L-Glutamine (PAN Biotech). Cells were grown at 37°C in the presence of 5% $CO_2$.

## Cell transfection and electroporation
Depletion of endogenous Spindly was achieved through reverse transfection with 50 nM Spindly siRNA (5′-GAAAGGGUCUCA AACUGAA-3′ obtained from Sigma-Aldrich [Gassmann et al., 2010]) or only with Opti-MEM as a control for 48 h with RNAiMAX (Invitrogen). For rescue experiments, 24 h after Spindly depletion, we electroporated recombinant Spindly constructs labeled with an N-terminal mCh, at a concentration of 7 μM in the electroporation slurry (as previously described in Alex et al., 2019; Neon Transfection System, Thermo Fisher Scientific). Control cells were electroporated with mCh. Following an 8-h recovery, cells were treated with 9 μM RO3306 (Calbiochem) for 15 h. Subsequently, cells were released into mitosis in the presence of 3.3 μM Nocodazole (Sigma-Aldrich) for 1 h before fixation for immunofluorescence or harvesting for immunoblotting.

## Immunofluorescence
Cells were grown on coverslips pre-coated with Poly-L-lysine (Sigma-Aldrich). Cells were fixated with 4% PFA in PHEM (Pipes, pH 6.9, Hepes, EGTA, $MgCl_2$) for 10 min. Subsequently, the cells were permeabilized for 10 min with PHEM supplemented with 0.5% Triton-X100 (PHEM-T). After blocking with 5% boiled goat serum in PHEM buffer, the cells were incubated for 2 h at RT with the following primary antibodies: CENP-C (guinea pig, #PD030, 1:1,000; MBL), Dynactin-p150 (mouse, #610473, 1:400; BD Trans. Lab.), Spindly (rabbit, A301-354A, 1:1,000; Bethyl) diluted in 2.5% boiled goat serum–PHEM-T (PHEM supplemented with 0.1% Triton-X100). Subsequently, cells were incubated for 1 h at RT with the following secondary antibodies: Goat anti-mouse Alexa Fluor 488 (A11001; Invitrogen), donkey anti-rabbit Rhodamine Red (711-295-152; Jackson Immuno Research), and goat anti-guinea pig Alexa Fluor 647 (A-21450; Invitrogen). All washing steps were performed with PHEM-T buffer. DNA was stained with 0.5 μg/ml DAPI (Serva) and Mowiol (Calbiochem) was used as mounting media.

## Cell imaging
Cells were imaged at room temperature using a spinning-disk confocal device on the 3i Marianas system equipped with an Axio Observer Z1 microscope (Zeiss), a CSU-X1 confocal scanner unit (Yokogawa Electric Corporation), 100 × /1.4NA Oil Objectives (Zeiss), and Orca Flash 4.0 sCMOS Camera (Hamamatsu). Images were acquired as z sections at 0.27 μm (using Slidebook Software 6 from Intelligent Imaging Innovations or using LCS 3D software from Leica). Images were converted into maximal intensity projections, exported, and converted into 16-bit TIFF files. Automatic quantification of single kinetochore signals was performed using the software Fiji with background subtraction. Measurements were exported in Excel (Microsoft) and graphed with GraphPad Prism 9.0 (GraphPad Software). Figures were arranged using Adobe Illustrator 2022.

## Statistical analysis
Normality distribution of the data was tested with a Kolmogorov–Smirnov test and found to be nonparametric. Statistical analysis was performed with a nonparametric *t* test comparing two unpaired groups (Mann–Whitney test). Symbols indicate: n.s., $P > 0.05$; *, $P ≤ 0.05$; **, $P ≤ 0.01$; ***, $P ≤ 0.001$; ****, $P ≤ 0.0001$.

## Immunoblotting
Mitotic cells were collected through shake-off and resuspended in lysis buffer (150 mM KCl, 75 mM Hepes [pH 7.5], 1.5 mM EGTA, 1.5 mM $MgCl_2$, 10% glycerol, and 0.075% NP-40 supplemented with protease inhibitor cocktail [Serva] and PhosSTOP phosphatase inhibitors [Roche]). After lysis, whole-cell lysates were centrifuged at 15,000 rpm for 30 min at 4°C. Subsequently, the supernatant was collected and resuspended in sample buffer for analysis by SDS-PAGE and Western blotting. The following primary antibodies were used: Spindly (rabbit, A301-354A, 1:1,000; Bethyl) and Tubulin (mouse monoclonal, 1:8,000; Sigma-Aldrich). As secondary antibodies, anti-mouse or anti-rabbit (1:10,000; NXA931 and NA934; Amersham) conjugated to

horseradish peroxidase were used. After incubation with ECL Western blotting reagent (GE Healthcare), images were acquired with the ChemiDoc MP System (Bio-Rad) using Image Lab 6.0.1 software.

### GST-LIC2 pulldown assay

GST and GST-LIC baits were added to 10 μl gluthathione beads (Serva) at a final concentration of 4 μM in Spindly buffer, in Pierce micro-spin columns (Thermo Fisher Scientific), and incubated at 4°C for 1 h. The unbound baits were removed by centrifugation, and the beads were washed twice. The Spindly preys were added at a final concentration of 8 μM in Spindly buffer. The bait-bound beads were incubated with the preys at 4°C for 1 h, after which the supernatant was removed, and the beads washed twice. After the final wash, both prey and bait were eluted in Spindly buffer supplemented with 50 mM glutathione. The inputs and eluates were analyzed by SDS-PAGE, and the unstained gels were imaged in the 546-nm channel. The signal from the main band was quantified using ImageLab (Biorad), in comparison with the signal of the SpindlyFL band. The gels were then stained with Coomassie and imaged.

### RZZS filaments

RZZS filaments were formed and imaged essentially as described in Raisch et al. (2022). 4 μM mCherry-RZZ was incubated with 8 μM prefarnesylated Spindly in the presence of 1 μM MPS1, in M-buffer (50 mM Hepes, pH 7.5, 100 mM NaCl, 1 mM MgCl$_2$, 2 mM ATP, 1 mM TCEP) at RT overnight. Flow chambers were assembled by placing two parallel strips of double-sided tape onto a glass slide, with a standard coverslip on top, creating a chamber of 5–10 μl volume. The filament sample was diluted to 0.5 μm and loaded into the flow chamber. Imaging was performed on a 3i Marianas system at 100× magnification in the 561-nm channel (see Cell imaging subsection for microscope details). Sample images were acquired as five-stacks of z-sections at 0.27 μm, converted into maximal intensity projections, and processed in Fiji (Schindelin et al., 2012).

### Online supplemental material

Fig. S1 supports Fig. 1 and presents alignment of structural elements discussed in the main text, related structural models, raw data for activation of crosslinker introduce by amber codon suppression, and additional mass photometry results. Fig. S2 also supports Fig. 1 and shows AF2 predictions for different fragments of Spindly. Fig. S3 supports Fig. 2 and shows a gallery of AF2 predictions of adaptors. Fig. S4 supports Fig. 2 and illustrates the production and validation of recombinant Dynactin in HEK293 cells and size-exclusion chromatography experiments obtained with this material. Fig. S5 supports Fig. 3 displaying additional size-exclusion chromatography experiments. Fig. S6 supports Fig. 5 with additional size-exclusion chromatography experiments and in vitro polymerization assays. Fig. S7 supports Fig. 7 and displays the schematics of the Spindly RNAi and electroporation experiments, Spindly depletion, cellular levels of electroporated constructs, and additional analyses of localization data. Table S1 is a Word file containing data and refinement statistics of the crystallization experiment. Table S2

is an Excel file collecting all XL-MS data presented in the main text. Table S3 is an Excel file that reports a summary of the composition of recombinant Dynactin.

### Data availability

All vectors, reagents, and data described in this manuscript are available from Andrea Musacchio upon reasonable request. Coordinates of Spindly[1–100] have been deposited to the PDB with accession no. 8ARF.

## Acknowledgments

We are grateful to Geert Kops for agreeing on renaming the CC2 box as HBS1. We thank Giuseppe Ciossani, Stefan Raunser, and Thomas Surrey for helpful discussions, Dongqing Pan for help with XL-MS, Malte Metz for MS data analysis, and Raphael Gasper-Schönenbrücher for help with biophysical experiments.

This work was supported by the Max Planck Society, the Marie-Curie Training Network DivIDE (project number 675737), the European Research Council through Synergy Grant 951439 (BIOMECANET), the Deutsche Forschungsgemeinschaft (German Research Foundation) through SFB1430 (Project-ID 424228829). Open Access funding provided by the Max Planck Society.

The authors declare no competing financial interests.

Author contributions (following CRediT model): Conceptualization: E. d'Amico, A. Musacchio, A. Perrakis. Funding acquisition: A. Carter, A. Musacchio, A. Perrakis. Investigation: V. Cmentowski, E. d'Amico, F. Müller, S. Maffini, M. Ud Din Ahmad, I.R. Vetter. Project Administration: A. Musacchio, A. Perrakis. Resources: A. Brockmeyer, A. Carter, P. Janning, M. Girbig, S. Wohlgemuth. Supervision: A. Carter, A. Musacchio, A. Perrakis. Validation: A. Musacchio, A. Perrakis, I.R. Vetter. Visualization: V. Cmentowski, E. d'Amico, A. Musacchio, I.R. Vetter. Writing—original draft: E. d'Amico, A. Musacchio. Writing—review & editing: All authors.

Submitted: 1 July 2022

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

# Supplemental material

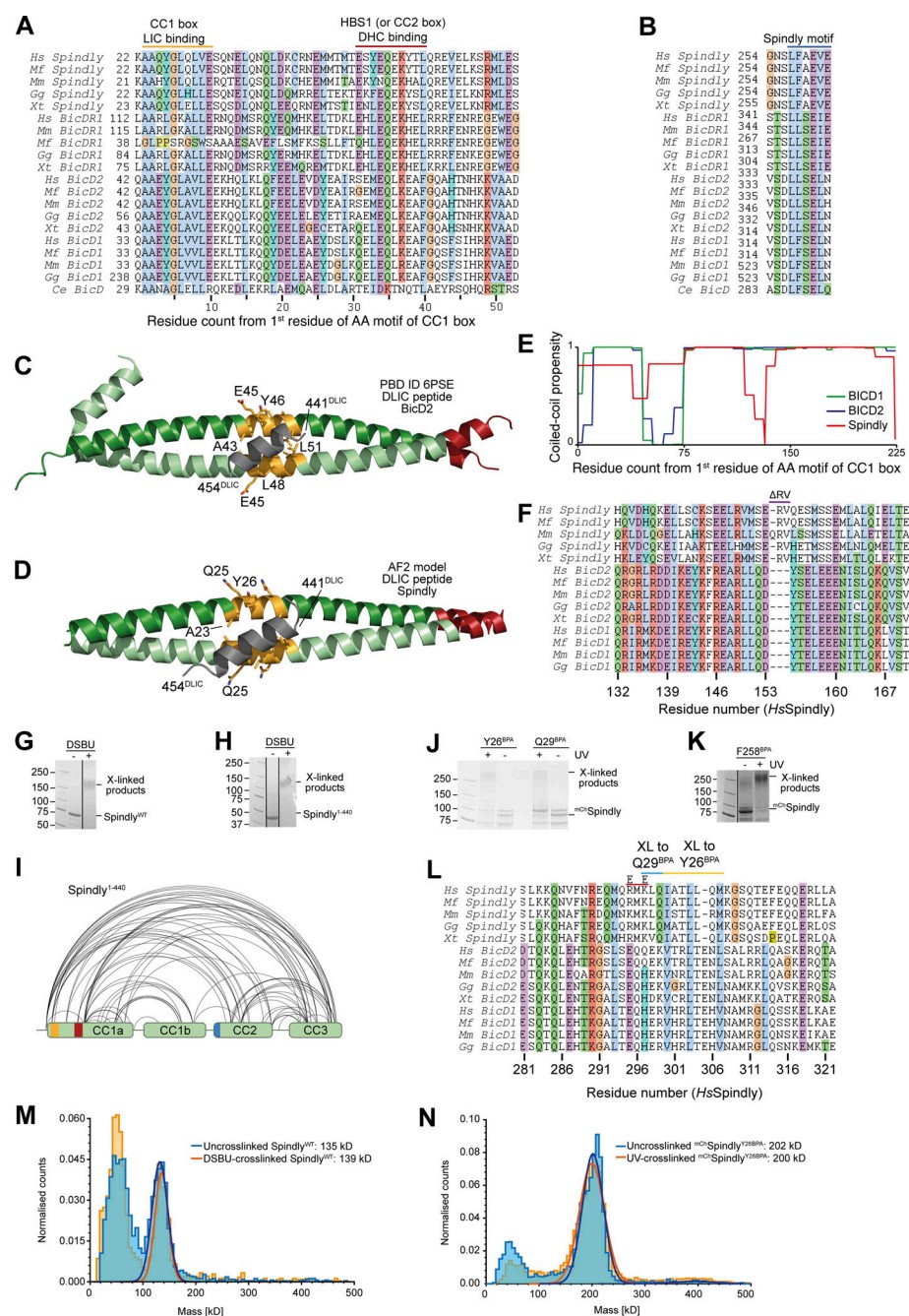

Figure S1.  **Additional analyses of Spindly motifs and their influence on Spindly conformation. (A)** Multiple sequence alignment of the first part of the CC1 region of the indicated adaptors containing the CC1 box and the HBS1 (or CC2 box). Hs, *Homo sapiens*; Mf, *Macaca fascicularis*; Mm, *Mus musculus*; Gg, *Gallus gallus*; Xt, *Xenopus tropicalis*; Ce, *Caenorhabditis elegans*. **(B)** Multiple sequence alignment of the Spindly motif. **(C)** Cartoon model of PDB accession no. 6PSE (Lee et al., 2020) showing the mode of binding of a LIC peptide (gray) to the CC1 box (yellow-orange). **(D)** ColabFold prediction model of the Spindly CC1:LIC peptide complex. Coloring as in C. **(E)** Coiled-coil propensity was predicted with the COILS program within ExPasY suite (Duvaud et al., 2021) and displayed for all indicated adaptors from the first residue of the CC1 box (see A). The coiled-coil propensity for Spindly has a deep that corresponds to a two-residue insertion shown in F. **(F)** Multiple sequence alignment of the region of CC1 around the two-residue insertion in Spindly that causes a deep in the coiled-coil prediction profile (see E). **(G)** SDS-PAGE documenting crosslinking of the full-length Spindly proteins with DSBU. **(H)** SDS-PAGE documenting crosslinking of the Spindly[1–440] proteins with DSBU. G and H were obtained from the same original gel and the marker lane is the same in the two panels. **(I)** Summary of XL-MS data reporting Spindly intramolecular crosslinks. for ease of viewing, only crosslinks detected ≥3 times and involving sites ≥40 residues apart are depicted. See also Table S2. **(J and K)** Coomassie-stained SDS-PAGE gels documenting crosslinking of the indicated BPA mutants upon treatment with UV light. **(L)** Multiple sequence alignment of the indicated adaptors in the main region targeted by Q29[BPA] and Y26[BPA]. The CC2* mutant discussed in the text is the charge reversal mutant (E–E) at the two indicated positively charged residues (R295 and K297 in human Spindly). **(M)** Mass-photometry analysis of DSBU-crosslinked vs. untreated Spindly. The analysis does not detect an enrichment of oligomeric products, and identifies both samples as dimers. The peak at the detection limit of the mass photometer (40–50 kD) is below the expected size of the Spindly monomer (70 kD) and is likely due to slight degradation of the sample. **(N)** Mass-photometry analysis of UV-crosslinked vs. untreated [mCh]Spindly[Y26BPA]. The analysis does not detect an enrichment of oligomeric products. Molecular weights are in kD. Source data are available for this figure: SourceData FS1.

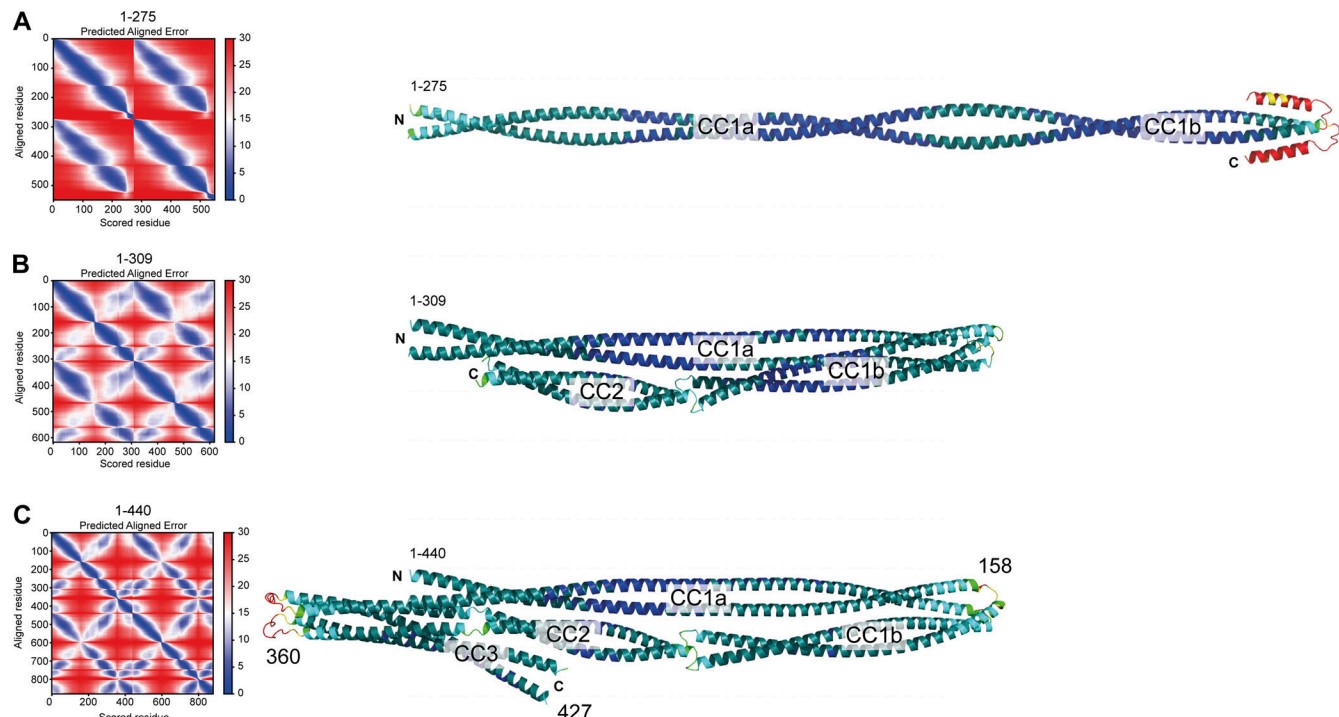

Figure S2. **PAE plots and pLDDT scores. (A–C)** PAE plots (left) and per-residue confidence score (pLDDT) of the three Spindly models discussed in the main text. **(A)** Spindly$^{1–275}$. **(B)** Spindly$^{1–309}$. **(C)** Spindly$^{1–440}$. The pLDDT scores are displayed on the AF2 Multimer predictions of Spindly shown on the right (Blue: high confidence; Red: low confidence). The models are shown, with the same orientation, in Figs. 1 and 2. The PAE matrices refer to models of Spindly dimers, and correspondingly numbering of residues on the left and bottom of the plot is the number of residues in each chain multiplied by 2, and the second chain is plotted directly following the first. The parallel coiled-coils give rise to off-diagonal signals (blue) parallel to the main diagonal. Besides straight models like the one shown, a few predicted models of Spindly$^{1–275}$ also showed a folded-back conformation. However, there are no additional off-diagonal signals for the Spindly$^{1–275}$ construct in the PAE plots, suggesting that even if Spindly$^{1–275}$ explores folded-back conformations, these are not stable. Off-diagonal signals perpendicular to the main diagonal are instead clearly visible in the Spindly$^{1–309}$ and Spindly$^{1–440}$ constructs, consistent with a folded back conformation. A predicted folded conformation of Spindly$^{1–440}$ was also observed with orthologous sequences (unpublished results).

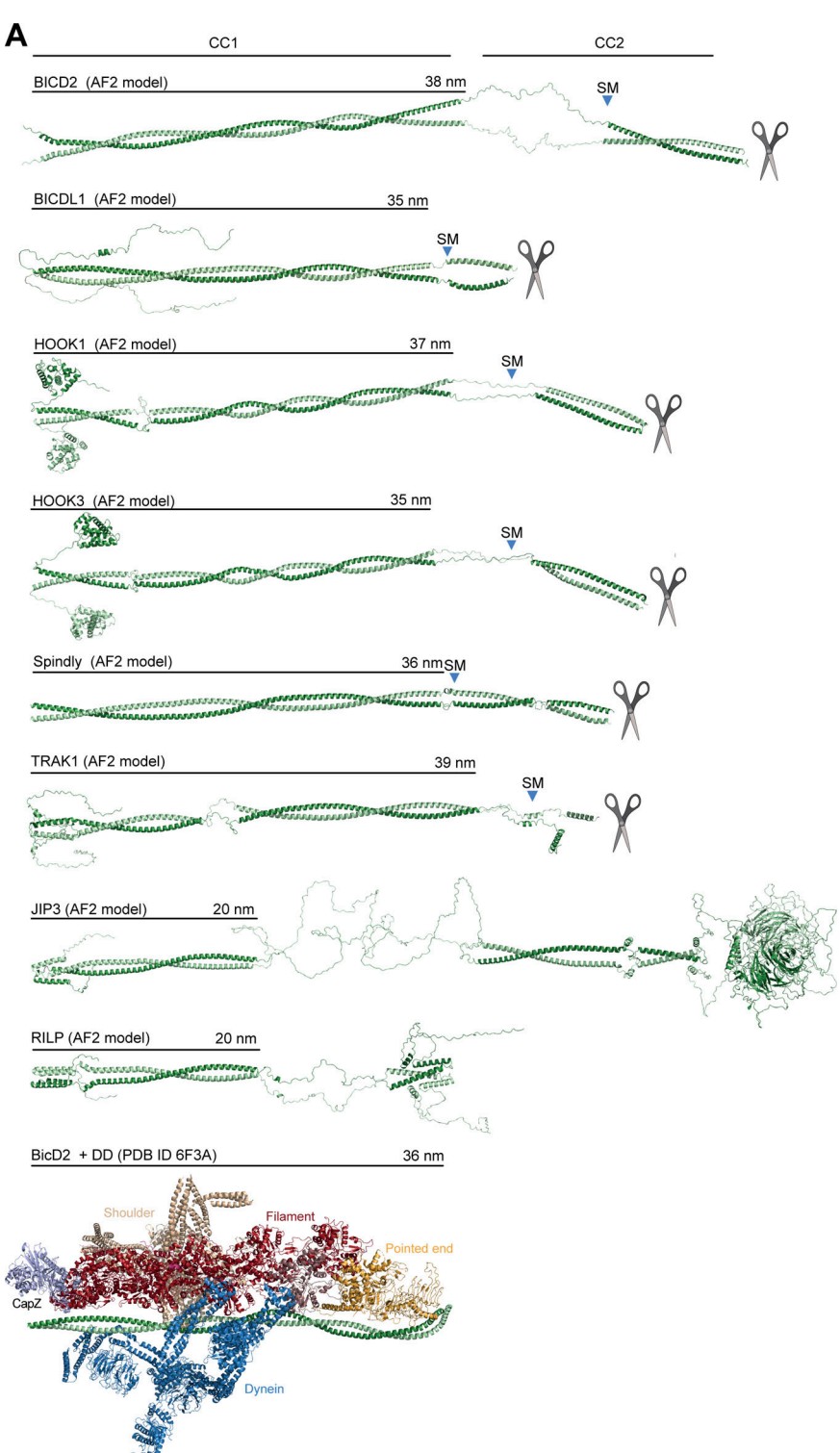

Figure S3. **Gallery of AF2 predictions of the structure of representative adaptors. (A)** AF2 ColabFold models of the indicated adaptors. The scissor symbol indicates that the displayed cartoon models were truncated after the CC2 region. SM is the Spindly motif. The length of CC1 coiled-coil is indicated. As indicated in the Materials and methods section, AF2 and variants can predict different quaternary structures for the adaptors, including trimeric or tetrameric coiled-coil formation, if three or four chains respectively are used as input. However, with trimers or tetramers, the PAE values estimated between the same positions on different protomers are high to very high (i.e., insignificant), indicating that the predictions are less likely to be accurate (unpublished results). We only present predictions of dimers here, as there is experimental evidence for several of them that their active conformation is the dimer (Isabet et al., 2009; Kelkar et al., 2000; Lee et al., 2018; Urnavicius et al., 2018; Urnavicius et al., 2015; Wu et al., 2005). For clarity, models were straightened as discussed in Materials and methods. Even if Spindly is predicted to adopt a closed conformation when CC2 is present (see main text), here for comparison we show the extended open conformation expected to bind DD. The model of the DD complex with BicD2 (PDB accession no. 6F3A; Urnavicius et al., 2018) shows that the length of the experimentally modeled BicD2 coiled-coil is approximately identical to the length of CC1 predicted by AF2 for many of the displayed adaptors.

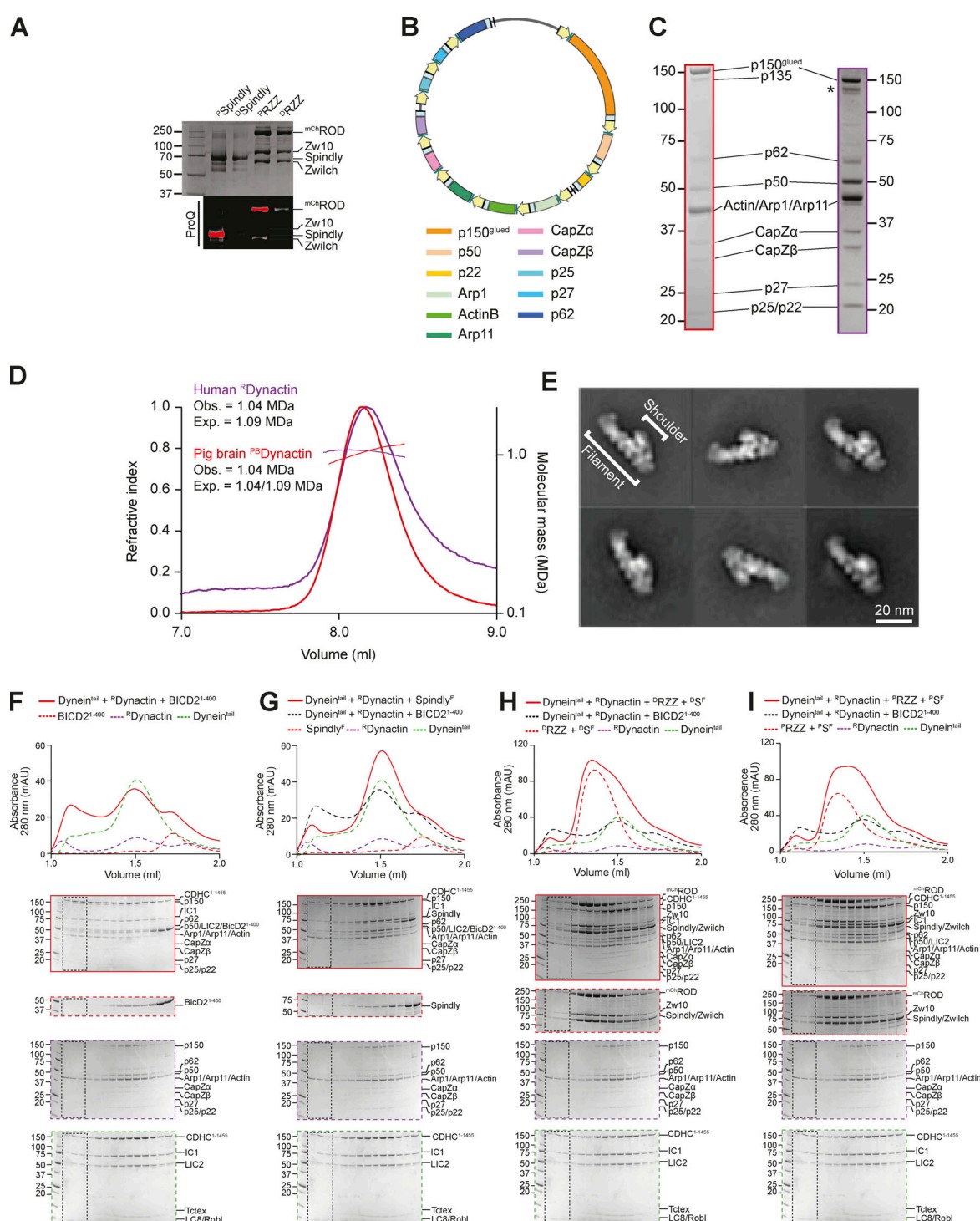

Figure S4. **Recombinant human Dynactin. (A)** Coomassie-stained SDS-PAGE and ProQ Diamond staining of phosphorylated and dephosphorylated RZZ and Spindly. Samples were initially dephosphorylated with λ-PPase (samples indicated as "D") and later re-phosphorylated with the mitotic kinases CDK1/CyclinB, MPS1, and Aurora B (samples indicated as "P"). **(B)** Map of the Dynactin expression plasmid. Individual subunits are labeled according to the list below. CMV promoters and enhancers are labelled in yellow. PolyA signals are labeled in light blue. **(C)** Comparison of SDS-PAGE (Coomassie staining) of PBDynactin (left, red) and recombinant human Dynactin (right, purple). **(D)** SEC-MALS analysis of human recombinant RDynactin (purple) and PBDynactin (red). PBDynactin contains a mix of the p150 isoforms p150glued and p135, yielding two different expected masses. **(E)** 2D class averages from negative stain imaging of RDynactin. **(F–I)** Analytical size-exclusion chromatography on a Superose 6 5/150 column to assess complex formation between an adaptor–cargo/adaptor complex (red, dashed), PBDynactin (purple, dashed), Dynein tail (green, dashed), and with the complex run shown in red. **(F)** BiCD2[1–400]. **(G)** Spindly[FL]. **(H)** RZZ-Spindly[FL] treated with λ-phosphatase. **(I)** RZZ-Spindly[FL] pretreated with a mix of mitotic kinases (MPS1, Aurora B, CDK1/Cyclin B). Dynein tail and RDynactin controls are shared in F–I. In all SEC experiments in this figure, Spindly[FL] was farnsylated. Dynein concentration: 750 nM, Dynactin concentration: 750 nM, Spindly/BicD2 concentration: 4 µM, RZZ concentration: 1 µM. mAU, milli absorbance units. Molecular weights are in kD. Source data are available for this figure: SourceData FS4.

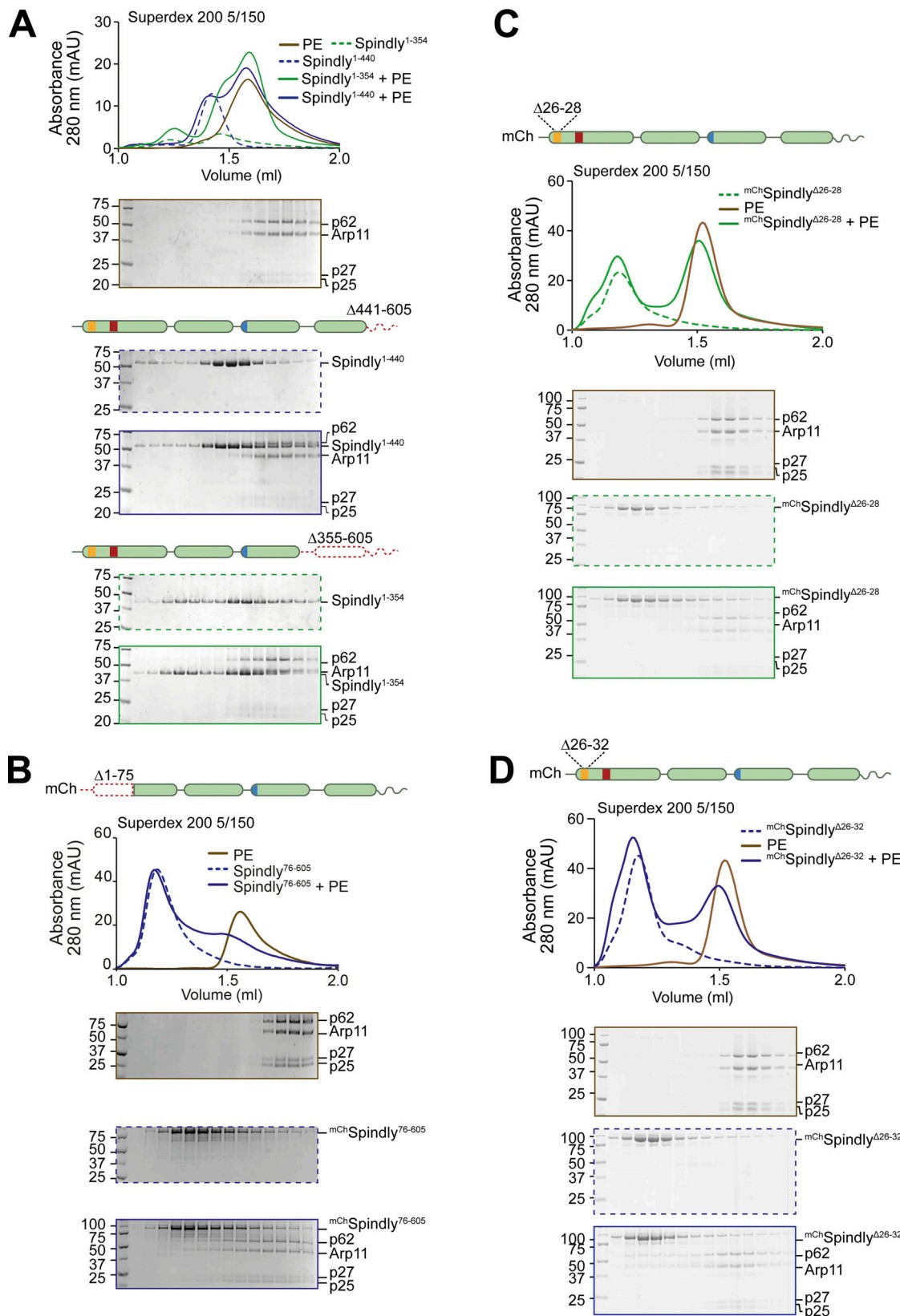

Figure S5. **Additional analyses of the Spindly N-terminal autoinhibitory region. (A–D)** Additional analytical SEC interaction assays between the Dynactin PE (brown) and the indicated Spindly constructs. The complex run is always represented with a continuous line, the Spindly construct with a dashed line. **(A)** Spindly[1–440] (blue); Spindly[1–354] (green). **(B)** mChSpindly[76–605] (blue). **(C)** mChSpindly[Δ26–28] (green). **(D)** mChSpindly[Δ26–32] (blue). PE: 3 µM; Spindly construct: 8 µM. The PE alone control in A is shared with Fig. 3 D, between C and D, and between B and Fig. S6, A and B. mAU, milli absorbance units. Molecular weights are in kD. Source data are available for this figure: SourceData FS5.

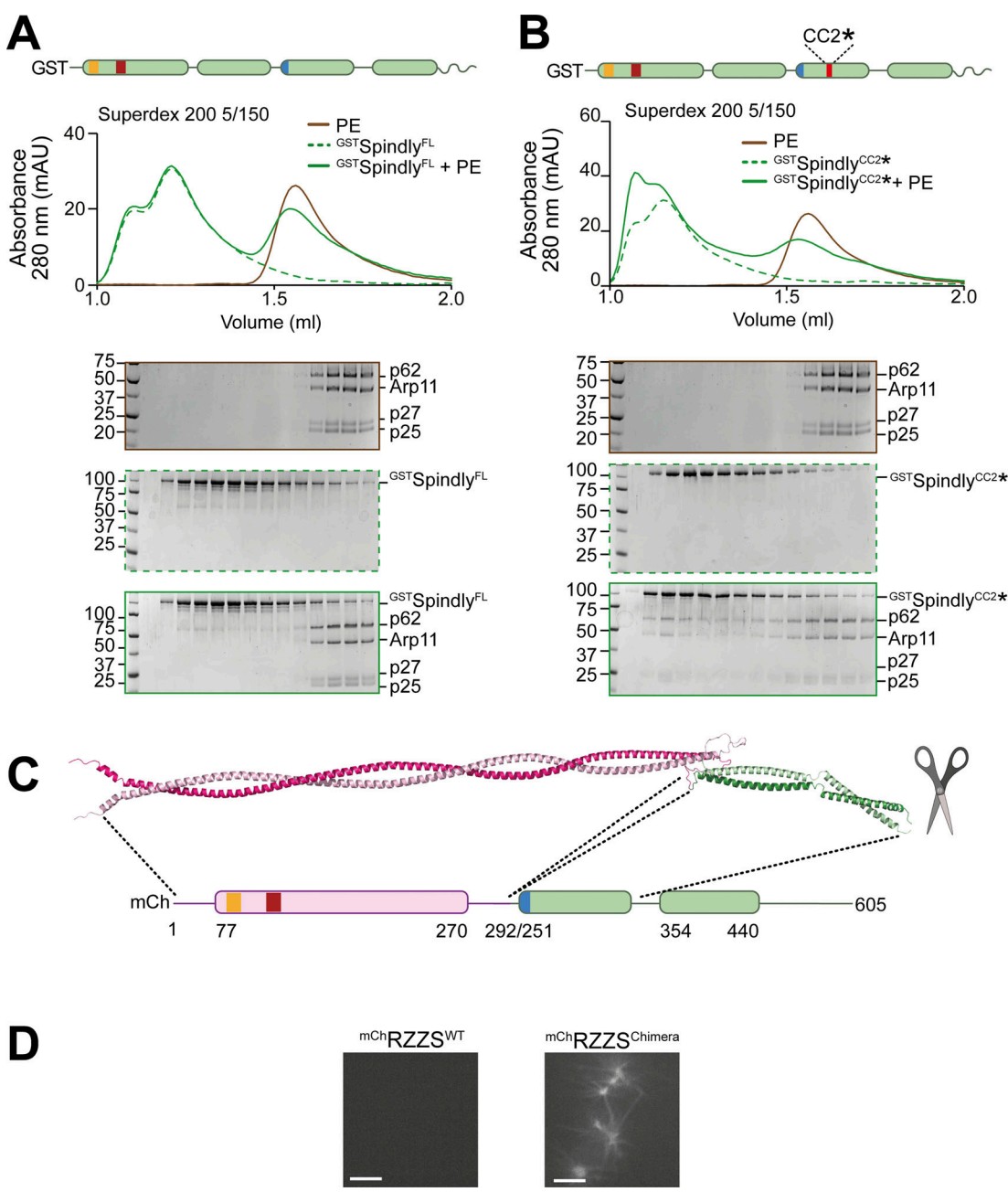

Figure S6. **Additional characterization of Spindly binding to PE. (A)** SEC separation of <sup>GST</sup>Spindly (dotted green line), the PE complex (brown line), and their mixture (continuous green line). **(B)** SEC separation of <sup>GST</sup>Spindly<sup>CC2</sup>* (dotted green line), the PE complex (brown line), and their mixture (continuous green line). The PE control is shared between the two shown experiments and with Fig. S3 B. PE: 3 µM, Spindly construct: 8 µM. **(C)** The AF2 ColabFold model of the BicD2-Spindly chimera shows CC1 is continuous. **(D)** Spinning-disk confocal fluorescence microscopy-based filamentation assay at 561 nm with the indicated <sup>mCh</sup>RZZS<sup>F</sup> species (4 µM RZZ, 8 µM Spindly<sup>F</sup>) at 20°C in absence of MPS1 kinase. mAU, milli absorbance units. Molecular weights are in kD. Scale bar: 5 µm. Source data are available for this figure: SourceData FS6.

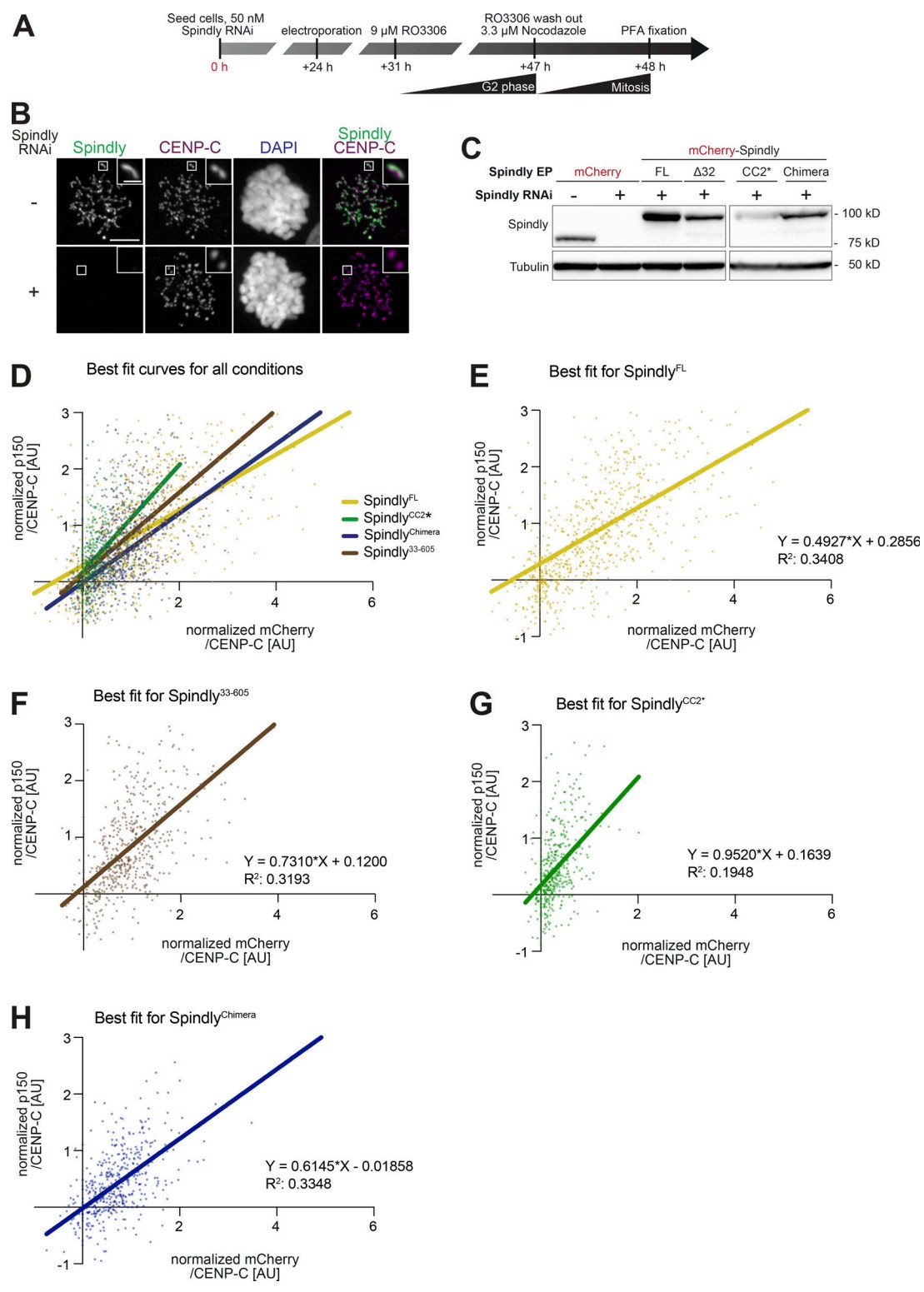

Figure S7. **Additional data on kinetochore levels of Dynactin with Spindly mutants. (A)** Schematic of the RNAi and complementation by electroporation with recombinant proteins. Scale bar: 5 µm (whole cell) or 1 µm (inset). **(B)** Representative images of RNAi control cells and cells depleted of Spindly by RNAi. **(C)** The indicated proteins were electroporated under the same conditions shown in A. 1 h after release from a G2 arrest into mitosis in presence of 3.3 µM Nocodazole, mitotic cells were collected, lysed, and analyzed by immunoblotting with the indicated antibodies. 60 µg of cleared lysate was used for each condition, and Tubulin is shown as a loading control. **(D–H)** Least square fitting through the distribution of data points reporting for each kinetochore the CENP-C–normalized $^{mCh}$Spindly intensity on the x-axis and the CENP-C–normalized p150$^{glued}$ intensity on the y-axis. Data and statistical analyses for these experiments is described in the legend to Fig. 7. **(D)** All fit curves with all data points. **(E)** Individual best fit for $^{mCh}$Spindly$^{FL}$. **(F)** Individual best fit for $^{mCh}$Spindly$^{33–605}$. **(G)** Individual best fit for $^{mCh}$Spindly$^{CC2*}$. **(H)** Individual best fit for $^{mCh}$Spindly$^{Chimera}$. Source data are available for this figure: SourceData FS7.

**Provided online are three tables. Table S1 shows crystallographic data. Table S2 collects all XL-MS data presented in the main text. Table S3 reports a summary of the composition of recombinant Dynactin.**

