## [Peer Review File · The Journal of Cell Biology]

Conformational transitions of the Spindly adaptor underlie its interaction with Dynein and Dynactin

Ennio d'Amico, Misbha Ud Din Ahmad, Verena Cmentowski, Mathias Girbig, Franziska Müller, Sabine Wohlgemuth, Andreas Brockmeyer, Stefano Maffini, Petra Janning, Ingrid Vetter, Andrew Carter, Anastassis Perrakis, and Andrea Musacchio

Corresponding Author(s): Andrea Musacchio, Max Planck Institute of Molecular Physiology

Review Timeline:

Submission Date:	2022-07-01
Editorial Decision:	2022-07-20
Revision Received:	2022-08-02

Monitoring Editor: Erika Holzbaur

Scientific Editor: Lucia Morgado-Palacin

Transaction Report:

DOI: <https://doi.org/10.1083/jcb.202206131>

Revision 0

Review #1

1. Evidence, reproducibility and clarity:

Evidence, reproducibility and clarity (Required)

In this article, Amico et al. explore how Spindly self-regulates its interaction with Dynein-Dynactin. They propose that Spindly adopts an auto-inhibited, closed conformation that blocks the CC1 box and Spindly motif, preventing its interaction with dynein-dynactin. The authors used a combination of X-ray crystallography, biochemistry, and structure predictions to detail the intramolecular interactions in Spindly that mediate this closed state. They then use analytical SEC to test their proposed auto-inhibition mechanism by monitoring Spindly binding to the pointed end complex. They suggest that auto-inhibited Spindly is unable to bind Dynein-Dynactin regardless of the presence or absence of Spindly's cargo, the RZZ complex. In contrast, by using mutagenesis to prevent this auto-inhibition, the authors show that uninhibited Spindly can interact with members of the Dynein-Dynactin complex. Finally, they use cellular experiments to show that relieving autoinhibition prevents the proper localization of Spindly and Dynein-Dynactin to kinetochores during mitosis, likely due to the formation of ectopic Spindly-Dynein-Dynactin complexes in these cells.

This is an interesting paper that provides important insights into the mechanism of Spindly regulation and its associations with its interacting partners. However, additional work is necessary to support some of their conclusions. In addition, the text is at times quite dense and harder to follow, which prevents their findings as being impactful as could be possible for the bigger picture paradigms of kinetochore function.

Major Points:

1. The crosslinking and mass photometry experiments are done at highly differing concentrations (5 μ M vs. 10 nM). The mass photometry should be performed at the same concentration as the crosslinking experiments to determine if Spindly forms a higher order oligomer at the higher concentration. These results will aid in the interpretation of the crosslinking mass spectrometry experiments, as the observed interactions could be intermolecular contacts rather than intramolecular contacts if Spindly is tetrameric at these concentrations, as is suggested in figure 4E for specific Spindly constructs.
2. In figure 2, more conclusive evidence is needed to show that full length Spindly does not form a complex with Dynein-Dynactin. My interpretation of the gels in figure 2D suggests that full length Spindly does form a complex with Dynein-Dynactin, as in the final gel (red outline) it looks as if full length Spindly is indeed peaking with the rest of the Dynein-Dynactin proteins, albeit with excess Spindly eluting later. Figure legends containing protein concentrations used in SEC assays would aid in the interpretation of this data. To conclusively show that full length

Spindly doesn't form a complex with Dynein-Dynactin, additional assays will be necessary, such as pull-down assays, or mass photometry.

3. In figure 3C, 3E, and figure 5C, there is a shift in the PE peaks in the presence of Spindly, but it isn't clear why doesn't the complex doesn't elute earlier than Spindly alone. If the complex is dissociating on the column, additional assays are necessary to confirm that these Spindly constructs stably interact with PE. If this shift is also accompanied by a major change in shape, thus allowing Spindly to elute later than it does alone, this needs to be explored or explained further.

4. The authors should provide better a rationale for why the pointed-end complex is used in figure 3 in lieu of the complex used in figure 2.

5. In Figure 5I, WT Spindly also binds to LIC, although less WT Spindly is bound to LIC than Spindly CC2* or Spindly deltaRV. This should be addressed in the text.

6. The authors claim that the mechanism they describe may be a paradigm for dynein activation by other adaptors at various cellular locations, but they aren't able to identify a mechanism for how Spindly converts from its auto-inhibited state to its permissive state. A more thorough examination of this mechanism is necessary to claim that this mechanism could be paradigmatic, or a revision of the text is needed.

****Minor Points:****

1. The manuscript could benefit from careful review of the text, captions, and figures, as a few minor typos and inconsistencies in the figures and text were present.

2. The list of common structural and functional features of Dynein-Dynactin adaptors could be indicated more clearly.

3. Several times the authors use alpha fold predictions to confirm their data. Although the predictions support several of their conclusions, saying that predictions can confirm the data is an overstatement.

4. Figure 1H would be improved by the addition of the amino acid numbers in the domain diagram.

5. Concentrations used for each protein for the analytical SEC experiments should be listed in the figure or caption.

6. In addition to the caption, it would be helpful to the reader to indicate which experiments use farnesylated Spindly.

7. Error bars are missing from the WT sample in figure 5J. This figure would benefit from statistical analysis.

2. Significance:

Significance (Required)

This paper builds on recent work from the Mussachio lab and others exploring the nature of the fibrous corona at kinetochores and the molecular basis for dynein recruitment. This paper is focused on the structural nature of the interactions that underlie Spindly recruitment to kinetochores and its interactions with dynein and other factors. Although reductionist in its approach, this paper has the potential to have broad implications for thinking about the control of

corona assembly and dynein recruitment with an elegant auto-regulation of Spindly. Researchers interested in cell division, chromosome segregation, kinetochore function, dynein regulation, and the structural basis for core cellular processes should be interested in this paper.

3. How much time do you estimate the authors will need to complete the suggested revisions:

Estimated time to Complete Revisions (Required)

(Decision Recommendation)

Between 1 and 3 months

4. Review Commons values the work of reviewers and encourages them to get credit for their work. Select 'Yes' below to register your reviewing activity at Publons; note that the content of your review will not be visible on Publons.

Reviewer Publons

Yes

Review #2

1. Evidence, reproducibility and clarity:

Evidence, reproducibility and clarity (Required)

The study by d'Amico et al. presents an in-depth analysis of how intramolecular folding of the coiled-coil adaptor Spindly regulates its interaction with the motor dynein and its obligatory co-factor dynactin. Using biochemical reconstitution and diverse biophysical approaches (including cross-linking mass spectrometry, X-ray crystallography, AF2-based structure prediction, size exclusion chromatography, and analytical ultracentrifugation), the authors uncover and dissect an intricate Spindly autoinhibition mechanism. At kinetochores Spindly is known to co-oligomerize into filaments with the RZZ complex (its kinetochore receptor/cargo), which drives expansion of the outermost kinetochore region (the corona). Here the authors show that Spindly is a dimer in solution and that successive coiled-coil segments interact with each other in an asymmetric 'closed' conformation that is unable to form a complex with dynein and dynactin. Specifically, a 2-residue insertion in the middle of Spindly's first coiled-coil (CC1) creates a kink that allows

CC1 to fold back on itself, which has two important structural consequences: it brings a key segment in CC2 (residues 276-309) in contact with a CC1 region called the CC1 box (previously shown to bind dynein light intermediate chain), and it blocks a motif at the beginning of CC2, called the Spindly motif, from accessing the pointed end complex that caps dynactin's minifilament. Mutations in either the CC1 box, the CC1 2-residue insertion, or the CC2(276-309) segment, 'open up' full-length Spindly and promote its interaction with the dynactin pointed end complex and, in case of the latter two types of mutants, with dynein light intermediate chain. CC1 box-deficient Spindly and the CC2 segment mutant (which corresponds to two charge-inverting point mutations) also support complex formation of Spindly and intact dynein-dynactin. Interestingly, while the CC2 mutant can bind to RZZ, the interaction between RZZ and wild-type Spindly is insufficient to make Spindly competent for dynein-dynactin binding (even when RZZ-Spindly are phosphorylated by mitotic kinases). The authors therefore propose that releasing Spindly from autoinhibition requires an additional trigger at the kinetochore, which likely involves an interaction between the Spindly CC2(276-309) segment and an as yet unidentified kinetochore component. The CC2 mutant is also shown to be defective in kinetochore recruitment and in Spindly-RZZ filament formation *in vitro*, suggesting kinetochore recruitment of Spindly is coupled to kinetochore expansion through a mechanism involving CC2(276-309).

The experiments are of excellent technical quality and the results are presented in a logical and concise manner. There is clarity in the writing (the introduction deserves particular praise), and the authors' conclusions are fully supported by the data. Although there is no direct structural evidence for Spindly's closed conformation, as the authors themselves are careful to point out, the numerous Spindly mutants that are characterized (only some of which are mentioned in the summary above) in aggregate make a convincing case for the proposed autoinhibition mechanism.

****Minor comments:****

- Page 5: "605-residue adaptor Spindly". State that "605-residue" refers to the human protein.
- Page 88: "The region of Spindly downstream of the Spindly box (residues 281-322) is very conserved among Spindly orthologues, but not among other members of the BICD adaptor family (Figure 1 - Supplement 1L)." This is not very obvious from the alignment shown in the figure.
- Page 13: "... (A23V-A24V) mutant, which has been previously shown to inhibit the interaction with the LIC2 in a similar assay (Gama et al., 2017)." The LIC isoform used in the referenced study was LIC1.
- Figure 5J: Information about statistical significance should be added.
- Figure 7B - D: Red on black is not an ideal color choice for these graphs.
- Page 15: When discussing the recently discovered interphase functions of Spindly, also cite Clemente et al. (2018; doi:10.3390/jdb6020009) and Conte et al. (2018; doi:10.1242/bio.033233).
- Page 17: "Evidence supporting this idea is that mutations in the 276-306 region, including the deletion of this entire fragment or the introduction of charge-inverting point mutations at residues 295 and 297 respectively abolish or largely decrease the kinetochore recruitment of Spindly ((Raisch et al., 2021) and this study),...". Sacristan et al. (2018) should also be cited in

this context, as this study established the importance of residues 274-287 for Spindly recruitment to kinetochores.

- Page 17: "In vitro, the 276-306 region is also required for the assembly of RZZ-Spindly filaments (this study and (Raisch et al., 2021))." It could also be mentioned here that residues 274-287 of Spindly are necessary for RZZ-Spindly filament formation in cells, as shown by Sacristan et al. (2018).

- Page 17: "Plausibly, the solution to this conundrum will require biochemical reconstitutions addressing the spectrum of interactions that this protein establishes at the kinetochore." Presumably, "this protein" refers to Spindly, but this is not clear since the subject of the preceding sentence is RZZ.

2. Significance:

Significance (Required)

Cargo transport by cytoskeletal motors must be tightly regulated to establish and maintain intracellular organization and for faithful execution of development, including cell division. Much of this regulation occurs at the motor-cargo interface but remains poorly understood at the molecular level. In recent years it has become clear that adaptor proteins not only provide a physical link between motors and their cargo but also participate in motor activation. Adaptor-coupled activation is particularly important for dynein, because adaptors promote dynein's interaction with its essential co-factor dynactin.

BICD2 (along with other Bicaudal D proteins) is the most intensely studied dynein adaptor and has long been known to be subject to autoinhibition with regard to dynein-dynactin binding, which is relieved by cargo binding to the BICD2 C-terminal region. A important question has been whether the same regulatory logic applies to other dynein adaptors. The study by d'Amico et al. presents the first evidence that conformational inhibition extends to adaptors other than Bicaudal D proteins. The study also reveals that Spindly's autoinhibition mechanism is more complex than that of BICD2. This likely reflects Spindly's dual function in dynein-dynactin recruitment and kinetochore expansion. The results of d'Amico et al. suggest that the Spindly autoinhibition mechanism has evolved to coordinate the two processes, and this idea is further supported by a recent study on the RZZ-Spindly interaction from the same group (Raisch et al. 2021; doi:10.1101/2021.12.03.471119). One of the most important insights from d'Amico et al. is that there must be another binding partner of Spindly at kinetochores besides the RZZ complex that participates in the relief of Spindly autoinhibition. The study has therefore identified an important future research direction. It will be interesting to investigate whether additional adaptors follow the multi-step activation model proposed here for Spindly.

Regarding the technical aspects, the study illustrates that AF2-based structure prediction is a powerful tool for investigating conformational regulation, and it introduces an important innovation: the ability to generate recombinant human dynactin opens the door to the engineering of dynactin mutants, which promises to accelerate mechanistic dissection of this essential dynein co-factor.

In conclusion, the study represents a significant step forward in our understanding of how

dynein-cargo interactions are regulated by adaptor proteins and is therefore of general interest for researchers studying the molecular mechanisms of chromosome segregation as well as intracellular transport.

Reviewer expertise keywords: same as the keywords of this manuscript.

3. How much time do you estimate the authors will need to complete the suggested revisions:

Estimated time to Complete Revisions (Required)

(Decision Recommendation)

Less than 1 month

4. Review Commons values the work of reviewers and encourages them to get credit for their work. Select 'Yes' below to register your reviewing activity at Publons; note that the content of your review will not be visible on Publons.

Reviewer Publons

Yes

Review #3

1. Evidence, reproducibility and clarity:

Evidence, reproducibility and clarity (Required)

The Dynein-Dynactin (DD) complex interacts with different activating adaptors to assemble functional motor complexes capable of moving along microtubules while transporting various cargoes. However, it remains poorly understood how DD activation is precisely controlled so that Dynein-mediated transport is only stimulated at the appropriate time and place. DD adaptor regulation is likely a crucial piece of this puzzle. In this manuscript, the authors show that Spindly, a mitotic adaptor of DD complex, undergoes a series of conformational rearrangements that result in efficient Spindly autoinhibition and affect its ability to bind DD. The work from d'Amico et al includes an impressive amount of biochemical and biophysical data, supported by well-designed experiments that are carefully documented. Resorting to crosslinking experiments

and protein structural modelling, the authors find that several intramolecular contacts occur between specialized domains within Spindly N-terminus. The resulting compact conformation occludes important DD-binding motifs in Spindly and, thus, limits the access of DD to the adaptor. By utilizing different Spindly mutants predicted to render the adaptor more elongated, the authors bypass Spindly autoinhibition and rescue binding to DD in vitro. Surprisingly, unlike other DD adaptors, Spindly autoinhibition is not relieved upon binding to its cargo (the RZZ complex) arguing that the interaction with an additional binding partner is required to fully unleash the potential of Spindly to bind DD. In line with this, the authors identify a Spindly mutant that is unable to localize to kinetochores from human cells, despite its open conformation. Collectively, this work provides significant advances in the understanding of Spindly regulation and brings a new perspective to the mechanism of DD adaptor activation and therefore should be of interest for a wide audience.

****Major concerns:****

- The authors show that Spindly 33-605 is able to form a complex with DD which eventually enables the recruitment of Dynactin to kinetochores from Spindly 33-605-expressing cells. This result is unexpected since this Spindly mutant lacks CC1 box, which has been previously shown to be required for the kinetochore localization of Dynactin (Sacristan et al 2018). A more comprehensive discussion about this discrepancy would enrich the article and benefit the audience.
- In Fig.7, the authors show that two Spindly mutants (Spindly CC2* and Spindly chimera) are unable to fully decorate the kinetochores from human cells. The same is true for Spindly AA/VV mutant. Do the authors know whether these mutants are expressed as stable proteins in cells for example by performing a western blot analysis?
- In line with the previous point, could the authors tether each Spindly mutant to the kinetochore for example by fusing the construct to known kinetochore proteins such as Mis12 and test whether these fusion constructs are now able to recruit Dynactin to kinetochores?
- The authors conclude that the 2-step or multistep mechanism involved in the regulation of Spindly activation may be a common mechanism to different DD adaptors. However, the authors point out to existing differences between the conformational arrangement of Spindly and another DD adaptor, BICD2, arguing against a common mode of regulation for all adaptors. This needs to be clarified.

****Minor concerns:****

- In Fig.2D, full length Spindly does not bind DD in vitro. This is most likely to occur because Spindly N-terminus adopts a compacted conformation and hinders the access to DD-binding motifs. In Fig.2B, the authors show a structural prediction for Spindly 1-275 which should adopt a more elongated shape. According to prevailing model, this construct should now be able to bind DD in a similar biochemical assay.

- In Gama et al 2017, LIC1 was able to pull down a wild-type N-terminal Spindly construct. How do the authors reconcile this with the data presented in this manuecript?
- The section where the authors test point mutations to open Spindly ("Opening up Spindly with point mutations") should be better contextualized. The transition is difficult to follow as it is.
- In the text, it is not clear whether Mps1 kinase is required to promote RZZ oligomerization in the presence of Spindly chimera, an uninhibited Spindly mutant. According to the model, this mutant construct should drive oligomerization independently of Mps1 (as the N-terminal deletion construct from Sacristan et al 2018).
- The nomenclature the authors adopt for the CC1 second conserved motif (SCM) and for the Spindly motif (SM) can be confusing at some point when identifying each mutant in the text and figures. Nomenclature should be standardized.
- In Fig.6A, mCh-Spindly 33-605 and mCh-Spindly chimera lines have the same color.

2. Significance:

Significance (Required)

The work represents a significant advance in the field and it would be of interest for a wide range of audiences.

3. How much time do you estimate the authors will need to complete the suggested revisions:

Estimated time to Complete Revisions (Required)

(Decision Recommendation)

Less than 1 month

4. Review Commons values the work of reviewers and encourages them to get credit for their work. Select 'Yes' below to register your reviewing activity at Publons; note that the content of your review will not be visible on Publons.

Reviewer Publons

Yes

Manuscript number: RC-2022-01277

Corresponding author(s): Andrea, Musacchio

1. General Statements [optional]

In view of the considerable support received by the three reviewers during the review process, we felt there were the conditions for a full revision. A full list of changes to the original manuscript is appended at the end of the point-by-point description of the revisions.

This section is mandatory. Please insert a point-by-point reply describing the revisions that were already carried out and included in the transferred manuscript.

Reviewer #1

In this article, Amico et al. explore how Spindly self-regulates its interaction with Dynein-Dynactin. They propose that Spindly adopts an auto-inhibited, closed conformation that blocks the CC1 box and Spindly motif, preventing its interaction with dynein-dynactin. The authors used a combination of X-ray crystallography, biochemistry, and structure predictions to detail the intramolecular interactions in Spindly that mediate this closed state. They then use analytical SEC to test their proposed auto-inhibition mechanism by monitoring Spindly binding to the pointed end complex. They suggest that auto-inhibited Spindly is unable to bind Dynein-Dynactin regardless of the presence or absence of Spindly's cargo, the RZZ complex. In contrast, by using mutagenesis to prevent this auto-inhibition, the authors show that uninhibited Spindly can interact with members of the Dynein-Dynactin complex. Finally, they use cellular experiments to show that relieving autoinhibition prevents the proper localization of Spindly and Dynein-Dynactin to kinetochores during mitosis, likely due to the formation of ectopic Spindly-Dynein-Dynactin complexes in these cells.

This is an interesting paper that provides important insights into the mechanism of Spindly regulation and its associations with its interacting partners. However, additional work is necessary to support some of their conclusions. In addition, the text is at times quite dense and harder to follow, which prevents their findings as being impactful as could be possible for the bigger picture paradigms of kinetochore function.

We thank the reviewer for a supportive assessment and for raising some concerns that we have now fully addressed in our revision.

Major Points:

1. The crosslinking and mass photometry experiments are done at highly differing concentrations (5 μ M vs. 10 nM). The mass photometry should be performed at the same

concentration as the crosslinking experiments to determine if Spindly forms a higher order oligomer at the higher concentration. These results will aid in the interpretation of the crosslinking mass spectrometry experiments, as the observed interactions could be intermolecular contacts rather than intramolecular contacts if Spindly is tetrameric at these concentrations, as is suggested in figure 4E for specific Spindly constructs.

We thank the reviewer for raising this point. Mass photometry (MP) requires very low sample concentrations as it is essentially a single molecule technique, and therefore the particle density cannot be increased arbitrarily. To assess whether the Spindly construct is prevalently tetrameric at the concentration of the crosslinking experiment, we performed the crosslinking experiments at the standard concentration, and only then diluted the samples and performed MP measurements. The results, displayed in two new panels (Figure 1 – Supplement 1M-N), show that crosslinked samples are primarily dimeric, providing further evidence that we are looking at *bona fide* intra-dimer contacts.

2. In figure 2, more conclusive evidence is needed to show that full length Spindly does not form a complex with Dynein-Dynactin. My interpretation of the gels in figure 2D suggests that full length Spindly does form a complex with Dynein-Dynactin, as in the final gel (red outline) it looks as if full length Spindly is indeed peaking with the rest of the Dynein-Dynactin proteins, albeit with excess Spindly eluting later. Figure legends containing protein concentrations used in SEC assays would aid in the interpretation of this data.

We have now added concentrations of binding species at the relevant points of the figure legends.

To conclusively show that full length Spindly doesn't form a complex with Dynein-Dynactin, additional assays will be necessary, such as pull-down assays, or mass photometry.

The essence of the reviewer's concern is that full length Spindly, like BicD2, binds the DD, which would invalidate our model that Spindly is auto-inhibited in absence of a second trigger (other than DD), or alternatively showing that auto-inhibition can be easily overcome. Our conclusion that Spindly remains auto-inhibited, however, is strongly supported by the gels in Figures 2D-G. There, the peak containing DD and BicD2 and eluting around 6.2 ml (panel D) is not visible when BicD2 is replaced with Spindly (panel E), and RZZ does not change this (panels F-G). Note that the peak at 6.6 ml appears to be a contaminant, possibly DNA, and it is visible also with individual Dynein and Dynactin samples. These experiments strongly support our point and we have tried to improve the presentation of the results by boxing relevant fractions of the displayed SDS-PAGEs.

We have now also repeated these experiments with recombinant human Dynactin. The new results are displayed in Figure 2 – Supplement 2. Also in this case, we see minimal complex formation with Spindly and complex formation with BicD2, even if the trailing of Dynein, Dynactin, and Spindly in the earlier elution fractions (already in the absence of complex

formation) makes the gels harder to interpret. We also note that these experiments are consistent with those with the isolated PE complex.

Regretfully, we cannot gather additional information by mass photometry because even our positive control dissociates at the extremely low concentrations required to image this very large complex.

3. In figure 3C, 3E, and figure 5C, there is a shift in the PE peaks in the presence of Spindly, but it isn't clear why doesn't the complex doesn't elute earlier than Spindly alone. If the complex is dissociating on the column, additional assays are necessary to confirm that these Spindly constructs stably interact with PE. If this shift is also accompanied by a major change in shape, thus allowing Spindly to elute later than it does alone, this needs to be explored or explained further.

Elution from a size exclusion chromatography column is dominated by the hydrodynamic radius of the macromolecule. In this particular case, Spindly is highly elongated and essentially sets an upper limit for the elution volume of both the un-complexed and complexed protein. We have described this behavior in many other cases of highly elongated proteins (e.g. Huis in 't Veld *et al.* eLife 2019). We are aware that the binding affinity for the interaction of Spindly and the PE complex is low, and therefore are not surprised to observe dissociation of the complex during the SEC run, i.e. upon dilution of the sample after incubation. In these experiments, we have tried to focus on the shift in elution volume of the PE complex from its elution position in isolation.

4. The authors should provide better a rationale for why the pointed-end complex is used in figure 3 in lieu of the complex used in figure 2.

We now write that the Spindly motif of adaptors binds the pointed end complex with measurable affinity also in absence of Dynein (near line 294). We then clarify that "As the Spindly motif is predicted to sit within the autoinhibited portion of the protein, we hypothesized that the PE-Spindly motif interaction could be used as a proxy to measure the autoinhibition status of Spindly, bypassing the need to form the entire Dynein-Dynactin-Spindly complex."

5. In Figure 5I, WT Spindly also binds to LIC, although less WT Spindly is bound to LIC than Spindly CC2* or Spindly deltaRV. This should be addressed in the text.

Thank you for pointing this out. We have now clarified this in the text near line 440.

6. The authors claim that the mechanism they describe may be a paradigm for dynein activation by other adaptors at various cellular locations, but they aren't able to identify a mechanism for how Spindly converts from its auto-inhibited state to its permissive state. A more thorough examination of this mechanism is necessary to claim that this mechanism could be paradigmatic, or a revision of the text is needed.

Following an additional concern by reviewer 3, we have now revised the text to meet this concern. So, both in the last sentence of the abstract, and in the last paragraph of the discussion, we do not any longer discuss our results as paradigmatic, although we have reasons to believe that they might be eventually recognized as such, after additional examples will have been analyzed.

Minor Points:

1) The manuscript could benefit from careful review of the text, captions, and figures, as a few minor typos and inconsistencies in the figures and text were present.

We have now re-reviewed the text and figures to try eliminate residual inconsistencies.

2) The list of common structural and functional features of Dynein-Dynactin adaptors could be indicated more clearly.

We have re-written this part of the Introduction, where we now indicate more clearly the features of the DD complex

3) Several times the authors use alpha fold predictions to confirm their data. Although the predictions support several of their conclusions, saying that predictions can confirm the data is an overstatement.

We thank the reviewer for pointing this out. We now replaced “confirmed” with “also supported” on line 190, where we explicitly referred to AF2 predictions as “confirmatory”. We also re-wrote a statement in the Discussion where we had commented on the power of AF2 and indicate that it “became available in the late phases of our work as a guiding and validation tool” (line 524).

4) Figure 1H would be improved by the addition of the amino acid numbers in the domain diagram.

Fixed – we also added amino acid numbers in 1G for consistency.

5) Concentrations used for each protein for the analytical SEC experiments should be listed in the figure or caption.

Thank you for suggesting this. We have now added the protein concentrations for these experiments directly in the legends.

6) In addition to the caption, it would be helpful to the reader to indicate which experiments use farnesylated Spindly.

Done in legends wherever applicable.

7) Error bars are missing from the WT sample in figure 5J. This figure would benefit from statistical analysis.

Done – see also point 4, Reviewer 2.

Significance:

This paper builds on recent work from the Mussachio lab and others exploring the nature of the fibrous corona at kinetochores and the molecular basis for dynein recruitment. This paper is focused on the structural nature of the interactions that underlie Spindly recruitment to kinetochores and its interactions with dynein and other factors. Although reductionist in its approach, this paper has the potential to have broad implications for thinking about the control of corona assembly and dynein recruitment with an elegant auto-regulation of Spindly. Researchers interested in cell division, chromosome segregation, kinetochore function, dynein regulation, and the structural basis for core cellular processes should be interested in this paper.

Reviewer #2

The study by d'Amico et al. presents an in-depth analysis of how intramolecular folding of the coiled-coil adaptor Spindly regulates its interaction with the motor dynein and its obligatory co-factor dynactin. Using biochemical reconstitution and diverse biophysical approaches (including cross-linking mass spectrometry, X-ray crystallography, AF2-based structure prediction, size exclusion chromatography, and analytical ultracentrifugation), the authors uncover and dissect an intricate Spindly autoinhibition mechanism. At kinetochores Spindly is known to co-oligomerize into filaments with the RZZ complex (its kinetochore receptor/cargo), which drives expansion of the outermost kinetochore region (the corona). Here the authors show that Spindly is a dimer in solution and that successive coiled-coil segments interact with each other in an asymmetric 'closed' conformation that is unable to form a complex with dynein and dynactin. Specifically, a 2-residue insertion in the middle of Spindly's first coiled-coil (CC1) creates a kink that allows CC1 to fold back on itself, which has two important structural consequences: it brings a key segment in CC2 (residues 276-309) in contact with a CC1 region called the CC1 box (previously shown to bind dynein light intermediate chain), and it blocks a motif at the beginning of CC2, called the Spindly motif, from accessing the pointed end complex that caps dynactin's minifilament. Mutations in either the CC1 box, the CC1 2-residue insertion, or the CC2(276-309) segment, 'open up' full-length Spindly and promote its interaction with the dynactin pointed end complex and, in case of the latter two types of mutants, with dynein light intermediate chain. CC1 box-deficient Spindly and the CC2 segment mutant (which corresponds to two charge-inverting point mutations) also support complex formation of Spindly and intact dynein-dynactin. Interestingly, while the CC2 mutant can bind to RZZ, the interaction between RZZ and wild-type Spindly is insufficient to make Spindly competent for dynein-dynactin binding (even when RZZ-Spindly are phosphorylated by mitotic kinases). The authors therefore propose that releasing Spindly from autoinhibition requires an additional trigger at the kinetochore, which

likely involves an interaction between the Spindly CC2(276-309) segment and an as yet unidentified kinetochore component. The CC2 mutant is also shown to be defective in kinetochore recruitment and in Spindly-RZZ filament formation in vitro, suggesting kinetochore recruitment of Spindly is coupled to kinetochore expansion through a mechanism involving CC2(276-309).

The experiments are of excellent technical quality and the results are presented in a logical and concise manner. There is clarity in the writing (the introduction deserves particular praise), and the authors' conclusions are fully supported by the data. Although there is no direct structural evidence for Spindly's closed conformation, as the authors themselves are careful to point out, the numerous Spindly mutants that are characterized (only some of which are mentioned in the summary above) in aggregate make a convincing case for the proposed autoinhibition mechanism.

We are very grateful to the reviewer for supporting our work

Minor comments:

- Page 5: "605-residue adaptor Spindly". State that "605-residue" refers to the human protein.

We have added this clarification

- Page 8: "The region of Spindly downstream of the Spindly box (residues 281-322) is very conserved among Spindly orthologues, but not among other members of the BICD adaptor family (Figure 1 - Supplement 1L)." This is not very obvious from the alignment shown in the figure.

We agree with the reviewer that the text, as written, was confusing. We have now rephrased it and write "Downstream of the Spindly box, sequences of Spindly orthologues and BICD family adaptors diverge"

- Page 13: "... (A23V-A24V) mutant, which has been previously shown to inhibit the interaction with the LIC2 in a similar assay (Gama et al., 2017)." The LIC isoform used in the referenced study was LIC1.

Thank you for identifying this error. We have corrected the text accordingly.

-Figure 5J: Information about statistical significance should be added.

Done. See also Minor point 7, Reviewer 1.

-Figure 7B - D: Red on black is not an ideal color choice for these graphs.

We now replaced red with yellow

-Page 15: When discussing the recently discovered interphase functions of Spindly, also cite Clemente et al. (2018; doi:10.3390/jdb6020009) and Conte et al. (2018; doi:10.1242/bio.033233).

We apologize for the involuntary omission of these two references, which have now been included in the revised manuscript.

-Page 17: "Evidence supporting this idea is that mutations in the 276-306 region, including the deletion of this entire fragment or the introduction of charge-inverting point mutations at residues 295 and 297 respectively abolish or largely decrease the kinetochore recruitment of Spindly ((Raisch et al., 2021) and this study),...". Sacristan et al. (2018) should also be cited in this context, as this study established the importance of residues 274-287 for Spindly recruitment to kinetochores.

We agree and apologize for the inadvertent omission. We have now included the Sacristan et al. reference in this context.

- Page 17: "In vitro, the 276-306 region is also required for the assembly of RZZ-Spindly filaments (this study and (Raisch et al., 2021))." It could also be mentioned here that residues 274-287 of Spindly are necessary for RZZ-Spindly filament formation in cells, as shown by Sacristan et al. (2018).

We have now reported this fact on lines 560-561.

- Page 17: "Plausibly, the solution to this conundrum will require biochemical reconstitutions addressing the spectrum of interactions that this protein establishes at the kinetochore." Presumably, "this protein" refers to Spindly, but this is not clear since the subject of the preceding sentence is RZZ.

Done – line 565

Significance

Cargo transport by cytoskeletal motors must be tightly regulated to establish and maintain intracellular organization and for faithful execution of development, including cell division. Much of this regulation occurs at the motor-cargo interface but remains poorly understood at the molecular level. In recent years it has become clear that adaptor proteins not only provide a physical link between motors and their cargo but also participate in motor activation. Adaptor-coupled activation is particularly important for dynein, because adaptors promote dynein's interaction with its essential co-factor dynactin.

BICD2 (along with other Bicaudal D proteins) is the most intensely studied dynein adaptor and has long been known to be subject to autoinhibition with regard to dynein-dynactin binding, which is relieved by cargo binding to the BICD2 C-terminal region. A important question has

been whether the same regulatory logic applies to other dynein adaptors. The study by d'Amico et al. presents the first evidence that conformational inhibition extends to adaptors other than Bicaudal D proteins. The study also reveals that Spindly's autoinhibition mechanism is more complex than that of BICD2. This likely reflects Spindly's dual function in dynein-dynactin recruitment and kinetochore expansion. The results of d'Amico et al. suggest that the Spindly autoinhibition mechanism has evolved to coordinate the two processes, and this idea is further supported by a recent study on the RZZ-Spindly interaction from the same group (Raisch et al. 2021; doi:10.1101/2021.12.03.471119). One of the most important insights from d'Amico et al. is that there must be another binding partner of Spindly at kinetochores besides the RZZ complex that participates in the relief of Spindly autoinhibition. The study has therefore identified an important future research direction. It will be interesting to investigate whether additional adaptors follow the multi-step activation model proposed here for Spindly.

Regarding the technical aspects, the study illustrates that AF2-based structure prediction is a powerful tool for investigating conformational regulation, and it introduces an important innovation: the ability to generate recombinant human dynactin opens the door to the engineering of dynactin mutants, which promises to accelerate mechanistic dissection of this essential dynein co-factor.

In conclusion, the study represents a significant step forward in our understanding of how dynein-cargo interactions are regulated by adaptor proteins and is therefore of general interest for researchers studying the molecular mechanisms of chromosome segregation as well as intracellular transport.

Reviewer #3

The Dynein-Dynactin (DD) complex interacts with different activating adaptors to assemble functional motor complexes capable of moving along microtubules while transporting various cargoes. However, it remains poorly understood how DD activation is precisely controlled so that Dynein-mediated transport is only stimulated at the appropriate time and place. DD adaptor regulation is likely a crucial piece of this puzzle. In this manuscript, the authors show that Spindly, a mitotic adaptor of DD complex, undergoes a series of conformational rearrangements that result in efficient Spindly autoinhibition and affect its ability to bind DD. The work from d'Amico et al includes an impressive amount of biochemical and biophysical data, supported by well-designed experiments that are carefully documented. Resorting to crosslinking experiments and protein structural modelling, the authors find that several intramolecular contacts occur between specialized domains within Spindly N-terminus. The resulting compact conformation occludes important DD-binding motifs in Spindly and, thus, limits the access of DD to the adaptor. By utilizing different Spindly mutants predicted to render the adaptor more elongated, the authors bypass Spindly autoinhibition and rescue binding to DD in vitro. Surprisingly, unlike other DD adaptors, Spindly autoinhibition is not relieved upon binding to its cargo (the RZZ complex) arguing that the interaction with an additional binding partner is required to fully unleash the potential of Spindly to bind DD. In line with this, the authors identify a Spindly mutant that is unable to localize to kinetochores from human cells, despite its open conformation. Collectively,

this work provides significant advances in the understanding of Spindly regulation and brings a new perspective to the mechanism of DD adaptor activation and therefore should be of interest for a wide audience.

We are very grateful to the reviewer for the support and for the thorough and constructive evaluation of our work.

Major concerns:

- The authors show that Spindly 33-605 is able to form a complex with DD which eventually enables the recruitment of Dynactin to kinetochores from Spindly 33-605-expressing cells. This result is unexpected since this Spindly mutant lacks CC1 box, which has been previously shown to be required for the kinetochore localization of Dynactin (Sacristan et al 2018). A more comprehensive discussion about this discrepancy would enrich the article and benefit the audience.

We thank the reviewer for pointing this out. We now write (line 492): “This result was unexpected, because the CC1 box has been previously shown to be required for kinetochore localization of Dynactin (Sacristan et al., 2018)”

- In Fig.7, the authors show that two Spindly mutants (Spindly CC2* and Spindly chimera) are unable to fully decorate the kinetochores from human cells. The same is true for Spindly AA/VV mutant. Do the authors know whether these mutants are expressed as stable proteins in cells for example by performing a western blot analysis?

In this revised version of our manuscript, we have explained more clearly that in this experiment we electroporate recombinant proteins. These are essentially the same proteins that we use for the experiments *in vitro*. This provides an internal test in these experiments, because we can verify, through their successful expression and purification, that the proteins are stable. We cannot exclude, however, that the proteins are “treated differently” in cells, for instance because they interact differently with certain binding partners in ways that modifies their stability. As the proteins are not expressed continuously, but rather introduced in the cells in a single electroporation event several hours before imaging, the overall levels of these proteins may differ. We have now included a representative western blot (Figure 7 – Supplement 1C) that demonstrates the levels of electroporated proteins in the experiments in Figure 7. Spindly^{CC2*} appears to be present at somewhat lower levels than the other constructs. ^{mCh}Spindly³³⁻⁶⁰⁵ and Spindly^{chimera}, on the other hand, were present at very similar levels, supporting our conclusion that a kinetochore-binding region is impaired in the latter. We now refer in the main text to the uncertainty created by the comparatively lower cellular levels of Spindly^{CC2*}. We have also chosen more representative kinetochores for the insets of CC2* and Chimera in Figure 7A.

- In line with the previous point, could the authors tether each Spindly mutant to the kinetochore for example by fusing the construct to known kinetochores proteins such as Mis12 and test whether these fusion constructs are now able to recruit Dynactin to kinetochores?

This would be a potentially interesting experiment. However, reasoning that Spindly is a strong dimer that needs to interact with another strong hexamer like the RZZ complex, discouraged us as these stoichiometries would almost certainly complicate the interpretation of these experiments. It is clear that further work will be required to define the complete picture for this complex system.

- The authors conclude that the 2-step or multistep mechanism involved in the regulation of Spindly activation may be a common mechanism to different DD adaptors. However, the authors point out to existing differences between the conformational arrangement of Spindly and another DD adaptor, BICD2, arguing against a common mode of regulation for all adaptors. This needs to be clarified.

The reviewer has a good point and we have indeed tuned this down. We have re-written the last sentence of the abstract and replaced it with "Thus, our work illustrates how Dynein can be specifically activated at a defined cellular locale." We also write (line 592): Whether a similar 2-step or multistep mechanism applies to additional cargo-adaptor systems is an important question for future studies.

Minor concerns:

- In Fig.2D, full length Spindly does not bind DD in vitro. This is most likely to occur because Spindly N-terminus adopts a compacted conformation and hinders the access to DD-binding motifs. In Fig.2B, the authors show a structural prediction for Spindly 1-275 which should adopt a more elongated shape. According to prevailing model, this construct should now be able to bind DD in a similar biochemical assay.

We agree with the reviewer that Spindly¹⁻²⁷⁵ (and Spindly^{Δ276-306}) might be expected to be strong DD binders based on our model. Indeed, these proteins bind to the PE, albeit apparently weakly. Nevertheless, as explained in lines 350 and following, these mutants appear to form higher oligomers and we have not been able to show convincingly that they are fully open and available to bind DD.

- In Gama et al 2017, LIC1 was able to pull down a wild-type N-terminal Spindly construct. How do the authors reconcile this with the data presented in this manuscript?

We have expanded the discussion, also to answer major point 5, reviewer 1, on line 446 and following, where we also refer to the observation of Gama et al. 2017.

- The section where the authors test point mutations to open Spindly ("Opening up Spindly with point mutations") should be better contextualized. The transition is difficult to follow as it is.

We have now rephrased this part of the text to make our thoughts clearer.

- In the text, it is not clear whether Mps1 kinase is required to promote RZZ oligomerization in

the presence of Spindly chimera, an uninhibited Spindly mutant. According to the model, this mutant construct should drive oligomerization independently of Mps1 (as the N-terminal deletion construct from Sacristan et al 2018).

The reviewer is correct and we have rephrased this part of the text to clarify

- The nomenclature the authors adopt for the CC1 second conserved motif (SCM) and for the Spindly motif (SM) can be confusing at some point when identifying each mutant in the text and figures. Nomenclature should be standardized.

We agree with the reviewer and have now adopted a different nomenclature for the CC2 box or second conserved motif, namely HBS1, for Heavy Chain Binding Site 1. This functional annotation derives from work of one of our laboratories (Carter) and has been discussed with, and approved by, Geert Kops, whose laboratory had originally proposed the name "CC2 box", as well as Reto Gassmann, Erika Holzbaaur, Roberto Dominguez, Sam Reck Peterson, Rick McKenney, and Ahmet Yildiz.

- In Fig.6A, mCh-Spindly 33-605 and mCh-Spindly chimera lines have the same color.

Thank you for spotting this subtle mistake. We have corrected the color line.

Significance

The work represents a significant advance in the field and it would be of interest for a wide range of audiences.

July 20, 2022

RE: JCB Manuscript #202206131T

Prof. Andrea Musacchio
Max Planck Institute of Molecular Physiology
Otto Hahn Strasse 11
Dortmund 44227
Germany

Dear Prof. Musacchio:

Thank you for submitting your revised manuscript entitled "Conformational transitions of the Spindly adaptor underlie its interaction with Dynein and Dynactin". The reviewers have now assessed your revised manuscript and are satisfied with revisions. Thus we would be happy to publish your paper in JCB pending minor revisions to address the two minor points raised by Reviewer #1 and pending final revisions necessary to meet our formatting guidelines (see details below).

To avoid unnecessary delays in the acceptance and publication of your paper, please read the following information carefully. Please go through all the formatting points paying special attention to those marked with asterisks.

A. MANUSCRIPT ORGANIZATION AND FORMATTING:

1) Text limits: Character count for Articles and Tools is < 40,000, not including spaces. Count includes title page, abstract, introduction, results, discussion, and acknowledgments. Count does not include materials and methods, figure legends, references, tables, or supplemental legends.

2) Figures limits: Articles and Tools may have up to 10 main text figures.

***** Please note that main text figures should be provided as individual, editable files.**

3) Figure formatting:

***** Molecular weight or nucleic acid size markers must be included on all gel electrophoresis. Please, include MWs in supplemental Figs. 2A and 2C.**

***** Scale bars must be present on all microscopy images, including inset magnifications. Please include scale bars in inset magnifications in Figs. 7A, 7F and supplemental Fig. 7B.**

Also, please avoid pairing red and green for images and graphs to ensure legibility for color-blind readers. If red and green are paired for images, please ensure that the particular red and green hues used in micrographs are distinctive with any of the colorblind types. If not, please modify colors accordingly or provide separate images of the individual channels.

4) Statistical analysis:

***** Error bars on graphic representations of numerical data must be clearly described in the figure legend. Please describe error bars in Figs. 5J, 7B-D and supplemental Fig. 7D-H.**

***** The number of independent data points (n) represented in a graph must be indicated in the legend. Please, indicate whether N refers to technical or biological replicates (i.e. number of analyzed kinetochores or cells, number of independent experiments) in Figs. 5J, 7B-D and supplemental Fig. 7D-H.**

***** Statistical methods should be explained in full in the materials and methods in a separate section.**

For figures presenting pooled data the statistical measure should be defined in the figure legends.

*** Please also be sure to indicate the statistical tests used in each of your experiments (both in the figure legend itself and in a separate methods section) as well as the parameters of the test (for example, if you ran a t-test, please indicate if it was one- or two-sided, etc.). Please indicate the statistical test, including parameters, used in Figs. 5J, 7B-D and supplemental Fig. 7D-H.

*** If you used parametric tests in your study (i.e. t-tests), you should have first determined whether the data was normally distributed before selecting that test. In the stats section of the methods, please indicate how you tested for normality. If you did not test for normality, you must state something to the effect that "Data distribution was assumed to be normal but this was not formally tested."

5) Abstract and title:

The abstract should be no longer than 160 words and should communicate the significance of the paper for a general audience.

The title should be less than 100 characters including spaces. Make the title concise but accessible to a general readership.

6) Materials and methods:

Should be comprehensive and not simply reference a previous publication for details on how an experiment was performed. The text should not refer to methods "...as previously described."

Also, the materials and methods should be included with the main manuscript text and not in the supplementary materials.

7) *** Please be sure to provide the sequences for all of your primers/oligos and RNAi constructs in the materials and methods. Please indicate the sequence of the control RNAi construct.

You must also indicate in the methods the source, species, and catalog numbers (where appropriate) for all of your antibodies.

8) Microscope image acquisition:

The following information must be provided about the acquisition and processing of images:

- a. Make and model of microscope
- b. Type, magnification, and numerical aperture of the objective lenses
- c. Temperature
- d. imaging medium
- e. Fluorochromes
- f. Camera make and model
- g. Acquisition software
- h. Any software used for image processing subsequent to data acquisition. Please include details and types of operations involved (e.g., type of deconvolution, 3D reconstitutions, surface or volume rendering, gamma adjustments, etc.).

10) Supplemental materials:

*** There are strict limits on the allowable amount of supplemental data. Articles/Tools may have up to 5 supplemental figures. There is no limit for supplemental tables. You currently have 7 supplemental figures, which is fine with us in this case. Please, rename supplemental figures independently of the main figure which they are associated to, and correct the callouts in the text.

*** Please note that supplemental figures and tables should be provided as individual, editable files.

*** A summary of all supplemental material should appear at the end of the Materials and Methods section (please see any recent JCB paper for an example of this summary).

11) eTOC summary:

*** A ~40-50 word summary that describes the context and significance of the findings for a general readership should be included on the title page. The statement should be written in the present tense and refer to the work in the third person. It should begin with "First author name(s) et al..." to match our preferred style.

12) Conflict of interest statement:

JCB requires inclusion of a statement in the acknowledgements regarding competing financial interests. If no competing financial interests exist, please include the following statement: "The authors declare no competing financial interests."

13) A separate author contribution section is required following the Acknowledgments in all research manuscripts.

*** All authors should be mentioned and designated by their first and middle initials and full surnames and the CRediT nomenclature should be used (<https://casrai.org/credit/>).

14) ORCID IDs: ORCID IDs are unique identifiers allowing researchers to create a record of their various scholarly contributions in a single place. At resubmission of your final files, please consider providing an ORCID ID for as many contributing authors as possible.

15) Materials and data sharing:

All animal and human studies must be conducted in compliance with relevant local guidelines, such as the US Department of Health and Human Services Guide for the Care and Use of Laboratory Animals or MRC guidelines, and must be approved by the authors' Institutional Review Board(s). A statement to this effect with the name of the approving IRB(s) must be included in the Materials and Methods section.

*** As a condition of publication, authors must make protocols and unique materials (including, but not limited to, cloned DNAs; antibodies; bacterial, animal, or plant cells; and viruses) described in our published articles freely available upon request by researchers, who may use them in their own laboratory only. All materials must be made available on request and without undue delay. Please, indicate whether the reagents generated in this study have been deposited in public repositories. If not, please state that they would be made available to the scientific community upon request in the 'Data availability' section.

*** All datasets included in the manuscript must be available from the date of online publication, and the source code for all custom computational methods, apart from commercial software programs, must be made available either in a publicly available database or as supplemental materials hosted on the journal website. Numerous resources exist for data storage and sharing (see Data Deposition: <https://rupress.org/jcb/pages/data-deposition>), and you should choose the most appropriate venue based on your data type and/or community standard. If no appropriate specific database exists, please deposit your data to an appropriate publicly available database. Please, deposit your mass spectrometry data, EM maps and crystallographic structures in a proper public repository and include the accession number in the Materials and Methods.

16) Please note that JCB now requires authors to submit Source Data used to generate figures containing gels and Western blots with all revised manuscripts. This Source Data consists of fully uncropped and unprocessed images for each gel/blot displayed in the main and supplemental figures. The Source Data files will be directly linked to specific figures in the published article.

*** Since your paper includes cropped gel and/or blot images, please be sure to provide one Source Data file for each figure that contains gels and/or blots along with your revised manuscript files. File names for Source Data figures should be alphanumeric without any spaces or special characters (i.e., SourceDataF#, where F# refers to the associated main figure number or SourceDataFS# for those associated with Supplementary figures). The lanes of the gels/blots should be labeled as they are in the associated figure, the place where cropping was applied should be marked (with a box), and molecular weight/size standards should be labeled wherever possible.

B. FINAL FILES:

-- Cover images: If you have any striking images related to this story, we would be happy to consider them for inclusion on the

journal cover. Submitted images may also be chosen for highlighting on the journal table of contents or JCB homepage carousel. Images should be uploaded as TIFF or EPS files and must be at least 300 dpi resolution.

****It is JCB policy that if requested, original data images must be made available to the editors. Failure to provide original images upon request will result in unavoidable delays in publication. Please ensure that you have access to all original data images prior to final submission.****

****The license to publish form must be signed before your manuscript can be sent to production. A link to the electronic license to publish form will be sent to the corresponding author only. Please take a moment to check your funder requirements before choosing the appropriate license.****

Thank you for your attention to these final processing requirements. Please revise and format the manuscript and upload materials within 7 days. Please let us know if any complication preventing you from meeting this deadline arises and we can work with you to determine a suitable revision period.

Please contact the journal office with any questions, cellbio@rockefeller.edu.

Thank you for this interesting contribution, we look forward to publishing your paper in Journal of Cell Biology.

Sincerely,

Erika Holzbaur
Monitoring Editor
Journal of Cell Biology

Lucia Morgado-Palacin, PhD
Scientific Editor
Journal of Cell Biology

Reviewer #1 at Review Commons (Comments to the Authors (Required)):

In this revised manuscript, the authors have largely satisfied the concerns raised during the initial review. The authors have addressed many of the major and minor points that were raised with new experiments and have clarified figures and text. Although these changes have addressed my major comments, there are a few minor changes that I would suggest prior to publication.

First, the language describing the results from Figure 3C and E need to be tweaked slightly. The authors state, "... bound to the PE complex, as indicated by a clear, albeit only partial shift in its elution volume." Although I agree that there is a partial shift in the elution volume, this data is not clear, but rather supportive of weak PE binding as the authors state later in the manuscript and in their response.

Second, in figure 7A the authors point out that "This result was unexpected, because the CC1 box has been previously shown to be required for kinetochore localization of Dynactin". It would be helpful if the authors could comment in the text/discussion or supplement on why the results showing that mChSpindly33-605 recruits Dynactin to kinetochores differ from previously published work.

Overall, this revised manuscript is improved significantly and is ready for publication after these minor writing changes have been addressed.

Reviewer #3 at Review Commons (Comments to the Authors (Required)):

The Dynein-Dynactin (DD) complex interacts with different activating adaptors to assemble functional motor complexes capable of moving along microtubules while transporting various cargoes. However, it remains poorly understood how DD activation is precisely controlled so that Dynein-mediated transport is only stimulated at the appropriate time and place. DD adaptor

regulation is likely a crucial piece of this puzzle. In this manuscript, the authors show that Spindly, a mitotic adaptor of DD complex, undergoes a series of conformational rearrangements that result in efficient Spindly autoinhibition and affect its ability to bind DD. The work from d'Amico et al includes an impressive amount of biochemical and biophysical data, supported by well-designed experiments that are carefully documented. Resorting to crosslinking experiments and protein structural modelling, the authors find that several intramolecular contacts occur between specialized domains within Spindly N-terminus. The resulting compact conformation occludes important DD-binding motifs in Spindly and, thus, limits the access of DD to the adaptor. By utilizing different Spindly mutants predicted to render the adaptor more elongated, the authors bypass Spindly autoinhibition and rescue binding to DD in vitro. Surprisingly, unlike other DD adaptors, Spindly autoinhibition is not relieved upon binding to its cargo (the RZZ complex) arguing that the interaction with an additional binding partner is required to fully unleash the potential of Spindly to bind DD. In line with this, the authors identify a Spindly mutant that is unable to localize to kinetochores from human cells, despite its open conformation. Collectively, this work provides significant advances in the understanding of Spindly regulation and brings a new perspective to the mechanism of DD adaptor activation and therefore should be of interest for a wide audience.

All the main points addressed in this manuscript are very well supported by the data that is provided in the revised version of the manuscript.

The authors, in their response to the review process have addressed all the issues initially raised by the review process and have made changes to the manuscript, including additional experiments and modifications to the text that respond appropriately to the questions that were raised. In my view this manuscript should be accepted for publication without additional changes.

We are grateful to reviewers 1 and 3 for their support. While Reviewer 3 supported publication in the current form, Reviewer 1 expressed two residual concerns that we have now addressed.

Reviewer #1 at Review Commons (Comments to the Authors (Required)):

In this revised manuscript, the authors have largely satisfied the concerns raised during the initial review. The authors have addressed many of the major and minor points that were raised with new experiments and have clarified figures and text. Although these changes have addressed my major comments, there are a few minor changes that I would suggest prior to publication.

First, the language describing the results from Figure 3C and E need to be tweaked slightly. The authors state, "... bound to the PE complex, as indicated by a clear, albeit only partial shift in its elution volume." Although I agree that there is a partial shift in the elution volume, this data is not clear, but rather supportive of weak PE binding as the authors state later in the manuscript and in their response.

We have followed the reviewer's recommendation and we now write "...bound to the PE complex, albeit weakly, as indicated by a partial shift in its elution volume (Fig. 3C)."

A few lines later, when discussing Figure 3E, we now write: "Indeed, an ^{mCh}Spindly mutant lacking residues 276-306 (^{mCh}Spindly^{Δ276-306}), a segment already identified for its interactions with the CC1 box, also showed affinity for the PE complex"

Second, in figure 7A the authors point out that "This result was unexpected, because the CC1 box has been previously shown to be required for kinetochore localization of Dynactin". It would be helpful if the authors could comment in the text/discussion or supplement on why the results showing that mChSpindly33-605 recruits Dynactin to kinetochores differ from previously published work.

In the Discussion, we attempted to explain this behavior. We write "Nonetheless, the new results with mChSpindly33-605 suggest that the deletion of the CC1 box or its mutation result in fundamentally distinct behaviors, and support a role of the LIC subunits as triggers of adaptor opening more than as decisive contributors to the binding affinity of the interaction, a speculative conclusion that will require further investigation" The speculation is that deletion of the CC1 box contributes to adaptor opening and makes the interaction with the LIC subunit unnecessary.

Overall, this revised manuscript is improved significantly and is ready for publication after these minor writing changes have been addressed.

MANUSCRIPT ORGANIZATION AND FORMATTING:

1) Text limits: Character count for Articles and Tools is < 40,000, not including spaces. Count includes title page, abstract, introduction, results, discussion, and acknowledgments. Count does not include materials and methods, figure legends, references, tables, or supplemental legends.

The original character sum for the indicated sections was ~46000 characters and we have now brought it down to ~43000 by applying a lot of small edits. We hope this is acceptable.

2) Figures limits: Articles and Tools may have up to 10 main text figures.

We have 7 main figures

3) Figure formatting:

*** Molecular weight or nucleic acid size markers must be included on all gel electrophoresis. Please, include MWs in supplemental Figs. 2A and 2C.

Done

*** Scale bars must be present on all microscopy images, including inset magnifications. Please include scale bars in inset magnifications in Figs. 7A, 7F and supplemental Fig. 7B.

Done

Also, please avoid pairing red and green for images and graphs to ensure legibility for color-blind readers. If red and green are paired for images, please ensure that the particular red and green hues used in micrographs are distinctive with any of the colorblind types. If not, please modify colors accordingly or provide separate images of the individual channels.

Done

4) Statistical analysis:

*** Error bars on graphic representations of numerical data must be clearly described in the figure legend. Please describe error bars in Figs. 5J, 7B-D and supplemental Fig. 7D-H.

The error bars are now described for Figs. 5J and 7B-D. Supplemental Figure 7D-H (now Figure S7) does not contain error bars

*** The number of independent data points (n) represented in a graph must be indicated in the legend. Please, indicate whether N refers to technical or biological replicates (i.e. number of analyzed kinetochores or cells, number of independent experiments) in Figs. 5J, 7B-D and supplemental Fig. 7D-H.

We have included this information

*** Statistical methods should be explained in full in the materials and methods in a separate section.

We have now included a separate section on statistical methods in the material and methods section

For figures presenting pooled data the statistical measure should be defined in the figure legends.

*** Please also be sure to indicate the statistical tests used in each of your experiments (both in the figure legend itself and in a separate methods section) as well as the parameters of the test (for example, if you ran a t-test, please indicate if it was one- or two-sided, etc.). Please indicate the statistical test, including parameters, used in Figs. 5J, 7B-D and supplemental Fig. 7D-H.

Done

*** If you used parametric tests in your study (i.e. t-tests), you should have first determined whether the data was normally distributed before selecting that test. In the stats section of the methods, please indicate how you tested for normality. If you did not test for normality, you must state something to the effect that "Data distribution was assumed to be normal but this was not formally tested."

Done

5) Abstract and title:

The abstract should be no longer than 160 words and should communicate the significance of the paper for a general audience.

Abstract is 156 words

The title should be less than 100 characters including spaces. Make the title concise but accessible to a general readership.

Title is 99 characters including spaces

6) Materials and methods:

Should be comprehensive and not simply reference a previous publication for details on how an experiment was performed. The text should not refer to methods "...as previously described."

We abided by this recommendation

Also, the materials and methods should be included with the main manuscript text and not in the supplementary materials.

We abides by this recommendation

7) *** Please be sure to provide the sequences for all of your primers/oligos and RNAi constructs in the materials and methods. Please indicate the sequence of the control RNAi construct.

We now specify that we use only Opti-MEM (the most toxic reagent) in the control. We call these conditions +/- RNAi accordingly.

You must also indicate in the methods the source, species, and catalog numbers (where appropriate) for all of your antibodies.

The information is provided

8) Microscope image acquisition:

The following information must be provided about the acquisition and processing of images:

- a. Make and model of microscope
- b. Type, magnification, and numerical aperture of the objective lenses
- c. Temperature
- d. imaging medium

- e. Fluorochromes
- f. Camera make and model
- g. Acquisition software
- h. Any software used for image processing subsequent to data acquisition. Please include details and types of operations involved (e.g., type of deconvolution, 3D reconstitutions, surface or volume rendering, gamma adjustments, etc.).

We have included this information

References are formatted according to the JCB style

10) Supplemental materials:

*** There are strict limits on the allowable amount of supplemental data. Articles/Tools may have up to 5 supplemental figures. There is no limit for supplemental tables. You currently have 7 supplemental figures, which is fine with us in this case. Please, rename supplemental figures independently of the main figure which they are associated to, and correct the callouts in the text.

Thanks. We have renamed the supplemental figures as indicated

*** Please note that supplemental figures and tables should be provided as individual, editable files.

OK

*** A summary of all supplemental material should appear at the end of the Materials and Methods section (please see any recent JCB paper for an example of this summary).

The summary has been added.

11) eTOC summary:

*** A ~40-50 word summary that describes the context and significance of the findings for a general readership should be included on the title page. The statement should be written in the present tense and refer to the work in the third person. It should begin with "First author name(s) et al..." to match our preferred style.

The eTOC is provided

12) Conflict of interest statement:

JCB requires inclusion of a statement in the acknowledgements regarding competing financial interests. If no competing financial interests exist, please include the following statement: "The authors declare no competing financial interests."

Done

13) A separate author contribution section is required following the Acknowledgments in all research manuscripts.

*** All authors should be mentioned and designated by their first and middle initials and full surnames and the CRediT nomenclature should be used (<https://casrai.org/credit/>).

The section is included

14) ORCID IDs: ORCID IDs are unique identifiers allowing researchers to create a record of their various scholarly contributions in a single place. At resubmission of your final files, please consider providing an ORCID ID for as many contributing authors as possible.

We provide ORCID IDs where possible.

15) Materials and data sharing:

All animal and human studies must be conducted in compliance with relevant local guidelines, such as the US Department of Health and Human Services Guide for the Care and Use of Laboratory Animals or MRC guidelines, and must be approved by the authors' Institutional Review Board(s). A statement to this effect with the name of the approving IRB(s) must be included in the Materials and Methods section.

This is not applicable to this study

*** As a condition of publication, authors must make protocols and unique materials (including, but not limited to, cloned DNAs; antibodies; bacterial, animal, or plant cells; and viruses) described in our published articles freely available upon request by researchers, who may use them in their own laboratory only. All materials must be made available on request and without undue delay. Please, indicate whether the reagents generated in this study have been deposited in public repositories. If not, please state that they would be made available to the scientific community upon request in the 'Data availability' section.

We have included a Data Availability section

*** All datasets included in the manuscript must be available from the date of online publication, and the source code for all custom computational methods, apart from commercial software programs, must be made available either in a publicly available database or as supplemental materials hosted on the journal website. Numerous resources exist for data storage and sharing (see Data Deposition: <https://rupress.org/jcb/pages/data-deposition>), and you should choose the most appropriate venue based on your data type and/or community standard. If no appropriate specific database exists, please deposit your data to an appropriate publicly available database. Please, deposit your mass spectrometry data, EM maps and crystallographic structures in a proper public repository and include the accession number in the Materials and Methods.

Not applicable

16) Please note that JCB now requires authors to submit Source Data used to generate figures containing gels and Western blots with all revised manuscripts. This Source Data consists of fully uncropped and unprocessed images for each gel/blot displayed in the main and supplemental figures. The Source Data files will be directly linked to specific figures in the published article.

Check Verena's too Source Data have been compiled and uploaded